# Beijing Climate Center Earth System Model version 1 (BCC-ESM1): Model Description and Evaluation of Aerosol Simulations

**Tongwen Wu[1*], Fang Zhang[1], Jie Zhang[1], Weihua Jie[1], Yanwu Zhang[1], Fanghua Wu[1],**

**Laurent Li[1,2], Jinghui Yan[1], Xiaohong Liu[3], Xiao Lu[4], Haiyue Tan[4], Lin Zhang[4],**

**Jun Wang[5], Aixue Hu[6]**

[1]Beijing Climate Center, China Meteorological Administration, Beijing, China

[2]Laboratoire de Météorologie Dynamique, IPSL, CNRS, Sorbonne Université, Ecole Normale

Supérieure, Ecole Polytechnique, Paris, France

[3] Texas A&M University, College Station, TX, USA

[4] Laboratory for Climate and Ocean-Atmosphere Studies, Department of Atmospheric and

Oceanic Sciences, School of Physics, Peking University, Beijing, China

[5] University of Iowa, Iowa City, IA 52242, USA

[6]National Center for Atmospheric Research, PO Box 3000, Boulder, Colorado 80307-3000,

USA

*Correspondence to:* Tongwen Wu (twwu@cma.gov.cn)

Submit to Geosci. Model Dev.

**Revised on Oct. 23, 2019**

**Revised on Dec. 14, 2019**

**Revised on Jan. 17, 2020**

**Revised on Jan. 31, 2020**

**Abstract.** BCC-ESM1 is the first version of a fully-coupled Earth System Model with
interactive atmospheric chemistry and aerosols developed by the Beijing Climate Center,
China Meteorological Administration. Major aerosol species (including sulfate, organic
carbon, black carbon, dust and sea salt) and greenhouse gases are interactively simulated with
a whole panoply of processes controlling emission, transport, gas-phase chemical reactions,
secondary aerosol formation, gravitational settling, dry deposition, and wet scavenging by
clouds and precipitation. Effects of aerosols on radiation, cloud, and precipitation are fully
treated. The performance of BCC-ESM1 in simulating aerosols and their optical properties is
comprehensively evaluated as required by the Aerosol Chemistry Model Intercomparison
Project (AerChemMIP), covering the preindustrial mean state and time evolution from 1850
to 2014. The simulated aerosols from BCC-ESM1 are quite coherent with
CMIP5-recommended data, in-situ measurements from surface networks (such as IMPROVE
in the U.S. and EMEP in Europe), and aircraft observations. A comparison of modeled
aerosol optical depth (AOD) at 550 nm with satellite observations retrieved from Moderate
Resolution Imaging Spectroradiometer (MODIS) and Multi-angle Imaging
SpectroRadiometer (MISR) and surface AOD observations from AErosol RObotic NETwork
(AERONET) shows reasonable agreements between simulated and observed AOD. However,
BCC-ESM1 shows weaker upward transport of aerosols from the surface to the middle and
upper troposphere, likely reflecting the deficiency of representing deep convective transport
of chemical species in BCC-ESM1. With an overall good agreement between BCC-ESM1
simulated and observed aerosol properties, it demonstrates a success of the implementation of
interactive aerosol and atmospheric chemistry in BCC-ESM1.

## 1. Introduction

Atmosphere is a thin gaseous layer around the Earth, consisting of nitrogen, oxygen and a large number of trace gases including important greenhouse gases (GHG) such as water vapor, tropospheric ozone ($O_3$), carbon dioxide ($CO_2$), methane ($CH_4$), nitrous oxide ($N_2O$), and chloro-fluoro-carbons (CFCs). Besides gaseous components, atmosphere also contains various aerosols, which are important for cloud formation and radiative transfer. Atmospheric trace gases and aerosols are actually interactive components of the climate system. Their inclusion in global climate models (GCMs) is a significant enhancement for most state-of-the-art climate models (Lamarque et al., 2013; Collins et al., 2017). Early attempts in coupling global climate dynamics with atmospheric chemistry can be traced back to late 1970s, when 3D transport of ozone and simple stratospheric chemistry were firstly incorporated into a GCM to simulate global $O_3$ production and transport (e.g., Cunnold et al. 1975; Schlesinger and Mintz 1979). Since mid-1980s, a large number of on-line global climate/chemistry models have been developed to address issues of the Antarctic stratospheric $O_3$ depletion (e.g., Cariolle et al. 1990; Austin et al. 1992; Solomon, 1999), tropospheric $O_3$ and sulfur cycle (e.g., Feichter et al. 1996; Barth et al. 2000), tropospheric aerosol and its interactions with cloud (e.g., Chuang et al. 1997; Lohmann et al. 2000; Ghan and Easter, 2006; Jacobson 2012). Aerosols and chemically reactive gases in the atmosphere exert important influences on global and regional air quality and climate (Collins et al., 2017).

Since 2013, the Beijing Climate Center (BCC), China Meteorological Administration, has continuously developed and updated its fully-coupled GCM, the Beijing Climate Center Climate System Model (BCC-CSM) (Wu et al., 2013; Wu et al., 2014; Wu et al., 2019). BCC-CSM version 1.1 was one of the comprehensive carbon-climate models participating in the phase five of the Coupled Model Intercomparison Project (CMIP5, Taylor et al. 2012). When forced by prescribed historical emissions of $CO_2$ from combustion of fossil fuels and land use change, BCC-CSM1.1 successfully reproduced the trends of observed atmospheric $CO_2$ concentration and global surface air temperature from 1850 to 2005 (Wu et al., 2013). During recent years, BCC-CSM1.1 has been used in numerous investigations on soil organic carbon changes (e.g. Todd-Brown et al., 2014), ocean biogeochemistry changes (e.g. Mora et al., 2013), and carbon-climate feedbacks (e.g. Arora et al., 2013; Hoffman et al., 2014).

BCC-CSM includes main climate-carbon cycle processes (Wu et al., 2013) and the global
mean atmospheric $CO_2$ concentration is calculated from a prognostic equation of $CO_2$ budget
taking into account global anthropogenic $CO_2$ emissions and interactive land-atmosphere and
ocean-atmosphere CO2 exchanges.
In recent years, BCC has put large efforts in developing a global
climate-chemistry-aerosol fully-coupled Earth System Model (BCC-ESM1) on the basis of
BCC-CSM2 (Wu et al., 2019). The objective is to interactively simulate global aerosols (e.g.
sulfate, black carbon, etc.) and main greenhouse gases (e.g. $O_3$, $CH_4$, $N_2O$ and $CO_2$) in the
atmosphere and to investigate feedbacks between climate and atmospheric chemistry.
BCC-ESM1 is at the point to be publicly released, and it is actively used by BCC for several
CMIP6-endorsed research initiatives (Eyring et al. 2016), including the Aerosol Chemistry
Model Intercomparison Project (AerChemMIP, Collins et al., 2017) and the Coupled
Climate–Carbon Cycle Model Intercomparison Project (C4MIP, Jones et al. 2016).
The purpose of this paper is to evaluate the performance of BCC-ESM1 in simulating
aerosols and their optical properties in the 20th century. The description of BCC-ESM1 is
presented in Section 2. The experimental protocol is given in Section 3. Section 4 presents the
evaluations of aerosol simulations with comparisons to CMIP5-recommended data (Lamarque
et al., 2010) and data obtained from both global surface networks and satellite observations.
The regional and global characteristics compared to observations and estimates from other
studies are analyzed. Simulations of aerosol optical properties in the 20th century are also
analyzed in Section 4. Conclusions and discussions are summarized in Section 5. Information
about code and data availability is given in Section 6.
**2.  Model description**
BCC-ESM1 is an Earth System Model with interactive chemistry and aerosol
components, in which the atmospheric component is BCC Atmospheric General Model
version 3 (Wu et al., 2019) with interactive atmospheric chemistry (hereafter
BCC-AGCM3-Chem), land component BCC Atmosphere and Vegetation Interaction Model
version 2.0 (hereafter BCC-AVIM2.0), ocean component Modular Ocean Model version 4
(MOM4)-L40, and sea ice component [sea ice simulator (SIS)]. Different components of
BCC-ESM1 are fully coupled and interact with each other through fluxes of momentum,

energy, water, carbon and other tracers at their interfaces. The coupling between the atmosphere and the ocean is done every hour.

The atmospheric component BCC-AGCM3-Chem is able to simulate global atmospheric composition and aerosols from anthropogenic emissions as forcing agents. Its resolution is T42 (approximately $2.8125° \times 2.8125°$ transformed spectral grid). The model has 26 levels in a hybrid sigma/pressure vertical coordinate system with the top level at 2.914 hPa. Details of the model physics are described in Wu et al. (2019). The BCC-AGCM3-Chem combines 66 gas-phase chemical species and 13 bulk aerosol compounds as listed in Table 1. Apart from 3 gas-phase species of dimethyl sulfide (DMS), sulfur dioxide ($SO_2$) and ammonia ($NH_3$), the other 63 gas-phase species are the same as those in the "standard version" of MOZART2 (Model for Ozone and Related chemical Tracers, version 2), a global chemical transport model for the troposphere developed by the National Center for Atmospheric Research (NCAR) driven by meteorological fields from either climate models or assimilations of meteorological observations (Horowitz et al., 2003). Advection of all tracers in BCC-AGCM3-Chem is performed through a semi-Lagrangian scheme (Williamson and Rasch, 1989), and vertical diffusion within the boundary layer follows the parameterization of Holtslag and Boville (1993). The gas-phase chemistry of the 63 MOZART2 gas-phase species as listed in Table 1 is treated in the same way as that in the "standard version" of MOZART2 (Horowitz et al., 2003), and there are 33 photolytic reactions and 135 chemical reactions involving 30 dry deposited chemical species and 25 soluble gas-phase species. Dry deposition velocities for the 15 trace gases including $O_3$, carbon monoxide (CO), $CH_4$, formaldehyde ($CH_2O$), acetic acid ($CH_3OOH$), hydrogen peroxide ($H_2O_2$), nitrogen dioxide ($NO_2$), nitric acid ($HNO_3$), polyacrylonitrile (PAN), acetone ($CH_3COCH_3$), peroxyacetic acid ($CH_3COOOH$), acetaldehyde ($CH_3CHO$), methylglyoxal ($CH_3COCHO$), nitric oxid (NO), and pernitric acid ($HNO_4$) are not computed interactively and directly interpolated from MOZART2 climatological monthly mean deposition velocities (https://en.wikipedia.org/wiki/MOZART(model)) which are calculated offline (Bey et al., 2001; Shindell et al., 2008) using a resistance-in-series scheme originally described in Wesely (1989). The dry deposition velocities for the other 15 species including peroxy acetyl nitrate (PAN), methyl nitroacetate (ONIT), organic nitrates (ONITR), ethyl alcohol ($C_2H_5OH$), organic

hydroxiperoxide (POOH), ethyl hydroperoxide ($C_2H_5OOH$), propylhydroperoxide
($C_3H_7OOH$), methylene glycol mono acetate (ROOH), glycolaldehyde (GLYALD), acetol
(HYAC), methanol ($CH_3OH$), propanoic acid (MACROOH), isoprene hydroxy hydroperoxide
(ISOPOOH), carboxylic acid (XOOH), formaldehyde (HYDRALD), and hydrogen ($H_2$) are
calculated using prescribed deposition velocities of $O_3$, CO, $CH_3CHO$, or land surface type
and surface temperature following the MOZART2 (Horowitz et al., 2003). Wet removal by
in-cloud scavenging for 25 soluble gas-phase species in the "standard version" of MOZART2
uses the parameterization of Giorgi and Chameides (1985) based on their temperature
dependent effective Henry's law constants. In-cloud scavenging is proportional to the amount
of cloud condensate converted to precipitation, and the loss rate depends on the amount of
cloud water, the rate of precipitation formation, and the rate of tracer uptake by the liquid
phase water. Other highly soluble species such as $HNO_3$, $H_2O_2$, ONIT, ISOPOOH,
MACROOH, XOOH, and lead (Pb-210) are also removed by below-cloud washout as
calculated using the formulation of Brasseur et al. (1998). Below-cloud scavenging is
proportional to the precipitation flux in each model layer and the loss rate depends on the
precipitation rate. Vertical transport of gas tracers and aerosols due to deep convection is not
yet included in the present version of BCC-AGCM3-Chem, which process is considered as a
part of the deep convection and occurs generally in a small spatial region on a GCM-box with
low-resolution (2.8 ˚lat.×2.8 ˚lon.). Another consideration is that a large uncertainty exists to
treat transport of those water-soluble tracers by deep convection. But this effect will be
involved in the next version of BCC model.
The BCC-AVIM2.0 is the land model with terrestrial carbon cycle. It is described in
details in Li et al. (2019) and includes biophysical, physiological, and soil carbon-nitrogen
dynamical processes. The terrestrial carbon cycle operates through a series of biochemical
and physiological processes on photosynthesis and respiration of vegetation. Biogenic
emissions from vegetation are computed online in BCC-AVIM2.0 following the algorithm of
the Model of Emissions of Gases and Aerosols from Nature version 2.1 (MEGAN2.1,
Guenther et al., 2012).
The oceanic component of BCC-ESM1 is the Modular Ocean Model version 4 with 40
levels (hereafter MOM4-L40), and the sea ice component Sea Ice Simulator (SIS).
MOM4-L40 uses a tripolar grid of horizontal resolution with 1 ° longitude by 1/3 ° latitude
between 30 °S and 30 °N ranged to 1 ° longitude by 1 ° latitude from 60 °S and 60 °N poleward
and 40 z-levels in the vertical. Carbon exchange between the atmosphere and the ocean are
calculated online in MOM4-L40 using a biogeochemistry module that is based on the
protocols from the Ocean Carbon Cycle Model Intercomparison Project–Phase 2 (OCMIP2,
http://www.ipsl.jussieu.fr/OCMIP/phase2/). SIS has the same horizontal resolution as
MOM4-L40 and three layers in the vertical, including one layer of snow cover and two layers
of equally sized sea ice. Details of oceanic component MOM4-L40 and sea-ice component
SIS that are used in BCC-ESM1 may be found in Wu et al. (2013) and Wu et al. (2019).
In the following sub-sections, we will describe the treatments in BCC-ESM1 for 3
gas-phase species of DMS, $SO_2$ and $NH_3$, 13 prognostic aerosol species including sulfate
($SO_4^{2-}$), 2 types of organic carbon (hydrophobic OC1, hydrophilic OC2), 2 types of black
carbon (hydrophobic BC1, hydrophilic BC2), 4 categories of soil dust (DST01, DST02,
DST03, DST04), and 4 categories of sea salt (SSLT01, SSLT02, SSLT03, SSLT04).
Concentrations of all aerosols in BCC-ESM1 are mainly determined by advective transport,
emission, dry deposition, gravitational settling, and wet scavenging by clouds and
precipitation, except for $SO_4^{2-}$ which gas-phase and aqueous phase conversion from $SO_2$ are
also considered. The present version of aerosol scheme belongs to a bulk aerosol model and
mainly refers to the scheme of CAM-Chem (Lamarque et al., 2012), but the nucleation and
coagulation of aerosols are still ignored.
**2.1 $SO_2$, DMS, $NH_3$, and Sulfate**
$SO_2$ is a main sulfuric acid precursor to form aerosol sulfate $SO_4^{2-}$. Conversions of $SO_2$
to $SO_4^{2-}$ occur by gas phase reactions (Table 2) and by aqueous phase reactions in cloud
droplets. The dry deposition velocity of $SO_2$ follows the resistance-in-series approach of
Wesely (1989) using the formula, $W_{SO2} = 1/(r_a + r_b + r_c)$, in which $r_a$, $r_b$, and $r_c$ are the
aerodynamic resistance, the quasi-laminar boundary layer resistance, and the surface
resistance, respectively and they are interactively computed in each model time step. The loss
rate of $SO_2$ due to wet deposition is computed following the scheme in the global Community
Atmosphere Model (CAM) version 4, the atmospheric component of the Community Earth
System Model (Lamarque et al., 2012).
The sources of SO$_2$ mainly come from fuel combustion, industrial activities, and
volcanoes. SO$_2$ can also be formed from the oxidation of DMS as listed in Table 2 in which
their reaction rates follow CAM-Chem (Lamarque et al. 2012). The main source of DMS is
from oceanic emissions via biogenic processes. It is prescribed with the climatological
monthly data that are extracted from MOZART2 package
(https://www2.acom.ucar.edu/gcm/mozart-4). SO$_4^{2-}$ is one of the prognostic aerosols in
BCC-AGCM3-Chem. Its treatment follows CAM4-Chem (Lamarque et al., 2012). It is
produced primarily by the gas-phase oxidation of SO$_2$ (in Table 2) and by aqueous phase
oxidation of SO$_2$ in cloud droplets. The gas phase reactions, rate constants, and gas-aqueous
equilibrium constants are given by Tie et al. (2001). The heterogeneous reactions of SO$_4^{2-}$
occur on all aerosol surfaces. Their treatment follows a Bulk Aerosol Model (BAM) used in
CAM4 (Neale et al., 2010). The heterogeneous reactions depend strongly on pH values in
clouds which are calculated from the concentrations of SO$_2$, HNO$_3$, H$_2$O$_2$, NH$_3$, O$_3$, HO$_2$, and
SO$_4^{2-}$. NH$_3$ is a gas tracer apart from MOZART2 (Table 1). Its sources include aircraft and
surface emissions due to anthropogenic activity, biomass burning, and biogenic emissions
from land soil and ocean surfaces (Table 4). SO$_4^{2-}$ is assumed to be all in aqueous phase due
to water uptake, although Wang et al. (2008a) showed that ~34% of sulfate particles are in
solid phase globally due to the hysteresis effect of ammonium sulfate phase transition.
However, in terms of radiative forcing, consideration of solid sulfate formation process
lowers the sulfate forcing by ~8% as compared to consideration of all sulfate particles in
aqueous phase (Wang et al., 2008b). Future model development may consider the life cycle of
NH$_3$. The sulfate in- and below-cloud scavenging follows Neu and Prather (2011). Washout
of SO$_4^{2-}$ is set to 20% of the washout rate of HNO$_3$ following Tie et al. (2005) and Horowitz
(2006). Dry deposition velocity of SO$_4^{2-}$ is also calculated by the resistance-in-series
approach.
**2.2 Aerosols of organic carbon and black carbon**
BCC-AGCM3-Chem treats two types of organic carbon (OC), i.e. water-insoluble tracer
OC1 and water-soluble tracer OC2, and two types of black carbon (BC), i.e. water-insoluble
tracer BC1 and water-soluble tracer BC2. As shown in Table 2, hydrophobic BC1 and OC1
can be converted to hydrophilic BC2 and OC2 with a constant rate of $7.1 \times 10^{-6}$ s$^{-1}$ (Cooke and
Wilson, 1996). The 4 tracers of organic carbon and black carbon are mainly from emissions
including both fossil fuel and biomass burning, and are from the CMIP6 data package
(https://esgf-node.llnl.gov/search/input4mips/, Hoesly et al., 2018). Beside anthropogenic and
biomass burning emissions, hydrophilic organic carbon OC2 can also come from natural
biogenic volatile organic compound (VOC) emissions. Dry deposition velocities for all the 4
OC and BC tracers are set to $0.001 m.s^{-1}$. OC2 and BC2 are soluble aerosols, and their sinks
are primarily governed by wet deposition. Their in- and below-cloud scavenging follows the
scheme of Neu and Prather (2011).
**2.3 Sea salt aerosols**
As shown in Table 3, sea salt aerosols in the model are classified into four size bins (0.2–
1.0, 1.0–3.0, 3.0–10, and 10–20 μm) in diameter. They originate from oceans and are
calculated online by BCC-ESM1. The upward flux $F_{sea-salt}$ of sea salt productions for four
bins is proportional to the 3.41 power of the wind speed $u_{10m}$ at 10 m height near the sea
surface (Mahowald et al., 2006) and is expressed as
$$F_{sea-salt} = S \cdot (u_{10m})^{3.41}, \tag{1}$$
where $S$ is a scaling factor and set to $4.05 \times 10^{-15}$, $4.52 \times 10^{-14}$, $1.15 \times 10^{-13}$, $1.20 \times 10^{-13}$ for four
size bins of sea salt aerosols in BCC-ESM1, respectively.
Dry deposition of sea salts depends on the turbulent deposition velocity in the lowest
atmospheric layer using aerodynamic resistance and the friction velocity, and the settling
velocity through the whole atmospheric column for each bin of sea salts. The turbulent
deposition velocity and settling velocity depend on particle diameter and density (listed in
Table 3). In addition, the fact that the size of sea salts changes with humidity is also
considered. The wet deposition of sea salts follows the scheme for soluble aerosols used in
CAM4, and depends on prescribed solubility and size-independent scavenging coefficients.
**2.4 Dust aerosols**
Dust aerosols behave in a similar way as sea salts. Their variations involve three major
processes: emission, advective transport, and wet/dry depositions. The dust emission is based
on a saltation-sandblasting process, and depends on wind friction velocity, soil moisture, and
vegetation/snow cover (Zender et al., 2003). The vertical flux of dust emission is corrected by
a surface erodible factor at each model grid cell which has been downloaded from NCAR
website    (https://svn-ccsm-inputdata.cgd.ucar.edu/trunk/inputdata/atm/cam/dst/).    Soil
erodibility is prescribed by a physically-based geomorphic index that is proportional to the
runoff area upstream of each source region (Albani et al., 2014). Like sea salts, dry deposition
of dust aerosols includes gravitational and turbulent deposition processes, while wet
deposition results from both convective and large scale precipitation and is dependent on
prescribed size-independent scavenging coefficients.
**2.5 Effects of aerosols on radiation, clouds, and precipitation**

The mass mixing ratios of bulk aerosols are prognostic variables in BCC-ESM1 and

directly affect the radiative transfer in the atmosphere with their treatments following the
NCAR Community Atmosphere Model (CAM3, Collins et al., 2004). Indirect effects of
aerosols are taken into account in the present version of BCC-AGCM3-Chem (Wu et al.,
2019). Aerosol particles act as cloud condensation nuclei and exert influence on cloud
properties and precipitation, and ultimately impact the hydrological cycle. Prognostic aerosol
masses are used to estimate the liquid cloud droplet number concentration $N_{cdnc}$ (cm$^{-3}$) in
BCC-AGCM3-Chem. $N_{cdnc}$ is explicitly calculated using the empirical function suggested
by Boucher and Lohmann (1995) and Quaas et al. (2006):

$$N_{cdnc} = \exp\left[5.1 + 0.41\ln\left(m_{aero}\right)\right] \tag{2}$$

where $m_{aero}$ ($\mu$ g.m$^{-3}$) is the total mass of all hydrophilic aerosols,

$$m_{aero} = m_{SS} + m_{OC} + m_{SO_4} + m_{NH_4NO_2}, \tag{3}$$

i.e. the first bin of sea salt (m$_{SS}$), hydrophilic organic carbon ($m_{OC}$), sulphate ($m_{SO_4}$), and
Ammonium nitrite (NH$_4$NO$_2$). A dataset of NH$_4$NO$_2$ from NCAR CAM-Chem (Lamarque et
al., 2012) is used in our model.

$N_{cdnc}$ is an important factor in determining the effective radius of cloud droplets for

radiative calculation. The effective radius of cloud droplets $r_{el}$ is estimated as

$$r_{el} = \beta \cdot r_{l,vol}, \tag{4}$$

where $\beta$ is a parameter dependent on the droplets spectral shape and follows the calculation
proposed by Peng and Lohmann (2003),

$$\beta = 0.00084 N_{cdnc} + 1.22. \tag{5}$$

$r_{l,vol}$ is the volume-weighted mean cloud droplet radius,
$$r_{l,vol} = \left[ (3LWC)/(4\pi\rho_w N_{cdnc}) \right]^{1/3}, \qquad (6)$$

where $\rho_w$ is the liquid water density and $LWC$ the cloud liquid water content (g cm$^{-3}$).

Aerosols also exert impacts on precipitation efficiency (Albrecht, 1989), which is taken

into account in the parameterization of non-convective cloud processes. There are five
processes that convert condensate to precipitate: auto-conversion of liquid water to rain,
collection of cloud water by rain, auto-conversion of ice to snow, collection of ice by snow,
and collection of liquid by snow. The auto-conversion of cloud liquid water to rain (*PWAUT*)
is dependent on the cloud droplet number concentration and follows a formula that was
originally suggested by Chen and Cotton (1987),
$$PWAUT = C_{l,aut} \, \hat{q}_l^2 \, \rho_a / \rho_w \left( \frac{q_l \rho_a}{\rho_w N_{ncdc}} \right)^{1/3} H\left(r_{l,vol} - r_{lc,vol}\right) \qquad (7)$$

Where $\hat{q}_l$ is in-cloud liquid water mixing ratio, $\rho_a$ and $\rho_w$ are the local densities of air and
water respectively, and $C_{l,aut}$ is a constant. $H(x)$ is the Heaviside step function with the
definition,
$$H\left(x\right) = \begin{cases} 0, & x < 0 \\ 1, & x \geq 0 \end{cases}. \qquad (8)$$

$r_{lc,vol}$ is the critical value of mean volume radius of liquid cloud droplets $r_{l,vol}$, and set to 15
μ m.

The treatment of aerosol single scattering (optical) properties (such as mass extinction

efficiency, single scattering albedo, and asymmetric factor) follows the look-up table
approach in CAM (Collins et al., 2004). The optics for black, organic carbon, sea salt, and sea
salt particles is assumed to be same as the optics for soot and water-soluble aerosols in the
Optical Properties of Aerosols and Clouds (OPAC) data set (Hess et al., 1998). The optics for
dust is derived by Mie calculations for the size distribution represented by each size bin
(Zender et al., 2003). Similarly, for sulfate and nitrate particles, same set of aerosol optical
properties for ammonium sulfate are used and are taken from Wang et al. (2008b) with
treatment of aerosol hygroscopicity. The volcanic stratospheric aerosols are assumed to be
comprised of 75% sulfuric acid and 25% water, as in Hess et al. (1998). For each model year,
different aerosol types are assumed to be externally mixed in the calculation of bulk aerosol
single scattering properties that are in turn used in the radiative transfer calculations.

## 3. Experiment design for the 20[th] century climate simulation

There is an Aerosol Chemistry Model Intercomparison Project (AerChemMIP, Collins et

al., 2017) endorsed by the Coupled-Model Intercomparison Project 6 (CMIP6) for
documenting and understanding past and future changes in the chemical composition of the
atmosphere, and estimating the global-to-regional climate response from these changes.
Modelling groups with full chemistry and aerosol models are encouraged to perform all
AerChemMIP simulations (Collins et al., 2017). To assess the ability of our model to simulate
aerosols (mean and variability), we have followed the historical simulation designed by
CMIP6 (Eyring et al., 2016) which is named as "historical" experiment in the Earth System
Grid Federation (ESGF). The historical experiment is forced with emissions evolving from
1850 to 2014 that include biomass burning emissions (Van Marle et al. 2017), anthropogenic
and open burning emissions (Hoesly et al., 2018; Feng et al., 2019). $O_3$ in the historical
simulation is an interactive prognostic variable and feedbacks on radiation, and the
concentrations of other WMGHG, e.g. $CH_4$, $N_2O$, $CO_2$, CFC11, and CFC12 are prescribed
using CMIP6 historical forcing data (Meinshausen et al., 2017). Although $CH_4$ and $N_2O$ are
prognostic variables in the chemistry scheme (Table 1), their prognostic values at each model
step in the historical experiment are replaced by CMIP6 data (Meinshausen et al., 2017)
throughout the model domain. The rest of historical forcing data include: (1) yearly global
gridded land-use forcing data sets (Hurtt et al., 2011; Hurtt et al., 2017), and (2) solar forcing
(Matthes et al., 2017). All these datasets were downloaded from
https://esgf-node.llnl.gov/search/input4mips/. Climate feedback processes that involve
changes to the atmospheric composition of reactive gases and aerosols may affect the
temperature response to a given WMGHG concentration level.

**3.1 Surface emissions**

Surface emissions of chemical species from different sources are summarized in Table

4. They include anthropogenic emissions from fossil fuel burning and other industrial
activities, biomass burning (including vegetation fires, fuel wood and agricultural burning),
biogenic emissions from vegetation and soils, and oceanic emissions. Most historical

emissions from anthropogenic source (surface, aircraft plus ship) and biomass burning from 1850 to 2014 are CMIP6-recommended data (available at https://esgf-node.llnl.gov/search/input4mips). Anthropogenic or biomass burning sources of some tracers which are not included in the CMIP6 dataset (see Table 4), anthropogenic emission of $H_2$ and $N_2O$ are from monthly climatological dataset provided by the MOZART-2 standard package. $N_2O$ is a prognostic variable in BCC-ESM1 but it is replaced by CMIP6 prescribed concentration in the historical run. Other emissions including biomass burning ($CH_3COCH_3$) and anthropogenic emission ($CH_3CHO$, $CH_3OH$, and $CH_3COCH_3$) are from the Atmospheric Chemistry and Climate Model Intercomparison Project (ACCMIP) emission inventory (http://accent.aero.jussieu.fr/ACCMIP.php) covering the period from 1850 to 2010 with 10-year intervals (see Table 4). Monthly lumped emissions of black carbon and organic carbon aerosols from 1850 to 2014 are downloaded from CMIP6-recommended data, but we used 80% (for BC) and 50% (for OC) of them in their hydrophobic forms (BC1 and OC1) and the rest in their hydrophilic forms (BC2 and OC2), following the work of Chin et al. (2002).

Five tracers of ISOP, ACET ($CH_3COCH_3$), $C_2H_4$, $C_3H_8$, and Monoterpenes ($C_{10}H_{16}$) in Table 1 belong to biogenic volatile organic carbons (VOCs). As shown in Table 4, those VOCs emissions are online calculated in BCC-ESM1 following the modeling framework of the Model of Emissions of Gases and Aerosols from Nature version 2.1 (MEGAN2.1, Guenther et al., 2012) using simple mechanistic algorithms to account for major known processes controlling biogenic emissions. The MEGAN2.1 can provide a flexible scheme for estimating 16 tracers of biogenic emissions from terrestrial ecosystems including five VOCs emissions used in BCC-ESM1 (Table 4). All the VOCs emissions depend on current and past surface air temperature, solar flux, and the landscape types. Their calculation requires global maps of plant functional type (PFT) and leaf area index (LAI) which is a prognostic variable from the land model BCC-AVIM2. The effect of atmospheric $CO_2$ concentration on isoprene emissions is included. 10% of the biogenic monoterpenes emissions as calculated online with the MEGAN2.1 algorithm in BCC-AVIM2 are converted to hydrophilic organic carbon (OC2) to account for formation of secondary organic aerosols following Chin et al. (2002) in this version of BCC-ESM1.

**3.2 Volcanic eruptions, lightning and aircraft emissions**

As there is no stratospheric aerosol scheme in BCC-ESM1, concentrations of sulfate
aerosol at heights from 5 to 39.5 km, which volcanic origin, are directly prescribed using the
CMIP6-recommended data (Thomasson et al., 2018) from 1850 to 2014. The effects of
surface $SO_2$ emissions from volcanic eruption on the variation of SO2 in the atmosphere and
then on the variation of tropospheric $SO_4^{2-}$ concentration are considered, and the $SO_2$
emissions from 1850 to 2014 are downloaded from the IPCC ACCMIP emission inventory
(http://accent.aero.jussieu.fr/ACCMIP.php). Aircraft emissions are provided for $NO_2$, CO,
$CH_4$, $NH_3$, NO, $SO_2$, and aerosols of OC and BC (Table 1). The emissions of NO from
lightning are online calculated in BCC-AGCM3-Chem following the parameterization in
MOZART2, and the globally-averaged mean during the period of 1850 to 2014 is 5.19
$Tg(N)$ $yr^{-1}$, which is in agreement with observations within the range of 3 to 6 $Tg(N)$ $yr^{-1}$
(Martin et al., 2002). The lightning frequency depends strongly on the convective cloud top
height, and the ratio of cloud-to-cloud versus cloud-to-ground lightning depends on the cold
cloud thickness from the level of 0℃ to the cloud top (Price and Rind, 1992).
**3.3 Upper boundary of the atmosphere**
As no stratospheric chemistry is included in the present version of BCC-AGCM3-Chem,
it is necessary to ensure a proper distribution of chemically-active stratospheric species.
Concentrations of different tracers ($O_3$, $CH_4$, $N_2O$, NO, $NO_2$, $HNO_3$, CO, and $N_2O_5$) at the top
two layers of the model are set to prescribed monthly climatological values, and
concentrations from below the top two layers to the tropopause are relaxed at a relaxation
time of 10-days towards the climatology. Climatological values of NO, $NO_2$, $HNO_3$, CO and
$N_2O_5$ at the top two layers are extracted from MOZART2 data package available at the
Website (https://www2.acom.ucar.edu/gcm/mozart-4), originated from the Study of Transport
and Chemical Reactions in the Stratosphere (STARS, Brasseur et al., 1997). Concentrations
for the other tracers ($O_3$, $CH_4$, and $N_2O$) at the top two model layers are the zonally-averaged
and monthly values from 1850 to 2014 derived from the CMIP6 data package.
**3.4 The preindustrial model states**
The preindustrial state of BCC-ESM1 is obtained from a piControl simulation of over 600
years in which all forcings including emissions data are fixed at 1850 conditions. The initial
state of the piControl simulation itself is obtained through individual spin-up runs of each
component of BCC-ESM1 in order for the piControl simulation to run stably and fast to reach
its equilibrium. Figures 1(a-c) show the time series of global yearly means of the net energy
budget at top of the atmosphere (TOA), near-surface air temperature (TAS), and sea surface
temperature (SST) from the piControl simulation for the last 450 years. It shows that the
surface climate in BCC-ESM1 nearly reaches its equilibrium after 600 years piControl
simulation. The whole system in BCC-ESM1 fluctuates around +0.7 Wm$^{-2}$ net energy flux at
TOA without obvious trend in 450 years (Fig. 1a). This level of TOA energy imbalance is
close to the average imbalance (1.0 Wm$^{-2}$) among CMIP5 models (Wild et al., 2013). It means
that there exists surplus energy of +0.7 Wm$^{-2}$ obtained by the whole system in BCC-ESM1,
but do not cause remarkable climate drift. The global mean TAS and SST keep around 288.1
K (Fig. 1b) and 295.05 K (Fig. 1c), respectively. During the last 450 years, there are ($\pm 0.2$ K
amplitude of TAS and SST) oscillations of centennial scale for the whole globe (Figs. 1b and
1c), which are certainly caused by internal variation of the system.
Figures 2a-2c show the time series of global annual total burdens of $SO_2$, DMS, and OH
in the troposphere (integrated from the surface to 100 hPa) in the last 450 years of the
piControl simulation. Without any anthropogenic source, the $SO_2$ amount in the troposphere
nearly keeps the level of 0.0868 Tg in the 450 years of the piControl simulation. Tropospheric
DMS varies around the value of 0.116 Tg. Tropospheric OH, as an important gas species
oxidizing $SO_2$ to form $SO_4^{2-}$ (Table 2), keeps at a stable level in the atmosphere. $SO_4^{2-}$ also
remains at a stable level of 0.556 Tg in the atmosphere in the whole period of the piControl
simulation (Figure 2d). The amounts of BC and OC in the troposphere vary around 0.0395 Tg
and 0.275 Tg (Figures 2e-2f), respectively. Dust and sea salt aerosols are at the level of 22 Tg
and 11.7 Tg (Figures 2g-2h), respectively. All those data are close to the global mean
concentrations of 0.604 Tg $SO_4^{2-}$, 0.046 Tg BC, 0.30 Tg OC, 22.18 Tg dust, and 11.73 Tg sea
salts in 1850 which are estimated based on the CMIP5 prescribed data in 1850 (Lamarque et
al., 2010).
Figure 3 shows the global spatial distributions of annual mean sulfate, organic carbon,
black carbon, dust, and sea salt aerosols in the whole atmospheric column averaged for the
last 100 years of the piControl simulation of BCC-ESM. We can compare them with CMIP5
recommended concentrations in year 1850, considered as the reference state in the
pre-industrial stage. At that time, there are fewer anthropogenic/biomass $SO_2$ emissions, the
$SO_4^{2-}$ over land are evidently smaller than those over oceans especially over the tropical
Pacific and Atlantic Oceans, where DMS can be oxidized to $SO_2$ and then form $SO_4^{2-}$. There
are several centers of high values of black carbon and organic carbon in East and South Asia,
Europe, Southeast America, and in the tropical rain forests in Africa and South America.
They mainly result from biomass burning including vegetation fires, fuel wood and
agricultural burning. Dust aerosols are mainly distributed in North Africa, Central Asia, North
China, and Australia, where arid and semi-arid areas locate. Dust emitted from Sahara Desert
can be transported to the tropical Atlantic by easterly wind. The sea salt aerosols are mainly
distributed over the mid-latitude Southern Oceans, the tropical southern Indian Ocean, and the
tropical northern Pacific Ocean, where wind speeds near the sea surface are strong. As shown
in Fig. 3, all the spatial distribution patterns of CMIP5-derived sulfate, black carbon, organic
carbon, dust, and sea salt aerosols (Lamarque et al., 2010) are well simulated in BCC-ESM1.
There are high spatial correlation coefficients, 0.76 for sulfate, 0.77 for black carbon, 0.77 for
organic carbon, 0.94 for dust, and 0.94 for sea salts, between CMIP5 data and BCC-ESM1
simulations. Relative lower relations for sulfate, black carbon and organic carbon are possibly
caused as different anthropogenic emission sources are used in BCC-ESM1 and to create
CMIP5 data. Dust and sea salts belong to natural aerosols and depend on the land and sea
surface conditions, so their spatial distributions are easy to be captured and have relatively
higher correlations between CMIP5 data and BCC-ESM1 simulations.

**4.  Evaluation of $O_3$ and aerosol simulations in the 20[th] century**
The rate of sulfate formation is dependent on the levels of oxidants in the troposphere.
$O_3$ is an important oxidant. So, the evaluation of simulated tropospheric $O_3$ is helpful to
understand the aerosols simulations. BCC-ESM1 is driven by most of the
CMIP6-recommended emission data. As shown in Figure 4, the zonal distributions of the total
amounts of tropospheric $O_3$ below 300 hPa to the ground and their changes with time from
1850 to 2014 from the CMIP6-recommend dataset (Table 4) are well simulated by
BCC-ESM1. Evident increasing trends since 1850 almost exist in every latitudes, especially
in the Northern Hemisphere where the contents of tropospheric $O_3$ are higher than those in the
Southern Hemisphere.
Figure 5 shows the vertical profiles of $O_3$ simulations with comparison to global
ozonesonde observations averaged for the monthly data over 2010-2014 from the World
Ozone and Ultraviolet Radiation Data Centre (WOUDC; http://woudc.org/data.php, last
access: 24 September 2019) in nine regions which are averaged from 41 global WOUDC sites.
The details of WOUDC data may refer to Lu et al. (2019). As shown in Figure 5, BCC-ESM1
well captures the observed ozone vertical structure at all regions. At the lower and middle
troposphere (i.e. below 6 km), the model typically shows positive bias within 5 ppbv for the
Southern Hemisphere and 10 ppbv for the Northern mid-latitudes, similar to those simulated
from many other global atmospheric chemical models (Young et al., 2013, 2018). The model
has larger ozone overestimation in the upper troposphere and stratosphere at most regions, at
least partly due to the use of prescribed stratospheric ozone as upper boundary conditions
and/or errors in modeling ozone exchange between the stratosphere and the troposphere.
Global tropospheric ozone burden derived from our simulation is 335 Tg averaged over
2010-2014, in consistent with recent assessment from multi chemistry models (Young et al.,

2018).

**4.1 Global aerosols trends**
Figure 6(a)-(c) show the time series of global total emissions of $SO_2$, OC, and BC to the
atmosphere from natural and anthropogenic sources. Emissions of $SO_2$ are largely due to
industrial production. From 1850 to 1915, $SO_2$ emissions increased year by year as the
Industrial Revolution intensified and expanded. But from 1915 to 1945, the increase trend of
$SO_2$ emissions became slower as broke out the First and the Second World Wars. After that
period, with growing industrial productions, $SO_2$ emissions increased again and reached a
maximum around the end of 1970s. During the 1980s and 2000s, with a substantial decrease
of $SO_2$ emissions in Europe and the United States, the global $SO_2$ emissions has been
decreasing since the 1980s despite the rapid increase of $SO_2$ emissions in South and East Asia
as well as in developing countries in the Southern Hemisphere in recent years (Liu et al.,
2009). The OC and BC emissions substantially increased since 1950s just after the Second
World War. The global total OC emission in 2010 was nearly twice as much as that in
pre-industrial (year 1850) and increased by 18 Tg $\cdot$ yr$^{-1}$. Anthropogenic black carbon
emissions increased from 1 Tg yr$^{-1}$ in 1850 to nearly 8 Tg yr$^{-1}$ in 2010.
Anthropogenic $SO_2$, OC and BC emissions strongly affect the variations of atmospheric
concentrations of sulfate, OC, and BC. The global $0.5^o$x$0.5^o$ gridded data of
CMIP5-recommended aerosols masses with 10-years interval from 1850 to 2000 (Lamarque
et al., 2010) provides an important reference to evaluate the aerosol simulations in
BCC-ESM1. As shown in Figure 7b-7f, the annual total aerosol burdens of $SO_4^{2-}$, OC, and BC
in the whole atmosphere column as simulated by the BCC-ESM1 20$^{th}$ century historical
simulation are generally consistent with the values derived from CMIP5-recommended
aerosols concentrations. Due to increasing $SO_2$ emissions from 1850 to present day (Fig. 6),
the global $SO_2$ burden in the atmosphere increased from 100 Tg in 1850s to 200 Tg in 1980s
(Fig. 7a), and has a high correlation coefficient of 0.996 with the anthropogenic emissions
(Fig. 6a), as the lifetime of $SO_2$ is short. The burden directly followed the emission. DMS in
the atmosphere is oxidized by OH and $NO_3$ to form $SO_2$ (Table 2). Its natural emissions from
oceans from 1850 to 2010 in the model are the climatological monthly means (Dentener et al.,
2006) from MOZART2 data package. As shown in Fig 7a, the global amount of DMS in the
whole atmosphere was about 0.12 Tg during 1850-1900 and decreased to 0.055 Tg in 2010.
This decrease trend maybe partly results from the speeded rate of DMS oxidation with global
warming, and the loss of DMS gradually exceeds the source of ocean DMS emission to cause
a net loss of DMS in the atmosphere since 1910s. Largely driven by $SO_2$ anthropogenic
emissions, the sulfate burden shows three different stages from 1850 to present. In the first
period from 1850s to 1900s, the sulfate burden had a weak linear increase. It increased
significantly in the second stage from 1910's to 1940's, and then exploded since 1950's, until
the middle 1970s and early 1980s. The sulfate burden then remained nearly stable and even
showed slightly decreases as seen from the CMIP5 data. As for global BC and OC burdens,
BCC-ESM1 results show continuous increases since 1850s, especially from 1950 to present.
From 1910's to 1940's, the CMIP5 data show a slight decrease of BC and OC burdens in the
atmosphere.
The dust and sea salt aerosols in the atmosphere are largely determined by the
atmospheric circulations and states of the land and ocean surface. We can see that the global
dust burden in the atmosphere showed evident increase from 1980 to 2000, which could be
partly caused by evident global warming since 1980 and increasing soil dryness resulting in
more surface dust to be released in the atmosphere. Their details will be explored in the other
paper.
**4.2 Global aerosols budgets**
We further evaluate global aerosols budgets by comparing a 10-year average of
BCC-ESM results from 1990 to 2000 with various studies for sulfate, BC, OC, sea salt, and
dust. Their annual total emissions, average atmospheric mass loading, and mean lifetimes are
listed in Tables 5 and 6. It is worth emphasizing that the global mean total source and sink for
each type of aerosols in BCC-ESM1 are almost balanced.
The global DMS emission from the ocean is 27.4 $Tg(S)\ yr^{-1}$ in BCC-ESM. This
emission in BCC-ESM is nearly balanced by the gas-phase oxidation of DMS to form $SO_2$.
The DMS burden is 0.12 Tg with a lifetime of 0.78 days, which is within the range of other
models reported in the literature. As shown in Table 5, the total $SO_2$ production averaged for
the period of 1991 to 2000 is 76.93 $Tg(S)\ yr^{-1}$. A rate of 13.2 $Tg(S)\ yr^{-1}$ (about 17%) $SO_2$ is
produced from the DMS oxidation, only 0.1 $Tg(S)\ yr^{-1}$ $SO_2$ from airplane emissions to the
atmosphere, and the rest (63.63 $Tg(S)\ yr^{-1}$, near 82.7%) from anthropogenic activities and
volcanic eruption at surface. The amount of $SO_2$ produced from the DMS oxidation is in the
range of other works (10.0 to 24.7 $Tg(S)\ yr^{-1}$) reported in Liu et al (2005). All the $SO_2$
production is balanced by $SO_2$ losses by dry and wet deposition, and by gas- and
aqueous-phase oxidation. Half of its loss (38.74 $Tg(S)\ yr^{-1}$) occurs via its aqueous-phase
oxidation to form sulfate. Other losses through dry and wet depositions and gas-phase
oxidation to form $SO_4^{2-}$ are also important (Table 2). All the sinks are in the range from the
literature (Liu et al., 2005). The global burden of $SO_2$ in the atmosphere is 0.48 Tg with a
lifetime of 1.12 days, consistent with values in literature (Liu et al., 2005).
Sulfate aerosol is mainly produced from aqueous-phase $SO_2$ oxidation (38.73 $Tg(S)\ yr^{-1}$)
and partly from gaseous phase oxidation of $SO_2$ (10.32 $Tg(S)\ yr^{-1}$), and is largely lost by wet
scavenging (49.06 $Tg(S)\ y^{r-1}$). The total $SO_4^{2-}$ production in BCC-ESM is at the lower range
of values in other models reported in Textor et al. (2006). Its global burden is 1.89 Tg and the
lifetime is 4.69 days, which are within the range of 1.71 to 2.43 Tg and 3.3 to 5.4 days in the
literatures (Textor et al., 2006; Liu et al., 2012; Liu et al., 2016; Matsui and Mahowald, 2017;
Tegen et al., 2019; the value derived from CMIP5 data).
Sources of BC and OC are mainly from anthropogenic emissions. Based on the CMIP6
data, there are, on average, 7.22 Tg yr$^{-1}$ BC and 13.91 Tg yr$^{-1}$ OC from fossil and bio-fuel
emissions and 18.38 Tg yr$^{-1}$ OC from natural emission during the period of 1991 to 2000.
Most of them are scavenged through convective and large-scale rainfall processes. The rest
returns to the surface by dry deposition. The simulated global BC and OC burdens are 0.13
and 0.62 Tg, respectively (Table 6), all close to values of 0.114 Tg BC and 0.69 Tg OC
derived from the CMIP5 data, and within the range of 0.11-0.26 Tg BC (Textor et al., 2006;
Matsui and Mahowald, 2017; Tegen et al., 2019) and less than the values of 1.25-2.2 Tg OC
in other literatures (Textor et al., 2006; Tegen et al., 2019). The simulated BC and OC
lifetimes are 6.6 and 5.0 days respectively, and are close to the recent values of 5.0-7.5 days
BC and 5.4-6.6 days OC in literatures (Matsui and Mahowald, 2017; Tegen et al., 2019).
The emissions of dust and sea salt are mainly determined by winds near the surface. The
annual total dust emission in BCC-ESM1 is 2592 Tg yr$^{-1}$, higher than AeroCom multi-model
mean (1840 Tg yr$^{-1}$, Textor et al., 2006), but comparable to other studies (Chin et al., 2002;
Liu et al., 2012; Matsui and Mahowald, 2017). The average dust loading is 22.93 Tg, lower
than the value of 35.9 Tg in Ginoux et al. (2001) but slightly higher than the value of 20.41
Tg derived from CMIP5 data. The average lifetime for dust particles is 3.23 days that is
shorter than the AeroCom mean (4.14 days) and the value of 3.9 days in recent study (Matsui
and Mahowald, 2017). The simulated sea salt emission is 4667.2 Tg yr$^{-1}$, slightly lower than
the simulated value in Liu et al. (2012), and substantially lower than the AeroCom mean
(16600 Tg yr$^{-1}$, Textor et al., 2006). The simulated sea salt burdens are 11.89 Tg and close to
the CMIP5 data. Their averaged lifetimes are 0.93 days and close to the value in the recent of
Matsui and Mahowald (2017) but longer than the AeroCom mean (0.41days, Textor et al.,

2006).

**4.3 Global aerosol distributions at present day**
Figures 8-12 show December-January-February (DJF) and June-July-August (JJA) mean
column mass concentrations of sulfate (SO$_4^{2-}$), OC, BC, Dust, and Sea Salt aerosols averaged
for the period of 1991-2000, respectively. Here, BCC-ESM1 simulated results are compared
with the CMIP5-recommended data for the same period. Unlike the pre-industrial level of
sulfate shown in Fig. 2, sulfate concentrations at present day (Fig. 8) are strongly influenced
by anthropogenic emissions, and have maximum concentrations in the industrial regions (e.g.,
East Asia, Europe, and North America). Their seasonal variations are distinct and are
characterized by high concentrations in boreal summer and low concentrations in boreal
winter. These spatial distributions simulated by BCC-ESM1 are well consistent with the
CMIP5 data, with spatial correlation coefficients in DJF and JJA reaching 0.92 and 0.83
(Figure 13), respectively. The deviation of the spatial pattern in BCC-ESM1 is less from the
CMIP5 data in DJF but larger in JJA (Figure 13).
Unlike sulfate whose maximum concentrations are mainly distributed between 60 N
and the equator, peaking concentrations of BC and OC as shown in Figs. 9 and 10 are located
near the tropics in the biomass burning regions (e.g., the maritime continent, Central Africa,
South America), and their seasonal variations from DJF to JJA are evidently weaker than
those of sulfate except in South America. In boreal summer, there are centers of high values
in the industrial regions in the Northern Hemisphere mid-latitudes (i.e., East Asia, South Asia,
Europe, and North America). These main features of spatial and seasonal variations in CMIP5
data are well captured by BCC-ESM1, and the BCC-ESM1 vs. CMIP5 spatial correlation
coefficients (Figure 13) are 0.90 (OC in DJF), 0.91 (BC in DJF), 0.91 (OC in JJA) and 0.92
(BC in JJA). There are less deviations of spatial pattern for OC in DJF and JJA, but larger
deviation for BC from CMIP5 data (Figure 13).
As show in Figure 11, dust concentrations in the atmosphere show largest values over
strong source regions such as Northern Africa, Southwest and Central Asia, and Australia,
and over their outflow regions such as the Atlantic and the western Pacific. In DJF, the
CMIP5 data shows centers of high concentrations over East Asia and Central North America,
but both centers are missing in BCC-ESM1. However, these two high-value centers in the
CMIP5 data may not be true, since frozen soils in these areas in winter lead to unfavorable
conditions for soil erosion by winds. The spatial correlation coefficients between CMIP5 and
BCC-ESM1 remain high: 0.95 in JJA and 0.88 in DJF (Figure 13). Small deviations of spatial
pattern for dust simulations in BCC-ESM1 show less magnitude of dust maximums against
with CMIP5 data (Figure 13).
As shown in Figure 12, high sea salt concentrations are generally over the storm track
regions over the oceans, e.g., mid-latitudes in the Northern Oceans in DJF and the Southern
Ocean in JJA where wind speeds and thus sea salt emissions are higher. In addition, there is a
belt of high sea salt concentrations in the subtropics of both hemispheres where precipitation
scavenging is weak. Their spatial distributions in BCC-ESM1 are consistent with the CMIP5
data with correlation coefficients of 0.92 in JJA and 0.90 in DJF (Figure 13). The spatial
deviations of sea salt are much closer to CMIP5 data than those of sulfate, OC, BC, and dust
distributions (Figure 13).

Figure 14 shows vertical distributions of zonally-averaged annual mean concentrations

of sulfate, organic carbon, black carbon, dust, and sea salt aerosols in the period of 1991-2000.
Both BCC-ESM1 and CMIP5 results show that strong sulfur, OC, and BC emissions in the
industrial regions of the Northern Hemisphere mid-latitudes can rise upward and be
transported towards the North Pole in the mid- to upper troposphere. Most of OC, BC, and
dust aerosols are confined below 500 hPa, while sulfate can be transported to higher altitudes.
Sea salt aerosols are mostly confined below 700 hPa, as the particles are large in size and
favorable for wet removal and gravitational settling towards the surface. It can be seen that
BCC-ESM1 tends to simulate less upward transport of aerosols than the CMIP5 data, likely
reflecting the omission of deep convection transport of tracers in BCC-ESM1.

The CMIP5 data used here are mainly from model simulations. We will further evaluate

the BCC-ESM1 model results with ground observations. Annual mean $SO_4^{2-}$, BC and OC
aerosol observations from the Interagency Monitoring of Protected Visual Environments
(IMPROVE)       sites       over       1990-2005       in       the       United       States
(http://vista.cira.colostate.edu/IMPROVE/) and from the European Monitoring and Evaluation
Programme (EMEP) (http://www.emep.int) sites over 1995-2005 are used. As shown in
Figure 15a and 15b, the BCC-ESM simulated sulfate concentrations are in general
comparable to the EMEP observations in Europe, but are systematically by about 1 μg m$^{-3}$
higher than the U.S. IMPROVE observations. As for BC, there are large model biases at both
European and U.S. sites (Figs. 15c and 15d), especially BCC-ESM overestimates BC
concentrations at the IMPROVE sites. The observed OC concentrations are slightly
overestimated for IMPROVE sites but systematically underestimated for EMEP sites. Some
statistical features for simulated concentrations versus EMEP and IMPROVE observations are
listed in Table 7. These comparisons are overall fairly reasonable considering the
uncertainties in emissions and the coarse model resolution.

We then evaluate the simulated BC concentrations from BCC-ESM1 with the HIAPER

(High-Performance Instrumented Airborne Platform for Environmental Research)
Pole-to-Pole Observations (HIPPO) (Wofsy et al., 2011). The HIPPO campaign provided
observations of black carbon concentration profiles over Pacific Ocean and North America
between 2009 and 2011. Following Tilmes et al. (2016), model results here are sampled along
the HIPPO flight tracks and then averaged to different latitude and altitude bands for
comparison. As shown in Figure 16, BCC-ESM1 and HIPPO aircraft observations shows
reasonable agreement in terms of the spatial distributions and seasonal variations of BC levels.
BCC-ESM1 generally reproduces the observed hemispheric gradients of BC, i.e. the larger
burden in the NH compared to the SH, in consistent with Figures 10 and 14. The mean value
of modelled results along the flight track is 11.1 ng/kg, comparable to 8.2 ng/kg of the HIPPO
observations. The model shows large overestimations of BC observations over the tropics,
which is also found in the CAM4-chem global chemical model (Tilmes et al., 2016).
**4.4 Aerosol Optical Properties**

Aerosol optical depth (AOD) is an indicator of the reduction in incoming solar

radiation (at a particular wavelength) due to scattering and absorption of sunlight by aerosols.
In this study, we calculate the AOD at 550 nm for all aerosols including sulfate, BC, organic
carbon, sea salt and dust as the product of aerosol dry mass concentrations, aerosol water
content, and their specific extinction coefficients. The total AOD is calculated by summing
the AOD in each model layer for each aerosol species using the assumption that they are
externally mixed. The AOD observations retrieved from MODIS and MISR over the period of
1997-2003, and from AERONET over the period of 1998–2005 (http://aeronet.gsfc.nasa.gov)
are used to evaluate the averaged AOD at 550 nm in BCC-ESM. Figure 17 shows averages of
MISR and MODIS AOD with corresponding averages from BCC-ESM. The BCC-ESM1
simulated AOD generally captures the spatial distribution of MISR and MODIS retrievals.
The model overestimates AOD over East China. It also systematically underestimates the
MODIS observations in the Southern Hemisphere, but is closer to MISR observations. Figure

18 shows multi-years annual means of BCC-ESM1 simulated AOD values versus observations from AERONET over the period of 1998–2005. The basic pattern of modeled global AOD is similar to that of observations and their spatial correlation reaches 0.56. Large values of AOD are mainly distributed in land continents such as North African, South Asia, East Asia, Europe, and eastern part of North America. Figures 19a-19d present scatter plots of observed versus simulated multi-year monthly mean AOD at those sites of AERONET in Europe, North America, East Asia, and South Asia over the period of 1998-2005, respectively. Model simulated monthly AOD generally agrees with observations within a factor of 2 for most sites. BCC-ESM slightly overestimates the AOD in European and North American sites. In those regions, BCC-ESM also slightly overestimates MODIS and MISR AOD observations (Fig. 17).

## 5. Summary and discussions

This paper presents a primary evaluation of aerosols simulated in version 1 of the Beijing Climate Center Earth System Model (BCC-ESM1) with the implementation of the interactive atmospheric chemistry and aerosol based on the newly developed BCC-CSM2. Global aerosols (including sulfate, organic carbon, black carbon, dust and sea salt) and major greenhouse gases (e.g., $O_3$, $CH_4$, $N_2O$) in the atmosphere can be interactively simulated when anthropogenic emissions are provided to the model. Concentrations of all aerosols in BCC-ESM1 are determined by the processes of advective transport, emission, gas-phase chemical reactions, dry deposition, gravitational settling, and wet scavenging by clouds and precipitation. The nucleation and coagulation of aerosols are ignored in the present version of BCC-ESM1. Effects of aerosols on radiation, cloud, and precipitation are fully included.

We evaluate the performance of BCC-ESM1 in simulating aerosols and their optical properties in the 20th century following CMIP6 historical simulation according to the requirement of the AerChemMIP. It is forced with anthropogenic emissions evolving from 1850 to 2014 but some WMGHGs such as $CH_4$, $N_2O$, $CO_2$, CFC11 and CFC12 are prescribed using CMIP6 prescribed concentrations (to replace prognostic values of $CH_4$ and $N_2O$ from the chemistry scheme). Both direct and indirect effects of aerosols are considered in BCC-ESM1. Initial conditions of the CMIP6 historical simulation are obtained from a 600-year piControl simulation in the absence of anthropogenic emissions, which well captures

the pre-industrial concentrations of $SO_4^{2-}$, organic carbon (OC), black carbon (BC), dust, and
sea salt aerosols and are consistent with the CMIP5 recommended concentrations for the year
1850. With the CMIP6 anthropogenic emissions of $SO_2$, OC, and BC from 1850 to 2014 and
their natural emissions implemented in BCC-ESM1, the model simulated $SO_4^{2-}$, BC, and OC
aerosols in the atmosphere are highly correlated with the CMIP5-recommended data. The
long-term trends of CMIP5 aerosols from 1850 to 2000 are also well simulated by
BCC-ESM1. Global budgets of aerosols were evaluated through comparisons of BCC-ESM1
results for 1990-2000 with reports in various literatures for sulfate, BC, OC, sea salt, and dust.
Their annual total emissions, atmospheric mass loading, and mean lifetimes are all within the
range of values reported in relevant literature. Evaluations of the spatial and vertical
distributions of BCC-ESM1 simulated present-day $SO_4^{2-}$, OC, BC, Dust, and sea salt aerosol
concentrations against the CMIP5 datasets and in-situ measurements of surface networks
(IMPROVE in the U.S. and EMEP in Europe), and HIPPO aircraft observations indicate good
agreement among them. The BCC-ESM1 simulates weaker upward transport of aerosols from
the surface to the middle and upper troposphere (with reference to CMIP5-recommended
data), likely reflecting a lack of deep convection transport of chemical species in the present
version of BCC-ESM1. The AOD at 550 nm for all aerosols including sulfate, BC, OC, sea
salt, and dust aerosols was further compared with the satellite AOD observations retrieved
from MODIS and MISR and surface AOD observations from AERONET. The BCC-ESM1
model results are overall in good agreement with these observations within a factor of 2. All
these comparisons demonstrate the success of the implementation of interactive aerosol and
atmospheric chemistry in BCC-ESM1.
This work has only evaluated the ability of BCC-ESM1 to simulate aerosols. The
variations of aerosols especially for sulfate are related to other gaseous tracers such as OH
and $NO_3$ (Table 2), which are determined by the MOZART2 gaseous chemical scheme as
implemented in BCC-ESM1, and require further evaluation. As limited length of the text, the
other optical feature of aerosols such as extinction coefficients, single scattering albedo and
asymmetry parameters, and even their feedbacks on radiation and global temperature change
will be explored in the other paper. $O_3$ is evaluated in this work. Other GHGs such as $CH_4$ and
$N_2O$ concentrations can be simulated when forced with emissions and their simulations also
need to be evaluated in future.
**6. Code and data availability**
The source codes of BCC-ESM1, model input files, and scripts to reproduce the
simulations that are presented in the article have been archived and made publicly available
for downloading from https://zenodo.org/record/3609337 (Wu et al., 2020). Model output
data of BCC CMIP6 AerChemMIP simulations described in this paper are available on the
Earth System Grid Federation (ESGF)
(https://cera-www.dkrz.de/WDCC/ui/cerasearch/cmip6?input=CMIP6.AerChemMIP.BCC.B
CC-ESM1, https://doi.org/10.22033/ESGF/CMIP6.1733; Zhang et al., 2019). Details about
ESGF are presented on the CMIP Panel website at
http://www.wcrp-climate.org/index.php/wgcm-cmip/about-cmip.

**Author contributions**
Tongwen Wu led the BCC-ESM1 development. All other co-authors have contributions
to it. Fang Zhang and Jie Zhang designed the experiments and carried them out. Tongwen Wu,
Laurent Li, Lin Zhang, Xiaohong Liu, Aixue Hu, and Jun Wang wrote the final document
with contributions from all other authors.

**Acknowledgements**
This work was supported by The National Key Research and Development Program of China
(2016YFA0602100). All the figures are created by the NCAR Command Language (Version
6.6.2) [Software].

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

Table 1. Chemical species considered in BCC-AGCM3-Chem. Species marked with star (*) denote those added in BCC-ESM1 apart from the 63 species used in MOZART2. In the column of surface emission, interactive surface emissions are considered for sea salt and dust.

| Species | | Dry deposition | Wet deposition | Surface emission | Aircraft emission | Volcanic emission |
|---|---|:---:|:---:|:---:|:---:|:---:|
| | $O_3$ | √ | | | | |
| | $N_2O$ | | | √ | | |
| | N | | | | | |
| | NO | √ | | √ | √ | |
| | $NO_2$ | √ | | | | |
| | $NO_3$ | | | | | |
| | $HNO_3$ | √ | √ | | | |
| | $HO_2NO_2$ | √ | √ | | | |
| | $N_2O_5$ | | | | | |
| | $CH_4$ | √ | | √ | √ | |
| | $CH_3O_2$ | | | | | |
| | $CH_3OOH$ | √ | √ | | | |
| | $CH_2O$ | √ | √ | √ | | |
| | CO | √ | | √ | √ | |
| | OH | | | | | |
| | $HO_2$ | | | | | |
| | $H_2O_2$ | √ | √ | | | |
| | $C_3H_6$ | | | √ | | |
| | ISOP | | | √ | | |
| Gas tracers | $PO_2$ | | | | | |
| | $CH_3CHO$ | √ | √ | √ | | |
| | POOH | √ | √ | | | |
| | $CH_3CO_3$ | | | | | |
| | $CH_3COOOH$ | √ | √ | | | |
| | PAN | √ | | | | |
| | ONIT | √ | √ | | | |
| | $C_2H_6$ | | | √ | | |
| | $C_2H_4$ | | | √ | | |
| | $C_4H_{10}$ | | | √ | | |
| | MPAN | √ | | | | |
| | $ISOPO_2$ | | | | | |
| | MVK | | √ | | | |
| | MACR | | √ | | | |
| | $MACRO_2$ | | | | | |
| | MACROOH | √ | √ | | | |
| | $MCO_3$ | | | | | |
| | $C_2H_5O_2$ | | | | | |
| | $C_2H_5OOH$ | √ | √ | | | |
| | $C_{10}H_{16}$ | | | √ | | |

| Species name | | Dry deposition | Wet deposition | Surface emission | Aircraft emission | Volcanic emission |
|---|---|---|---|---|---|---|
| Gas tracers | $C_3H_8$ | | | √ | | |
| | $C_3H_7O_2$ | | | | | |
| | $C_3H_7OOH$ | √ | √ | | | |
| | $CH_3COCH_3$ | √ | | √ | | |
| | ROOH | | √ | | | |
| | $CH_3OH$ | √ | √ | √ | | |
| | $C_2H_5OH$ | √ | √ | √ | | |
| | GLYALD | √ | √ | | | |
| | HYAC | √ | √ | | | |
| | $EO_2$ | | | | | |
| | EO | | | | | |
| | HYDRALD | √ | √ | | | |
| | $RO_2$ | | | | | |
| | $CH_3COCHO$ | √ | √ | √ | | |
| | Rn-222 | | | | | |
| | Pb-210 | √ | √ | | | |
| | $ISOPNO_3$ | | √ | | | |
| | ONITR | √ | √ | | | |
| | $XO_2$ | | | | | |
| | XOOH | √ | √ | | | |
| | ISOPOOH | √ | √ | | | |
| | $H_2$ | √ | | √ | | |
| | Stratospheric $O_3$ | √ | | | | |
| | Inert $O_3$ | √ | | | | |
| | $SO_2^*$ | √ | √ | √ | √ | √ |
| | $DMS^*$ | | | √ | | |
| | $NH_3^*$ | | | √ | √ | |
| Aerosols | $SO_4^{2-*}$ | √ | √ | | | |
| | $OC1^*$ | √ | √ | √ | √ | |
| | $OC2^*$ | √ | √ | √ | √ | |
| | $BC1^*$ | √ | √ | √ | √ | |
| | $BC2^*$ | √ | √ | √ | √ | |
| | $SSLT01^*$ | √ | √ | | | |
| | $SSLT02^*$ | √ | √ | | | |
| | $SSLT03^*$ | √ | √ | | | |
| | $SSLT04^*$ | √ | √ | | | |
| | $DST01^*$ | √ | √ | | | |
| | $DST02^*$ | √ | √ | | | |
| | $DST03^*$ | √ | √ | | | |
| | $DST04^*$ | √ | √ | | | |


Table 2. Gas-phase chemical reactions for $NH_3$ and bulk aerosols precursors following
CAM-Chem (Lamarque et al., 2012). The reaction rates ($s^{-1}$) refer to Tie et al. (2001) and
Sander et al. (2003), and Cooke and Wilson (1996). Temperature (T) is expressed in K, air
density (M) in molecule $cm^{-3}$, ki and ko in $cm^3$ $molecule^{-1}$ $s^{-1}$.

| Chemical reactions | Rate |
|---|---|
| $NH_3 + OH \rightarrow H_2O$ | 1.70E-12*exp(–710/T) |
| $SO_2 + OH \rightarrow SO_4{}^{2-}$ | ko/(1.+ko*M/ki)*f**(1./(1.+log10(ko*M/ki)), in which |
|  | ko=3.0E-31*(300/T)**3.3; ki=1.E-12; f =0.6 |
| $DMS + OH \rightarrow SO_2$ | 9.60E-12*exp(-234./T) |
| $DMS + OH \rightarrow .5*SO_2 + .5*HO_2$ | 1.7E−42*exp(7810/T)*M*0.21/(1+5.5E−31*exp(7460/T)* M* 0.21) |
| $DMS + NO_3 \rightarrow SO_2 + HNO_3$ | 1.90E-13*exp( 520/T) |
| $BC1 \rightarrow BC2$ | 7.10E-06 |
| $OC1 \rightarrow OC2$ | 7.10E-06 |




Table 3. Size and density parameters of bulk aerosols.

| Aerosols | Species Name | Mean radius (μm) / bin size (μm) | Geometric standard deviation (μm) | Density (g cm$^{-3}$) |
|---|---|---|---|---|
| SO$_4^{2-}$ | Sulfate | 0.05 | 2.03 | 1.77 |
| BC1 | hydrophobic black carbon | 0.02 | 2.00 | 1.0 |
| BC2 | hydrophilic black carbon | 0.02 | 2.00 | 1.0 |
| OC1 | hydrophobic organic carbon | 0.03 | 2.24 | 1.8 |
| OC2 | hydrophilic organic carbon | 0.03 | 2.24 | 1.8 |
| DST01 | Dust | 0.55/ bin: 0.1-1.0 | 2.00 | 2.5 |
| DST02 | Dust | 1.75 / bin: 1.0-2.5 | 2.00 | 2.5 |
| DST03 | Dust | 3.75 / bin: 2.5-5.0 | 2.00 | 2.5 |
| DST04 | Dust | 7.50 / bin: 5.0-10. | 2.00 | 2.5 |
| SSLT01 | Sea salt | 0.52 / bin: 0.2-1.0 | 2.00 | 2.2 |
| SSLT02 | Sea salt | 2.38 / bin: 1.0-3.0 | 2.00 | 2.2 |
| SSLT03 | Sea salt | 4.86 / bin: 3.0-10. | 2.00 | 2.2 |
| SSLT04 | Sea salt | 15.14 / bin: 10.-20. | 2.00 | 2.2 |


Table 4. Source of emission data. MOZART2 data denote the standard tropospheric chemistry package for MOZART contains surface emissions from the EDGAR 2.0 data base (Olivier et al., 1996). ACCMIP data are downloaded from the IPCC ACCMIP emission inventory (http://accent.aero.jussieu.fr/ACCMIP.php) and they vary from 1850 to 2000, in 10-year steps (Lamarque et al., 2010). CMIP6 data are from https://esgf-node.llnl.gov/search/input4mips/. Anthropogenic emission includes Industrial and fossil fuel use, agriculture, ships, and etc. Biomass burning includes vegetation fires incl. fuel wood and agricultural burning.

| Species | Anthropogenic emission | Biomass burning | Biogenic emissions from vegetation | Biogenic emissions from soil | Oceanic emissions | Airplane emission | Volcanic emission |
|---|---|---|---|---|---|---|---|
| $C_2H_4$ | CMIP6 | CMIP6 | On-line computation | | MOZART2 | | |
| $C_2H_5OH$ | CMIP6 | CMIP6 | | | | | |
| $C_2H_6$ | CMIP6 | CMIP6 | ACCMIP | | MOZART2 | | |
| $C_3H_6$ | CMIP6 | CMIP6 | On-line computation | | MOZART2 | | |
| $C_3H_8$ | CMIP6 | CMIP6 | ACCMIP | | MOZART2 | | |
| $C_4H_{10}$ | CMIP6 | CMIP5 | MOZART2 | | MOZART2 | | |
| $CH_2O$ | CMIP6 | CMIP6 | | | | | |
| $CH_3CHO$ | ACCMIP | CMIP6 | | | | | |
| $CH_3COCHO$ | | CMIP6 | | | | | |
| $CH_3OH$ | ACCMIP | CMIP6 | ACCMIP | | | | |
| $CH_3COCH_3$ | ACCMIP | ACCMIP | On-line computation | | MOZART2 | | |
| ISOP | | CMIP5 | On-line computation | | | | |
| $C_{10}H_{16}$ | | CMIP6 | On-line computation | | | | |
| $CH_4$ | CMIP6 | CMIP6 | MOZART2 | | MOZART2 | CMIP6 | |
| CO | CMIP6 | CMIP6 | ACCMIP | MOZART2 | ACCMIP | CMIP6 | |
| $H_2$ | MOZART2 | CMIP6 | | MOZART2 | MOZART2 | | |
| $N_2O$ | MOZART2 | CMIP6 | | MOZART2 | MOZART2 | | |
| $NH_3$ | CMIP6 | CMIP6 | | ACCMIP | ACCMIP | CMIP6 | |
| NO | CMIP6 | CMIP6 | | ACCMIP | | CMIP6 | |
| $SO_2$ | CMIP6 | CMIP6 | | | | CMIP6 | ACCMIP |
| DMS | | | | | ACCMIP | | |
| OC1 | CMIP6 | CMIP6 | | | | CMIP6 | |
| OC2 | CMIP6 | CMIP6 | On-line computation | | | CMIP6 | |
| BC1 | CMIP6 | CMIP6 | | | | CMIP6 | |
| BC2 | CMIP6 | CMIP6 | | | | CMIP6 | |

Table 5. Global budgets for DMS, $SO_2$, and sulfate in the period of 1991 to 2000. Units are sources and sinks, Tg(S) yr$^{-1}$; burden, Tg; lifetime, days.

| | | BCC-ESM (1991-2000 mean) | Other studies and CMIP5 data |
|---|---|---|---|
| DMS | Sources | 27.4 | |
| | Emission | 27.4 | 10.7-23.7[a] |
| | Sinks | 28.0 | |
| | Gas-phase oxidation | 28.0 | |
| | Burden | 0.12 | 0.04-0.29[a] |
| | Lifetime | 0.78 | 0.5-3.0[a] |
| $SO_2$ | Sources | 76.93 | |
| | Emission at surface | 63.63 | |
| | Emission from airplane | 0.10 | |
| | DMS oxidation | 13.20 | 10.0-24.7[a] |
| | Sinks | 76.96 | |
| | Dry deposition | 18.53 | 16.0-55.0[a] |
| | Wet deposition | 9.36 | 0.0-19.9[a] |
| | Gas-phase oxidation | 10.33 | 6.1-16.8[a] |
| | Aqueous-phase oxidation | 38.74 | 24.5-57.8[a] |
| | Burden | 0.48 | 0.40-1.22[a] |
| | Lifetime | 1.12 | 0.6-2.6[a] |
| $SO_4^{2-}$ | Sources | 49.05 | 59.67±13.13[b] |
| | Emission | 0.00 | |
| | $SO_2$ aqueous-phase oxidation | 38.73 | |
| | $SO_2$ gas-phase oxidation | 10.32 | |
| | Sinks | 49.06 | |
| | Dry deposition | 2.20 | 4.96-5.51[d] |
| | Wet deposition | 46.86 | 39.34-40.20[d] |
| | Burden | 1.89 | 1.98±0.48[b], 1.71[c], 1.2[g], 2.22-2.43[h] |
| | Lifetime | 4.69 | 4.12±0.74[b], 3.72-3.77[d] 3.3[g], 3.7-4.0[h] |

Notes: References denote a for Liu et al. (2005), b for Textor et al. (2006), c for the values derived from CMIP5 prescribed aerosol masses averaged from 1991 to 2000, d for Liu et al. (2012), g for Matsui and Mahowald (2017), and h for Tegen et al. (2019). Values of DMS, $SO_2$, and sulfate burdens in the literature d are transferred from TgS to Tg (species) for units consistence.

Table 6. Same as Table 5, but for global budgets for black carbon, organic carbon, dust, and sea salts. Units are sources and sinks, Tg yr$^{-1}$; burden, Tg; lifetime, days.

| | | BCC-ESM (1991-2000 mean) | Other studies and CMIP5 data |
|---|---|---|---|
| BC | Sources | 7.22 | |
| | Emission | 7.22 | $11.9\pm2.7$[b], 7.8[g] |
| | Sinks | 7.24 | 7.75[d], 7.8[g] |
| | Dry deposition | 0.90 | 0.27[g], 1.30-1.64[e] |
| | Wet deposition | 6.34 | 7.5[g], 6.10-6.45[e] |
| | Burden | 0.13 | 0.114[c], $0.24\pm0.1$[b], 0.11[g], 0.14-0.26[h], 0.084-0.123[e] |
| | Lifetime | 6.60 | $7.12\pm2.35$[b], 3.95-4.80[e], 5.0[g], 6.3-7.5[h] |
| OC | Sources | 32.29 | |
| | Fossil and biofuel emission | 13.91 | |
| | Natural emission | 18.38 | |
| | Sinks | 32.30 | |
| | Dry deposition | 2.44 | |
| | Wet deposition | 29.86 | |
| | Burden | 0.62 | 0.69[c], $1.7\pm0.45$[b], 1.0-2.2[h] |
| | Lifetime | 5.00 | $6.54\pm1.76$[b], 4.56−4.90[d], 6.4[g], 5.4-6.6[h] |
| Dust | Sources | 2592.0 | 1840[b], 2943.5−3121.9[d], 2677[g] |
| | Sinks | 2592.0 | |
| | Dry deposition | 1630.8 | 1444[g] |
| | Wet deposition | 961.2 | 1245[g] |
| | Burden | 22.93 | 20.41[c], 22.424.7[d], 35.9[f], $19.2\pm7.68$[b], 28.5[g], 16.5-17.9[h] |
| | Lifetime | 3.23 | $4.14\pm1.78$[b], 2.61−3.07[d], 3.9[g], 5.3-5.7[h] |
| Sea Salt | Sources | 4667.2 | 4965.5-5004.1[d], 5039[g] |
| | Sinks | 4667.4 | |
| | Dry deposition | 2978.5 | 2158[g] |
| | Wet deposition | 1688.9 | 2918[g] |
| | Burden | 11.89 | 7.58−10.37[a], $6.4\pm3.4$[b], 11.84[c], 13.6[g], 3.9[h] |
| | Lifetime | 0.93 | $0.41\pm0.24$[b], 0.55−0.76[d], 0.98[g], 1.2-1.3[h] |

Notes: References denote a for Liu et al. (2005), b for Textor et al. (2006), c derived from CMIP5 prescribed aerosol masses averaged from 1991 to 2000, d for Liu et al. (2012), e for Liu et al. (2016), f for Ginoux (2001), g for Matsui and Mahowald (2017), and h for Tegen et al. (2019).

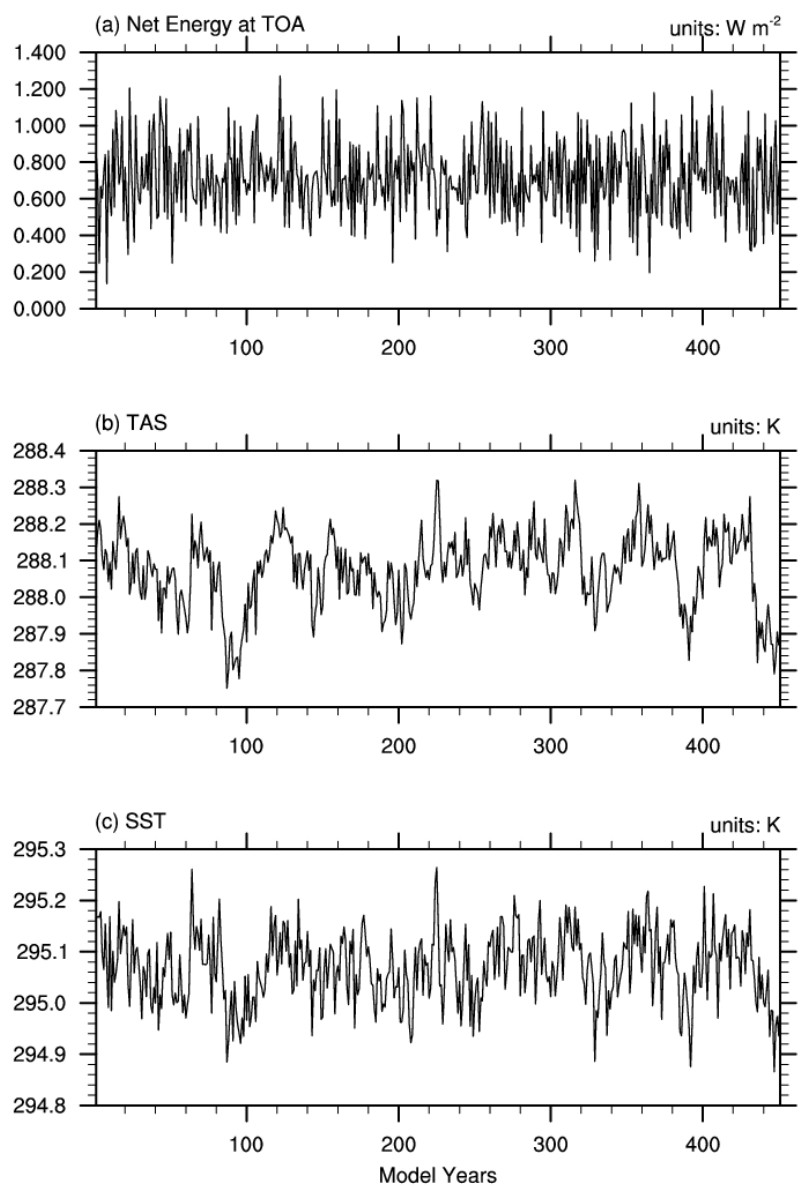

Figure 1. The time series of global and annual mean of (a) net energy budget at top of atmosphere (W m$^{-2}$), (b) near-surface air temperature (K), and (c) sea surface temperature (K) in the last 450 years of the piControl simulation.

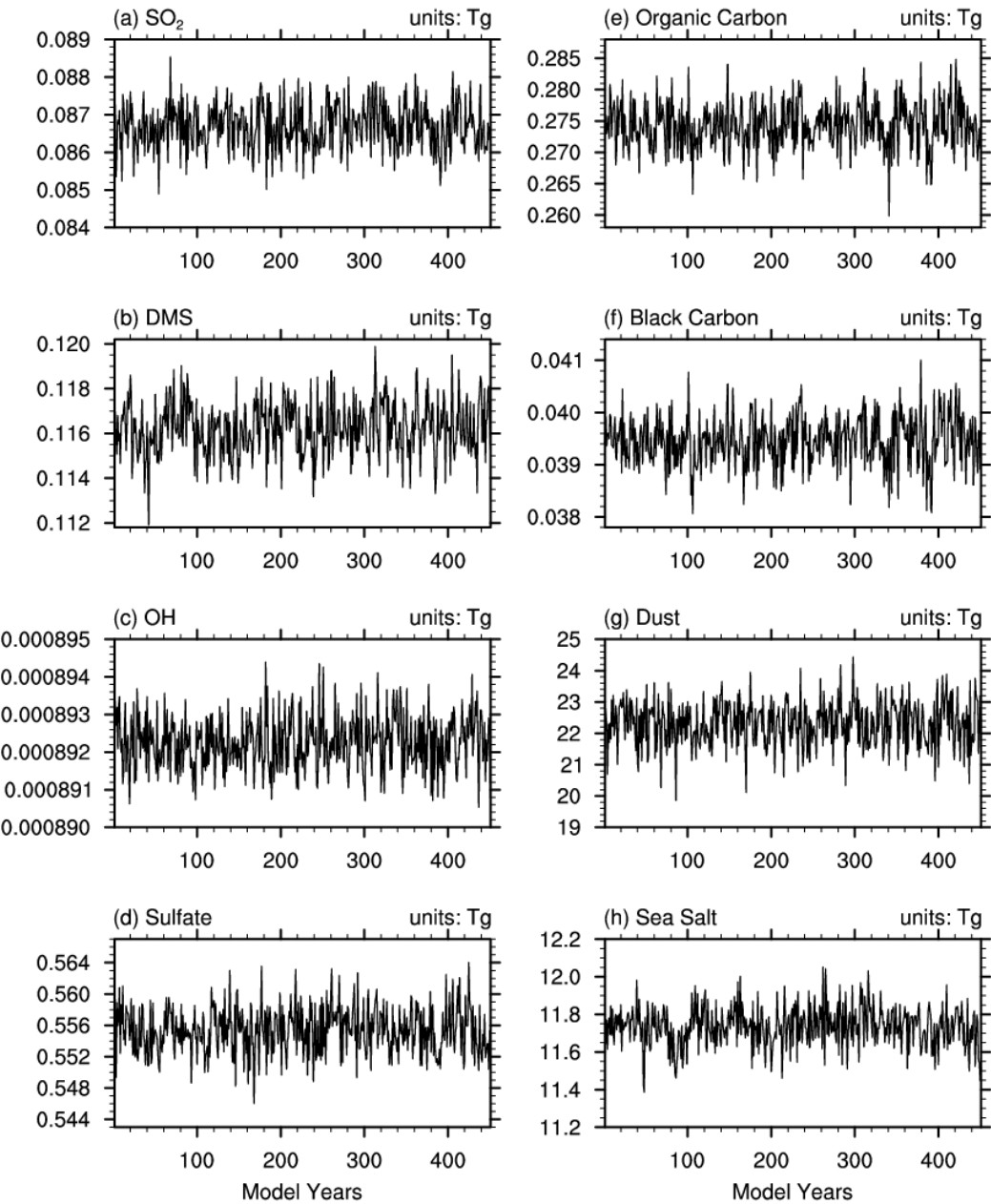

Figure 2. Same as in Figure 1, but for the global burdens of (a) SO₂, (b) DMS, (c) OH, and (d-h) different aerosols in the troposphere (below 100 hPa). Units are Tg.

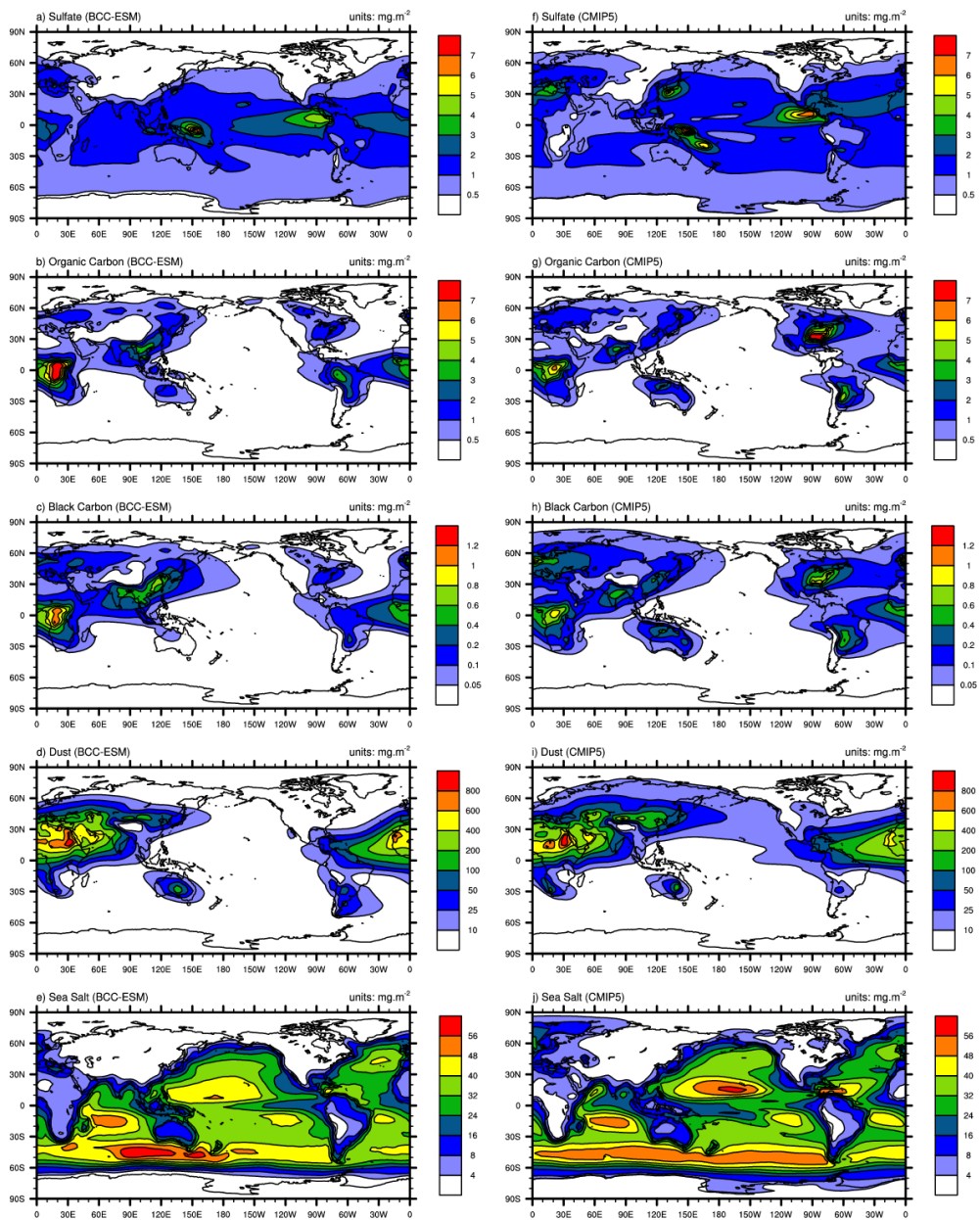

Figure 3. Global distributions of annual mean mass burdens of sulfate ($SO_4^{2-}$; first row), organic carbon (OC; second row), black carbon (BC; third row), dust (fourth row), and sea salt (fifth row) aerosols in the whole atmospheric column. The left panels show the mean averaged for the last 100 years of BCC-ESM pre-industrial piControl simulations, and the right panels show the CMIP5 recommended aerosol concentrations in year 1850 (the website at IIASA http://tntcat.iiasa.ac.at/RcpDb/.). Units: mg m$^{-2}$.

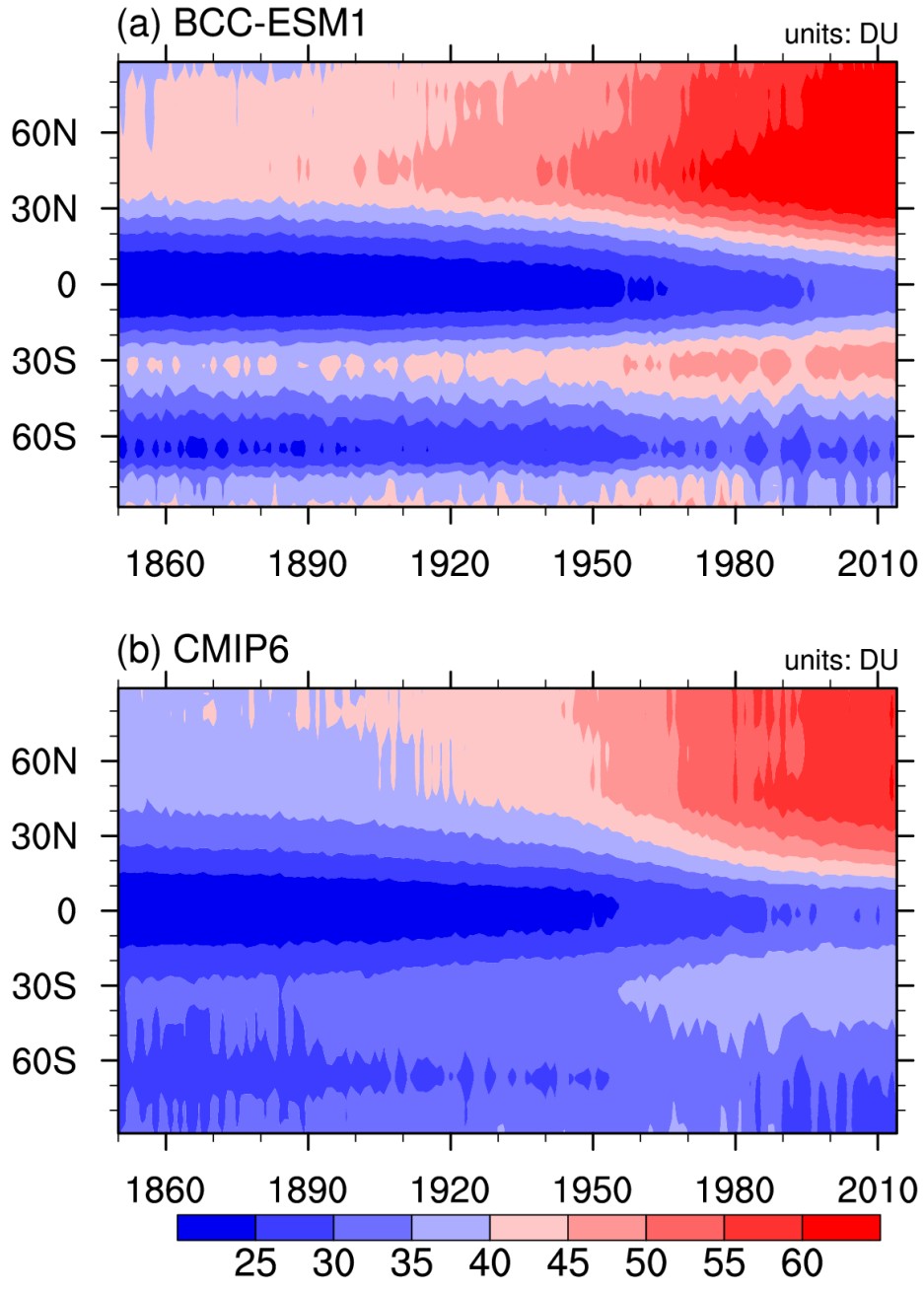

Figure 4. Zonal mean of yearly mean concentration of ozone column in the troposphere below 300 hPa to the ground from 1871 to 1999 for (a) BCC-ESM1 and (b) CMIP6 data. Unit: DU.

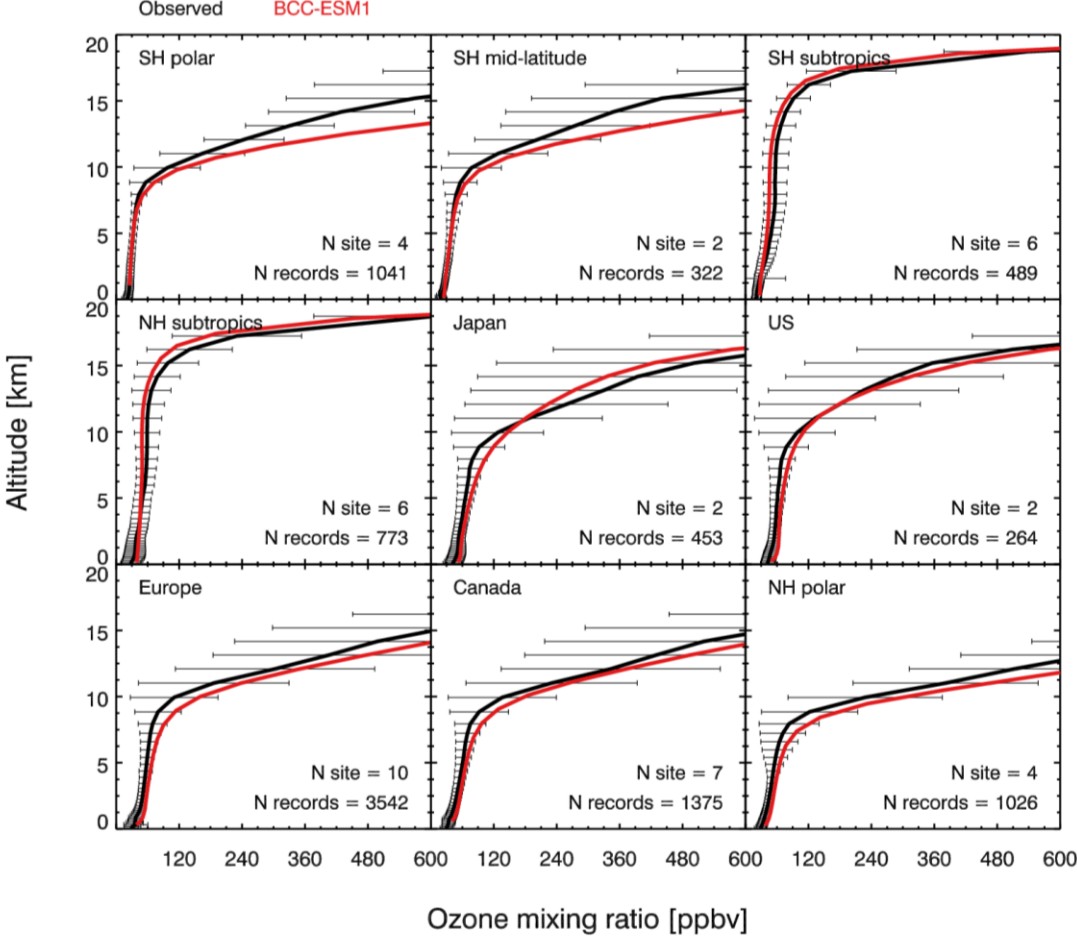

Figure 5. Vertical profiles of annual mean ozone concentrations from observations averaged for 2010-2014 in nine regions (black) and from the BCC-ESM1 simulations (red). The observations are derived from 41 global WOUDC sites.

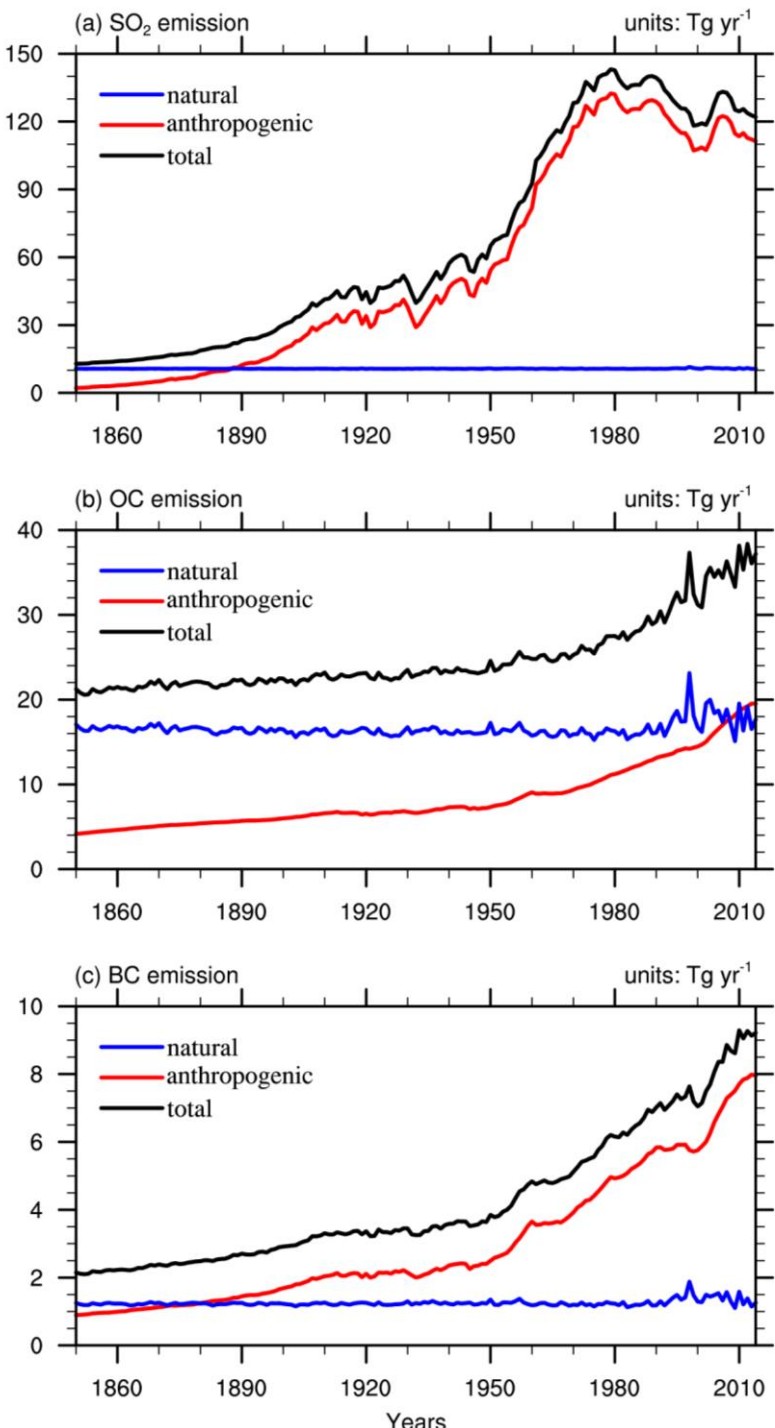

Figure 6. Global annual anthropogenic, natural, and total emissions of SO₂, organic carbon (OC), and black carbon (BC) in the BCC-ESM1 historical simulation. All the biomass burning emissions are included in natural emissions in (a)-(c). Units: Tg yr⁻¹.

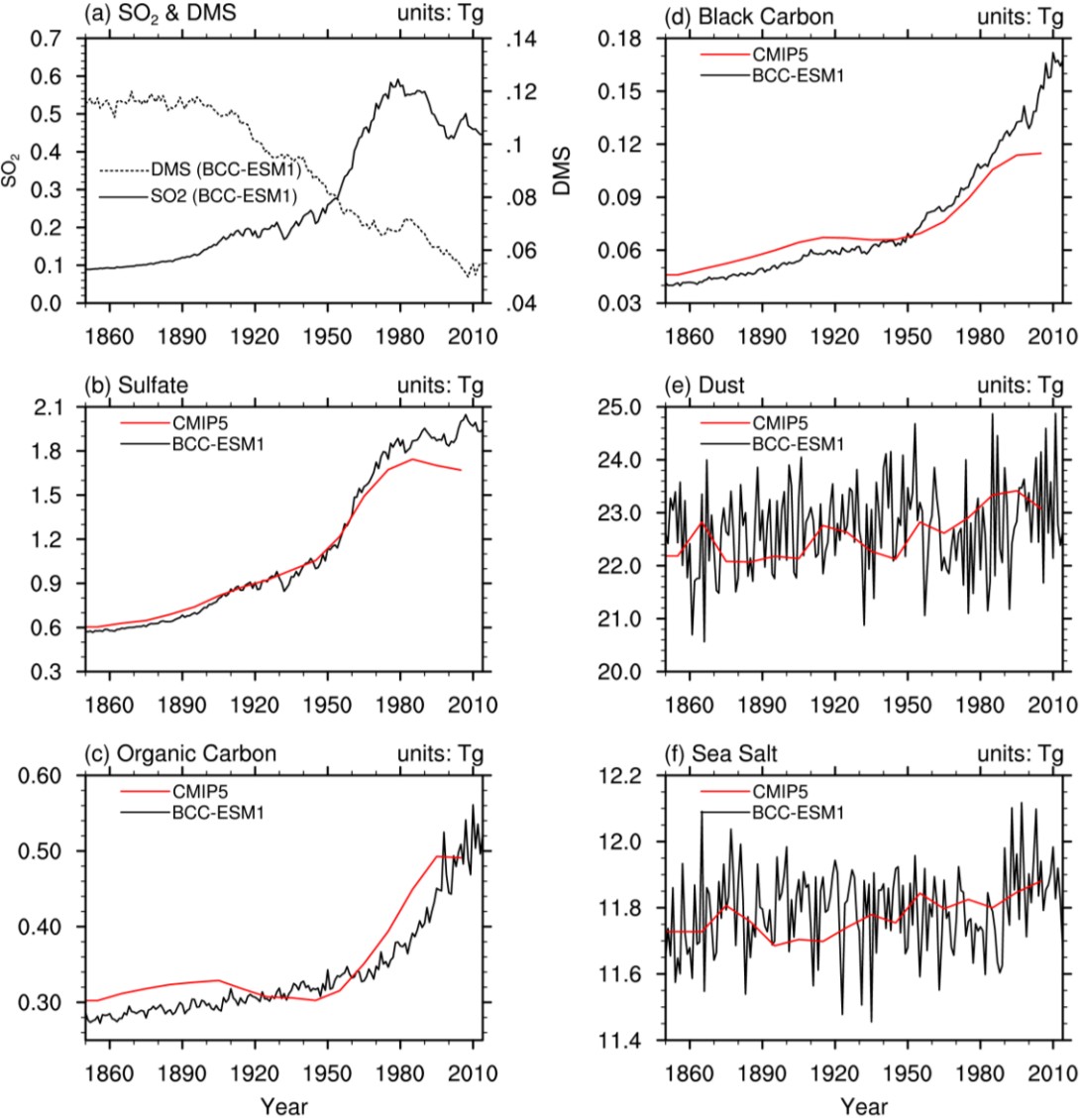

Figure 7. The time series of global yearly amounts of (a) SO$_2$ and DMS and (b-f) aerosols in the whole atmosphere column from the CMIP6   historical simulations of BCC-ESM1 (black lines) and the CMIP5-recommended aerosols masses (red lines). The yearly CMIP5 data are interpolated from the time series in 10-year interval. Units: Tg.

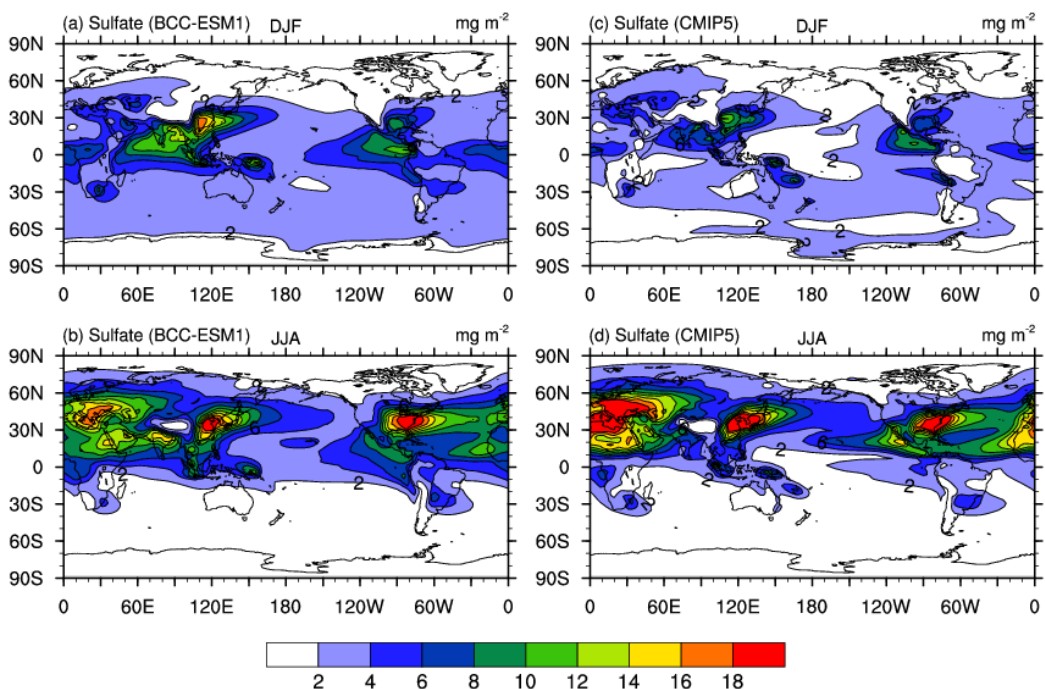

Figure 8. December-January-February (DJF; top panels) and June-July-August (JJA; bottom panels) mean sulfate ($SO_4^{2-}$) aerosol column mass concentrations averaged for the period of 1971-2000. Left panels show the historical simulations of BCC-ESM1, and right panels the CMIP5-recommended data. Units: mg.m$^{-2}$.

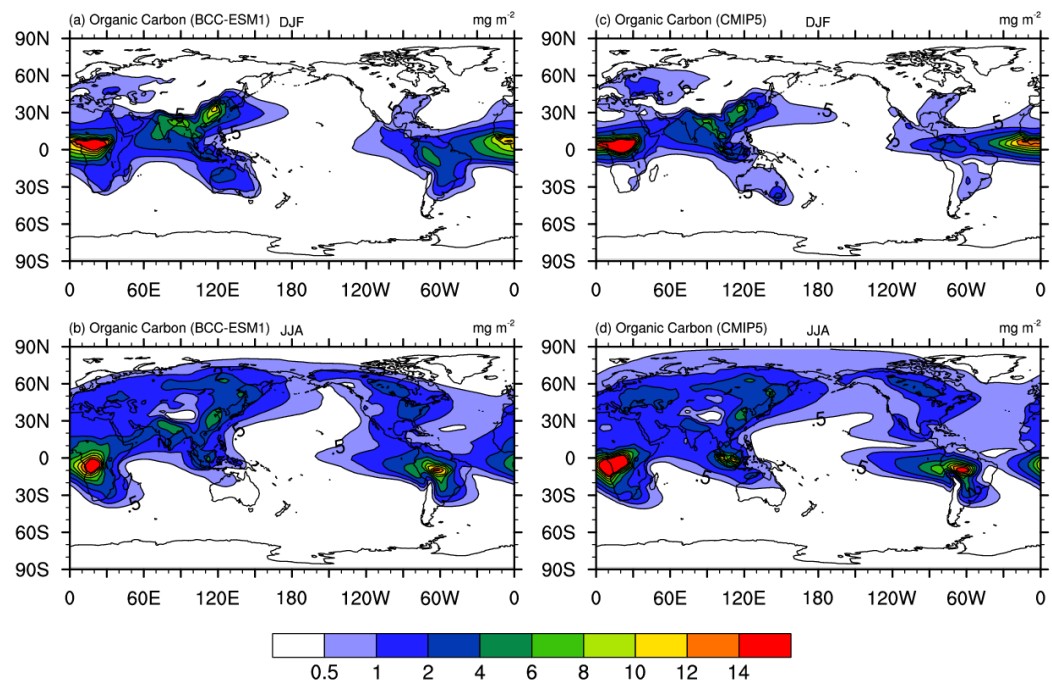

Figure 9. The same as in Figure 8, but for organic carbon (OC) aerosol column mass concentrations. Units: mg m$^{-2}$.

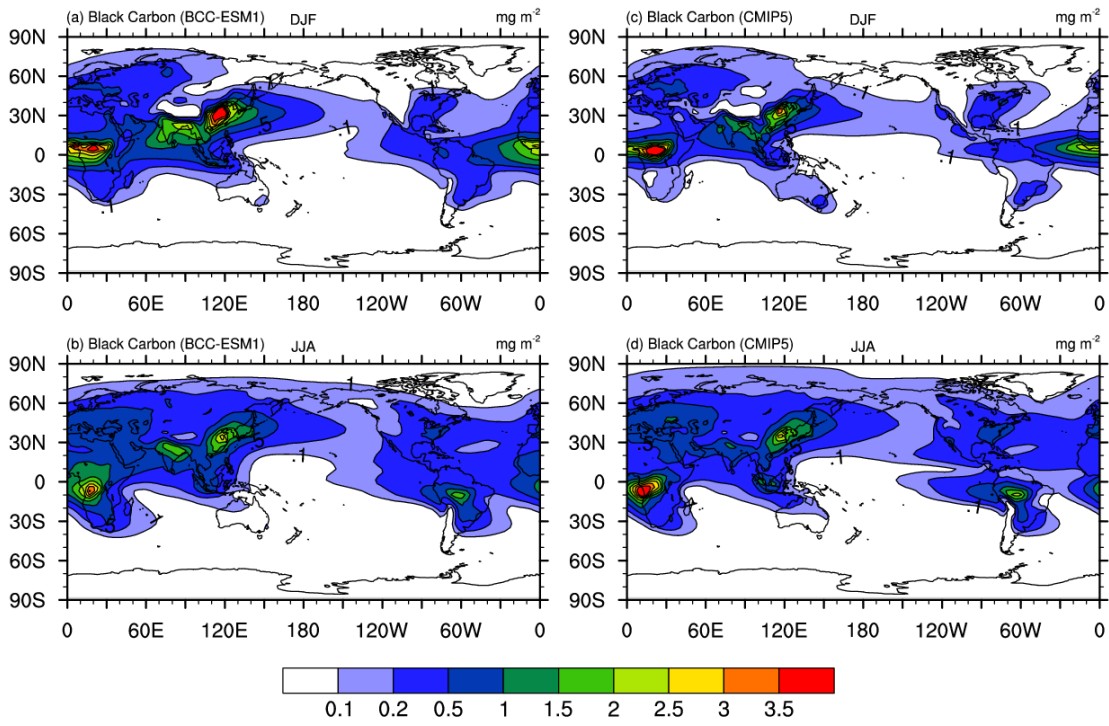

Figure 10. The same as in Figure 8, but for black carbon (BC) aerosol. Units: mg.m$^{-2}$.

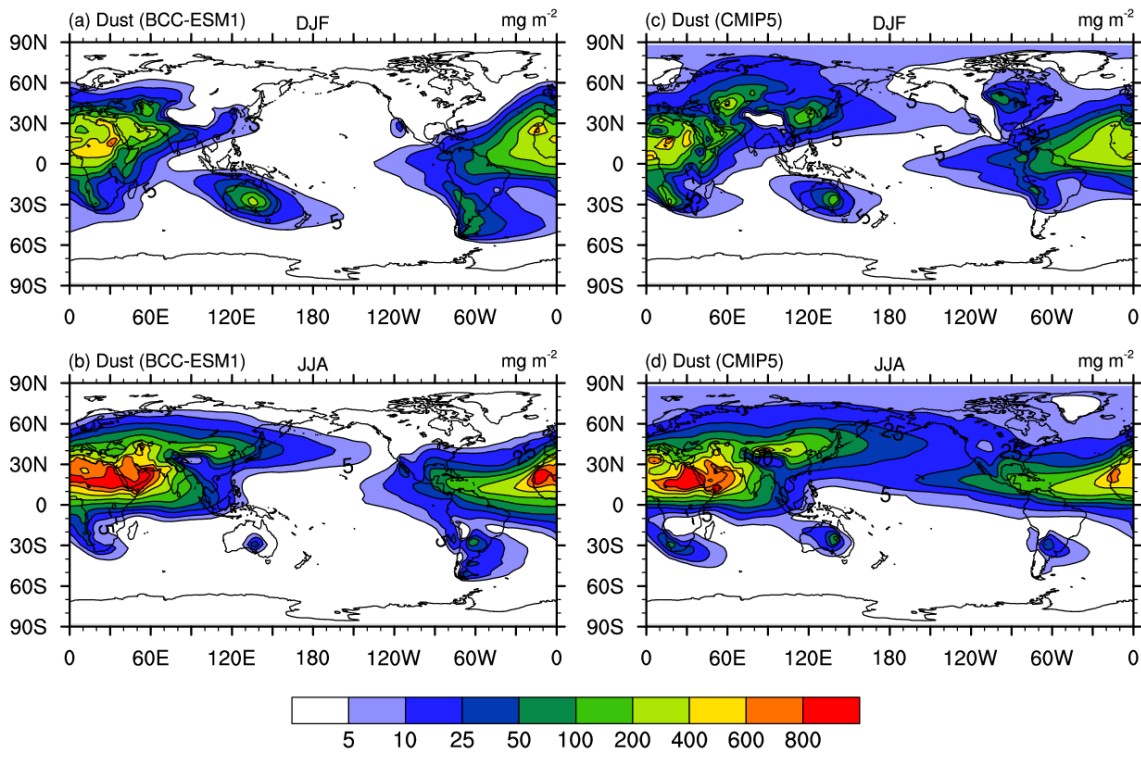

Figure 11. The same as in Figure 8, but for dust aerosol. Units: mg.m$^{-2}$.

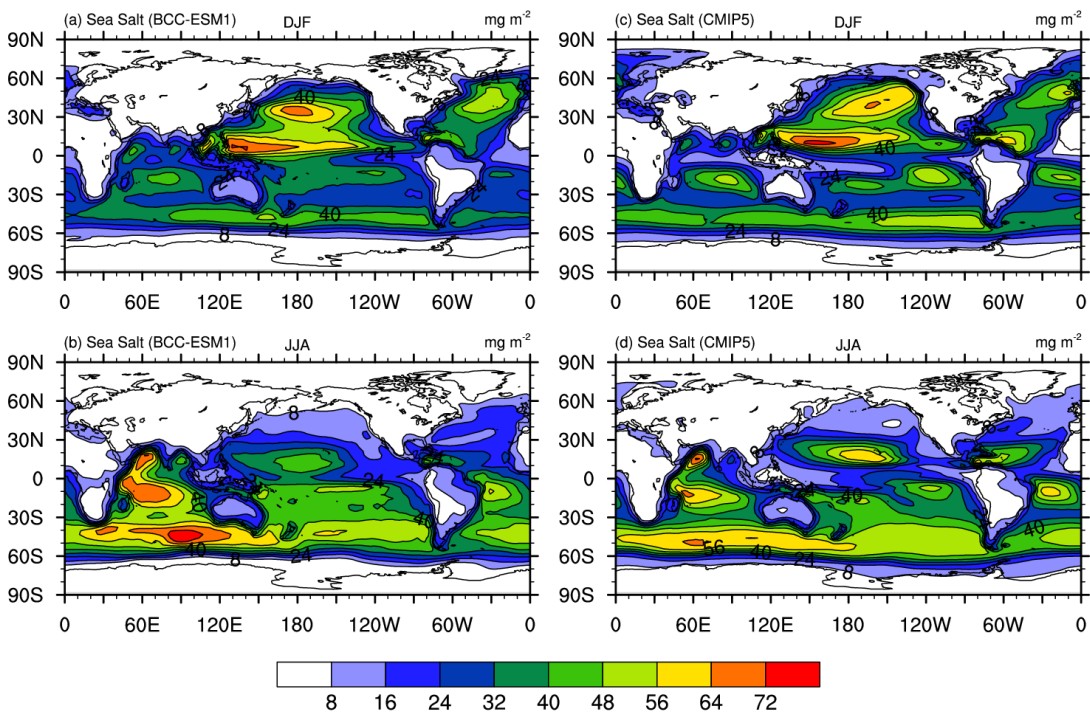

Figure 12. The same as in Figure 8, but for sea salt (SSLT) aerosol. Units: mg.m$^{-2}$.

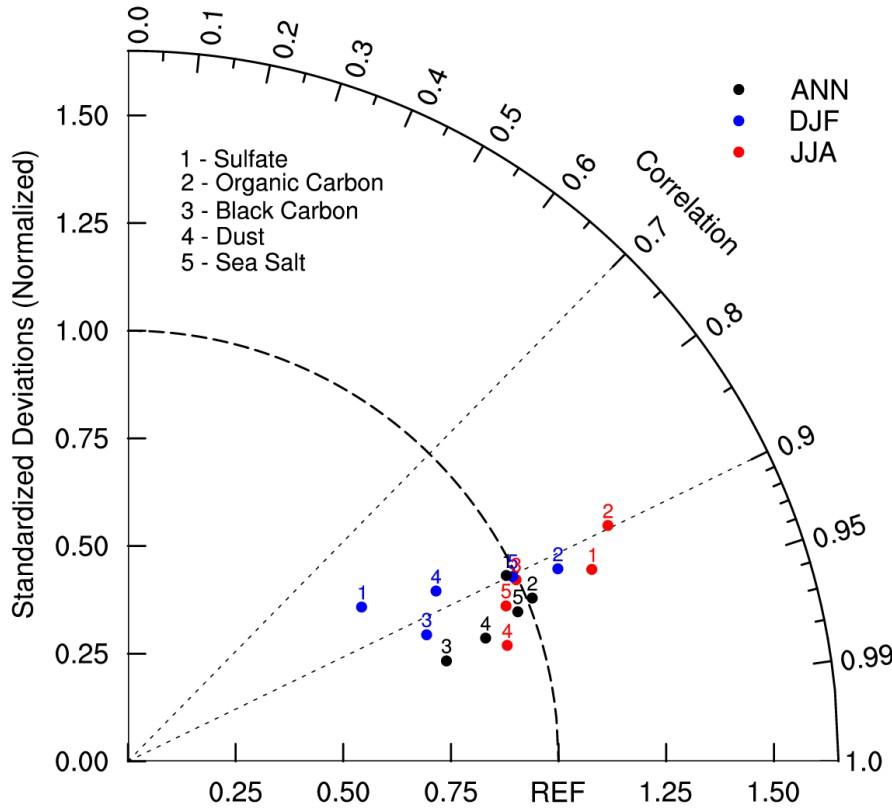

Figure 13. Taylor diagram for the global aerosols climatology (1971–2000) of sulfate, organic carbon, black carbon, dust, and sea salt averaged for December-January-February (DJF), June-July-August (JJA), and annual respectively. The radial coordinate shows the standard deviation of the spatial pattern, normalized by the observed standard deviation. The azimuthal variable shows the correlation of the modelled spatial pattern with the observed spatial pattern. Analysis is for the whole globe. The reference dataset is CMIP5-prescribed dataset.

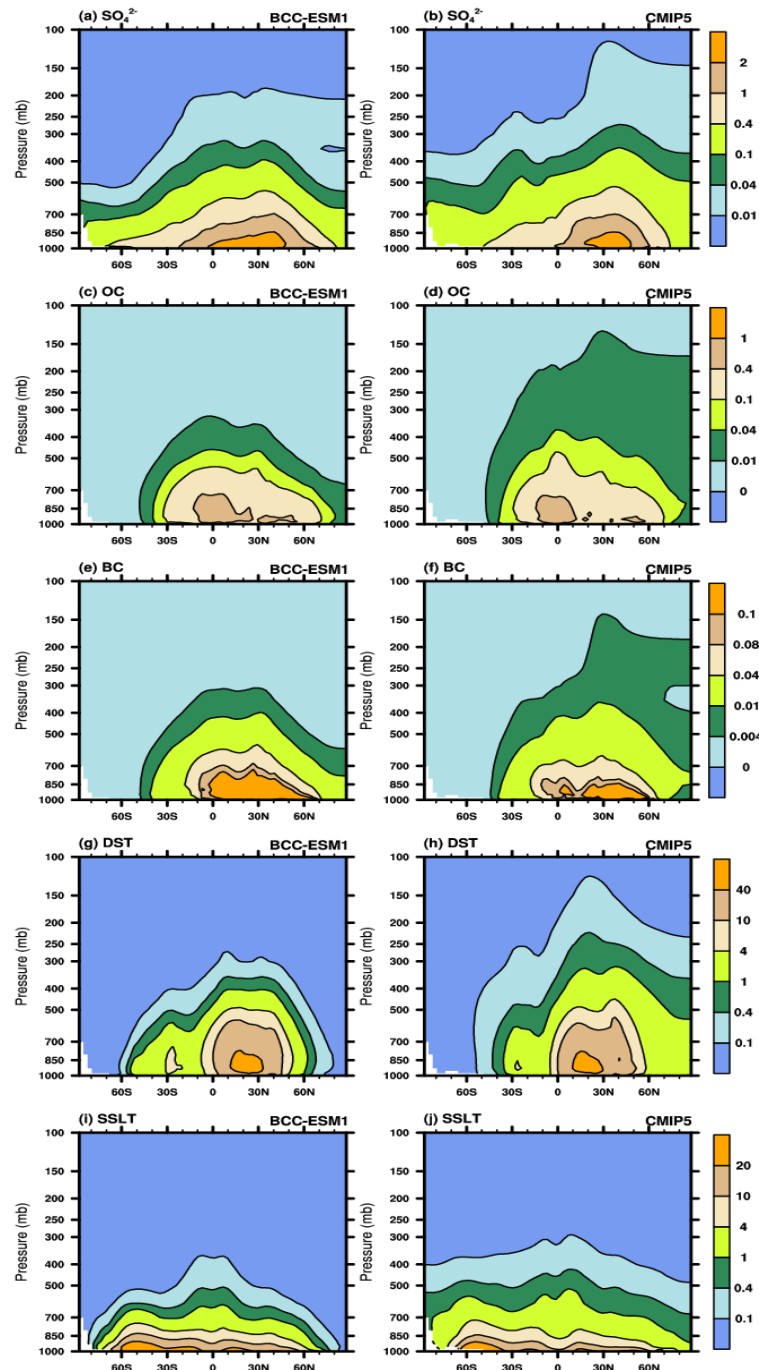

Figure 14. Latitude-pressure distributions of zonally-averaged annual mean sulfate, organic carbon, black carbon, dust, and sea salt aerosol concentrations for the period of 1971-2000. Left panels show the    CMIP6 historical simulation of BCC-ESM1, and right panels the CMIP5 recommendation data. Units: μg m$^{-3}$.

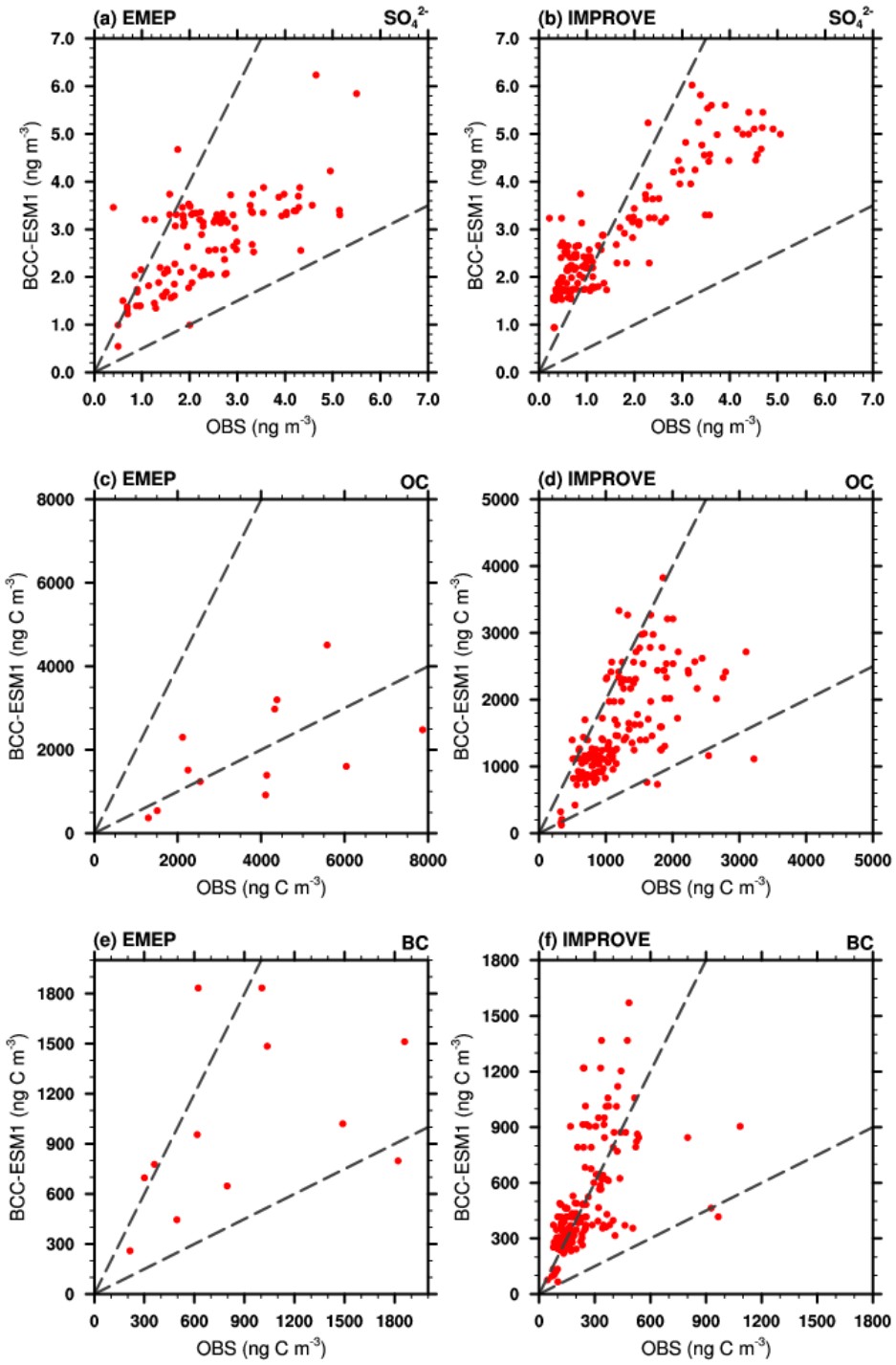

Figure 15. Scatter plots showing observed versus simulated multi-years averaged annual mean sulfate ($SO_4^{2-}$), organic carbon (OC), black carbon (BC) mixing ratios at IMPROVE and EMEP network sites. Observations are averages over the available years 1990–2005 for IMPROVE sites, and 1995–2005 for EMEP sites. Simulated values are those at the lowest layer of BCC-ESM1.

Table 7. Observed versus simulated concentrations of sulfate ($SO_4^{2-}$), organic carbon (OC), black carbon (BC) for the regional mean and spatial standard deviation, minimum and maximum values at HIPPO aircraft observations (BC only), IMPROVE and EMEP network sites, and the spatial correlation between observed and simulated multi-years averaged annual means. Simulated values are selected for the same locations and same valid observation time. The data used same as those in Figure 12.

| | EMEP | | | IMPROVE | | | HIPPO |
|---|---|---|---|---|---|---|---|
| | $SO4^{2-}$ (Obs/Model) | OC (OBS/Model) | BC (OBS/Model) | $SO4^{2-}$ (OBS/Model) | OC (OBS/Model) | BC (OBS/Model) | BC (OBS/Model) |
| **Mean Values** | 2.37/2.74 | 3844/1919 | 884/1022 | 1.53/2.79 | 1215/1565 | 249/504 | 8.2/11.1 |
| **Std Deviation** | 1.16/0.93 | 1997/1215 | 572/526 | 1.30/1.20 | 572/745 | 164/296 | 27.9/21.0 |
| **Min Values** | 0.40/0.55 | 1296/369 | 214/259 | 0.22/0.94 | 322/123 | 45/66 | 0.0025/0.066 |
| **Max Values** | 5.50/6.24 | 7867/4510 | 1859/1834 | 5.07/6.02 | 3219/3827 | 1084/1570 | 558.91/267.11 |
| **Correlation (Obs and Model)** | 0.67 | 0.56 | 0.40 | 0.90 | 0.63 | 0.55 | 0.51 |

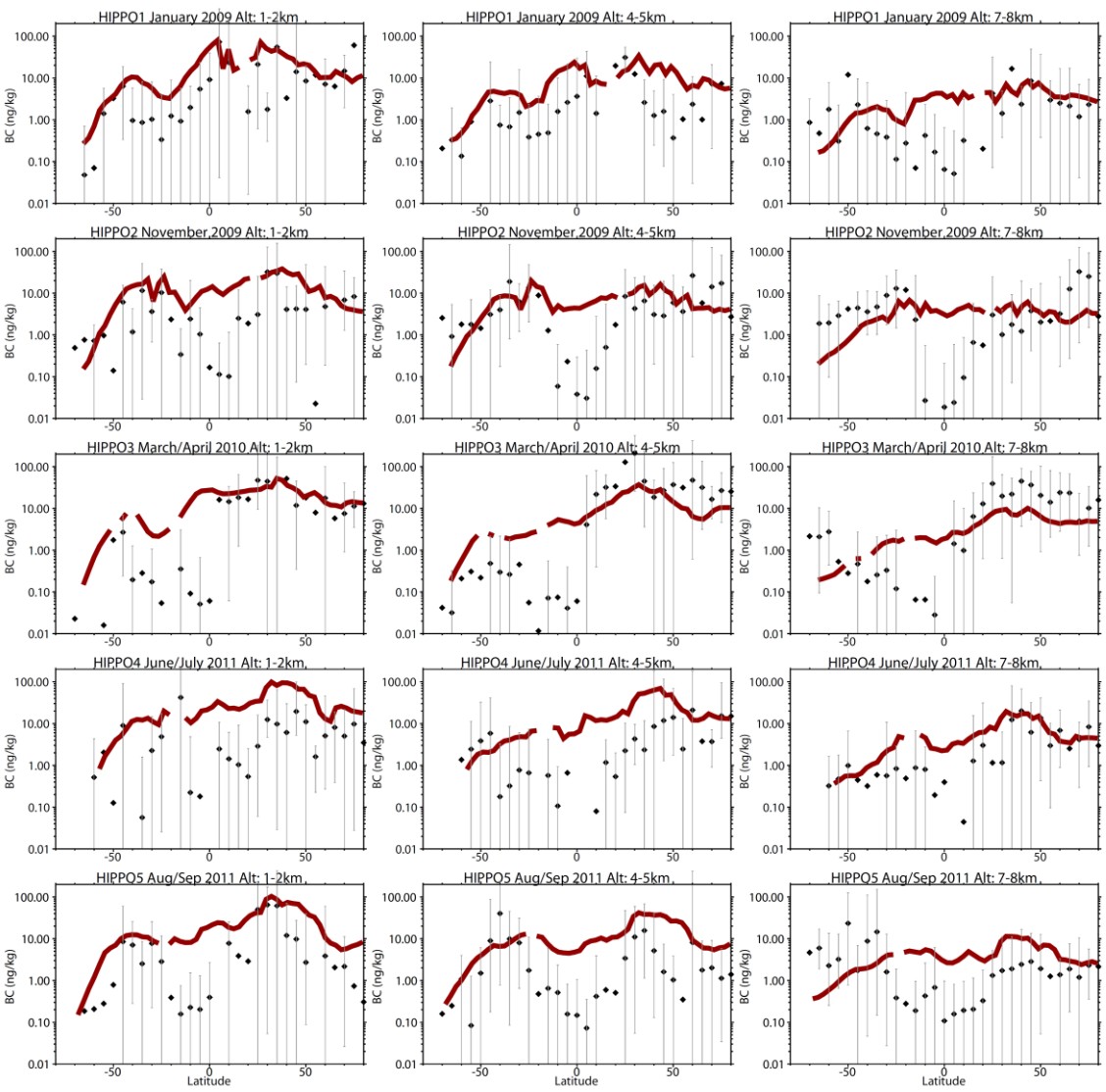

Figure 16. Comparison of modelled black carbon (BC) aerosol (red lines) with observations from HIPPO aircraft campaigns over the Pacific Ocean (black symbols, bars represent the full data range). Observations from different HIPPO campaigns were averaged over 5 °latitude bins and three different altitude bands (left column: 1-2 km, middle column: 4-5 km, and right column: 7-8 km) along the flight track over the Pacific Ocean. Model results were sampled along the flight track and then averaged over the abovementioned regions for comparison.

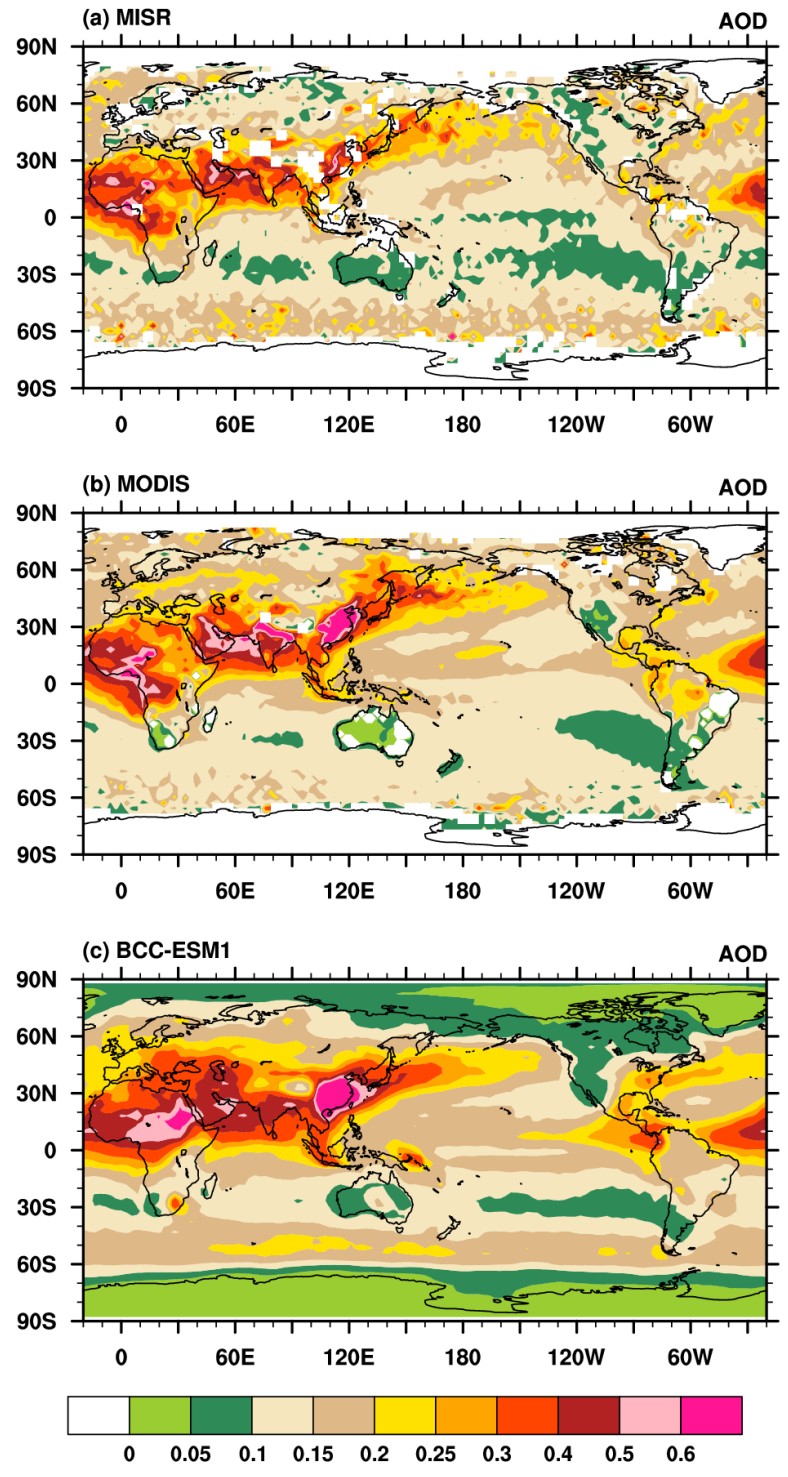

Figure 17. Global distribution of annual mean AOD simulated in BCC-ESM1 compared with the MISR and MODIS data for the year 2008.

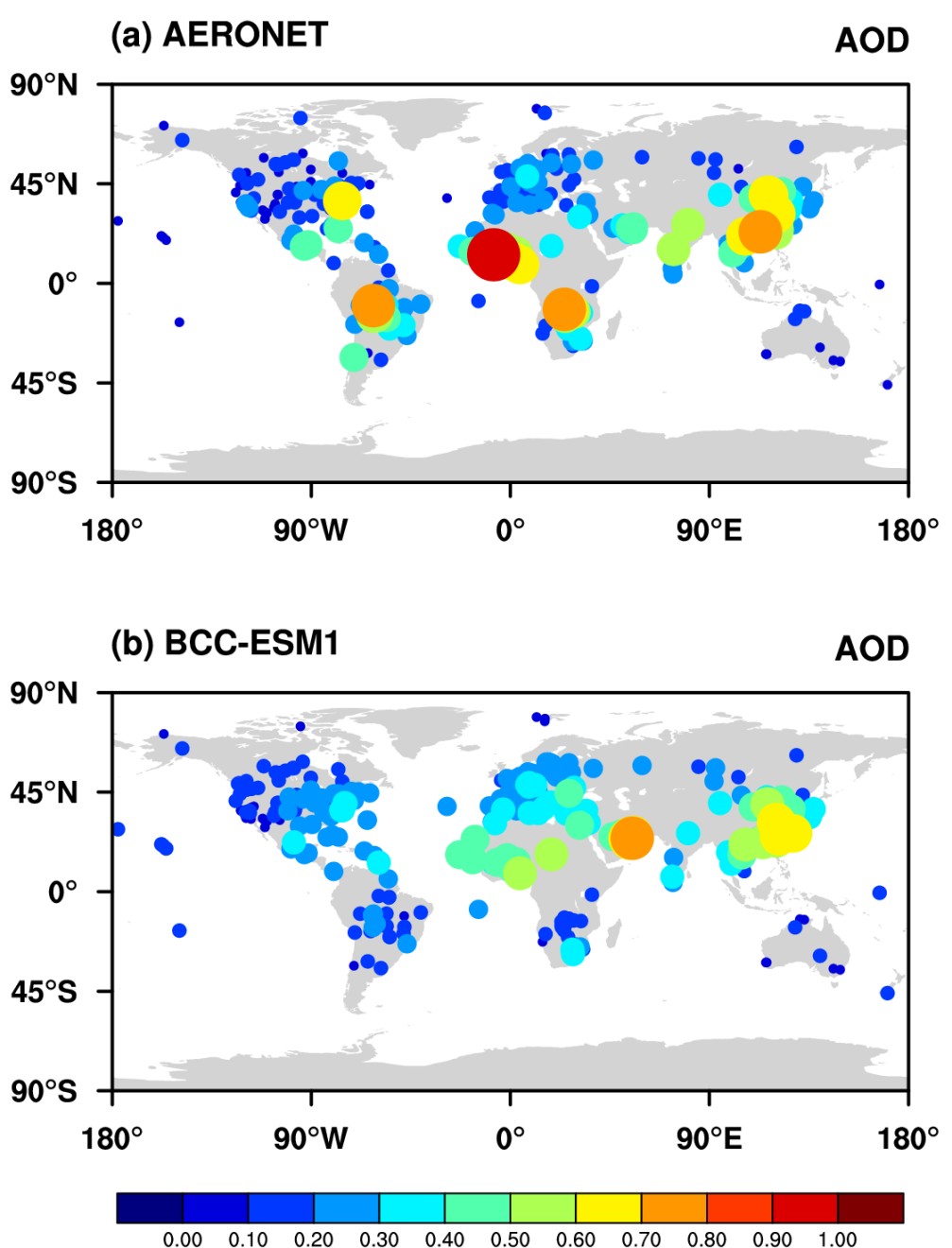

Figure 18. Observed versus simulated annual means of AOD at AERONET sites. Each data point represents the mean averaged for available monthly values of AOD. The dot sizes denote the magnitudes of AOD at sites. The spatial correlation is 0.56.

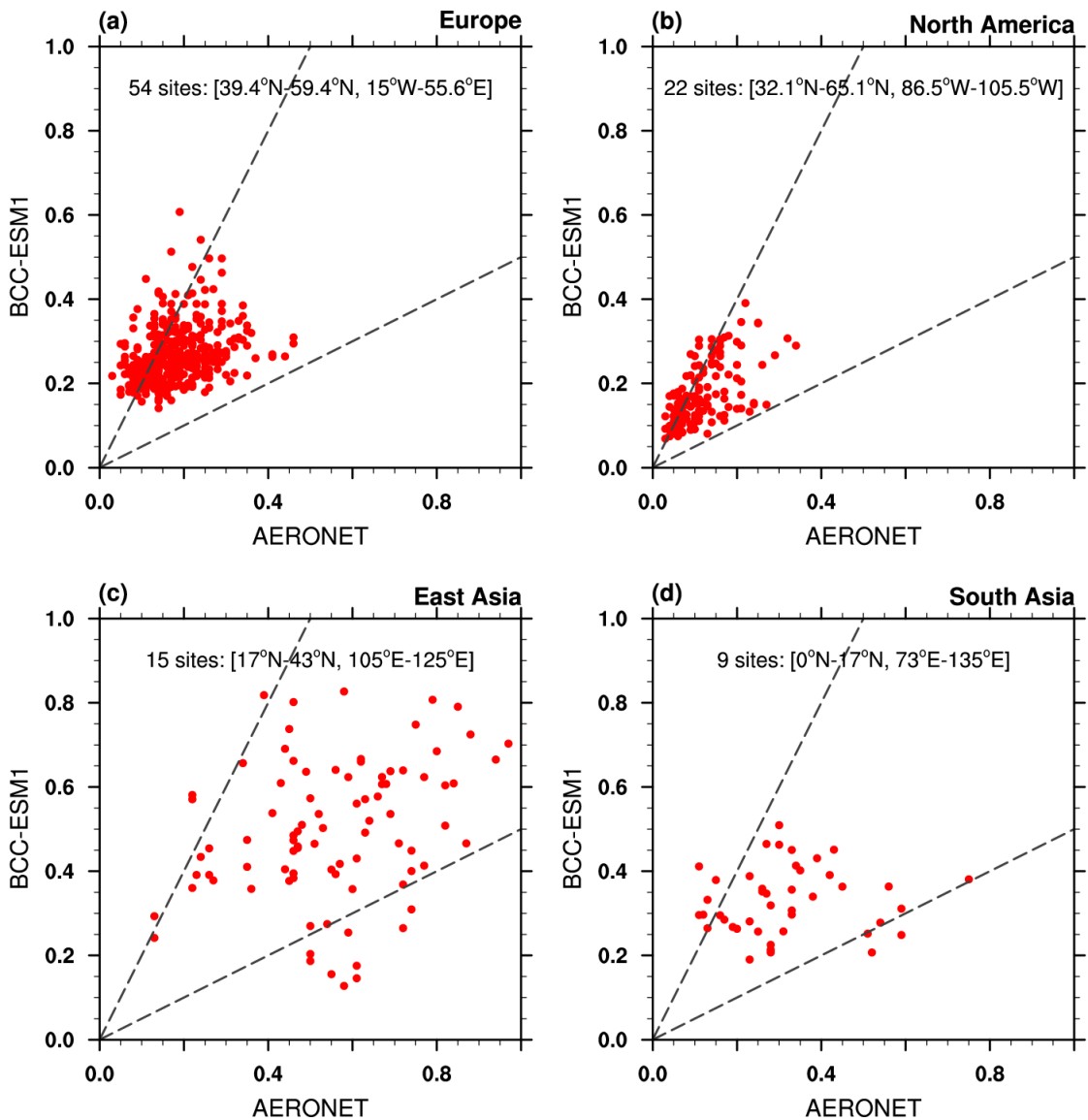

Figure 19. Scatter plots of observed versus simulated monthly mean AOD at AERONET sites in Europe, North America, East Asia, and South Asia over the period of 1998-2005.