# Peer review of "Beijing Climate Center Earth System Model version 1 (BCC-ESM1): Model Description and Evaluation of Aerosol Simulations"

_Geoscientific Model Development, 2019_

## Referee Comment (RC1) · Jean-Francois Lamarque (Referee) · 10 Sep 2019

This paper provides a description and evaluation of the aerosols in the BCC-ESM. The paper provides a reasonable overview of the model characteristics and sufficient comparisons to be useful. However, it suffers from a certain number of omissions and lack of details that should be fixed before publication moves forward.

My main concern in this paper is the statement at lines 316-318: "The whole system in BCC ESM1 fluctuates around +0.7 W m-2 net energy flux at TOM without obvious trend in 600 years (Fig. 1b), and the global mean surface air temperature shows only a small warming (Fig. 1a)". If this is the case, then there is a real problem with this model. There cannot be a significant TOA imbalance without a significant trend in surface

temperature, unless the ocean is taking up all that excessive forcing. Which would mean huge drifts in the mean ocean temperature. The authors need to clearly identify if this is a mistake, or the difference between TOA and TOM, or whether there is a drift in ocean temperatures. But as stated, this means there is a huge non-conservation of energy in the model.

Another concern is that the authors make considerable use of the CMIP5 concentrations (by the way, a correct reference to this data would be Lamarque et al., ACP, 2010), which is a somewhat circular evaluation. Indeed, the CMIP5 data were generated using a chemistry model very similar to the one used in BCC-ESM1. It is true that the emissions are different, but then the main evaluation this analysis provides is on the similarity of emissions. I would therefore encourage the authors to expand their model evaluations to include more observations. For example, the paper https://www.geosci-model-dev.net/9/1853/2016/gmd-9-1853-2016.pdf includes analysis against aircraft observations.

While I understand that the focus is on aerosols, it cannot be ignored that the rate of formation of sulfate is dependent on the levels of oxidants in the troposphere. It would therefore be very useful if some documentation and evaluation of oxidants (at the very least ozone) is included in the paper.

Minor comments

1. Lines 155-157: why is convective transport not considered?

2. Lines 189-191: Following the work done in CAM4, it would be quite straightforward to include some basic representation of NH3 chemistry (see Lamarque et al., GMD, 2012, section 5).

3. Line 207: reference Hoesly et al.

4. Lines 251-254: this is an important aspect of the model that needs more discussion. In particular, what is the aerosol indirect effect in this model?

5. Line 257 (and other places): it is AerChemMIP, not AeroChemMIP

6. Lines 273-276: which emissions are those? The CMIP6 (as the CMIP5) had all emissions necessary for tropospheric chemistry, as long as some splitting of lumped emissions (like total VOC emissions) were performed.

7. Line 288: volcanic, not volcano

8. Line 290: this is confusing. It is really not clear that stratospheric aerosols are represented in this model. Are those really stratospheric emissions, or tropospheric emissions of the non-eruptive volcanoes?

9. Line 293: what are the total NOx emissions from lightning (in TgN/year)?

10. Lines 301-303: this is not clear. Are you describing the relaxation time (of 10-days) of the concentrations towards the climatology? Is the climatology changing over the course of the historical period?

11. Line 337: there are some anthropogenic/biomass SO2 emissions in 1850, just small ones.

12. Line 373: the correlation really only reflects that the lifetime of SO2 is very short and not changing much, and therefore the burden will directly follow the emissions.

13. Line 376: what is the "NCAR data package"?

14. Line 400 (and others): a lot of analysis compares to Liu et al (2005). It would be useful to include more publications, especially more recent ones.

15. Figure 5: why is the BCC ESM1 data also shown as 10-year averages? Also, are those the results of a single ensemble member? More details on the simulation would be useful; in particular I am assuming that this is a fully coupled simulation.

---

## Referee Comment (RC2) · Anonymous Referee #2 · 14 Sep 2019

**General**

The paper presents a description and an evaluation of the tropospheric aerosols included in the Beijing Climate Center Earth System Model. The paper consists of a quite general overview and my main concerns relate firstly to the description of the aerosol scheme which is rather vague in several parts. Furthermore, it is not so clear what is specific or not to this aerosol scheme, what has been developed and/or adjusted, compared to other schemes already in place in other climate models. Secondly, the evaluation is mostly qualitative, and when some quantitative information is provided, it often refers to quite old references.

Both issues needs to be addressed quite thoroughly for the paper to progress in the

review process. This requires quite some work. My list of particular points appears below.

Questions/remarks

1. paragraph "Model description": this paragraph needs some rewriting as some features, eg ACGM3, AVIM2, are presented twice

2. L146: what is the reference for the "weighted-combination"? Please provide more details.

3. L156: is turbulent transport included? if not, then you are missing sub-grid scale transport and the overall distribution of chemical species would be quite different considering this sub-grid scale transport. Please explain what is your rationale for presenting an evaluation without these processes.

4. L173: the Wesely approach has 3 terms. Why did you retain only two terms? Please indicate if you compute the terms interactively or not. This is at the moment not clearly stated.

5. L179: it seems to me that the reactions listed in Table 2, and their reaction rates, are the same as the ones that appear in Lamarque et al. 2012. This should be noted in the paper, as therefore both the chemistry and the aerosol modules of the BCC-ESM1 and CAM-Chem used for generating what the authors refer as the "CMIP5 recommended" aerosol concentrations are quite similar. This should be made quite clear in the paper. Possibly a paragraph in the paper could be dedicated to what is specific to this scheme, if this is relevant.

6. L182: there is no reference to DMS in Benkowitz 1996. Please clarify what you mean

7. L191: is there a reference for this assumption?

8. L215: please clarify why in this paragraph about OC and BC your write about "soluble gases"?

9. L224: what are the values of this scaling factor?

10. L252: Wu et al 2019 is not in the list of references ; and what do you mean by "it is parameterized", what is "it"? Do you refer to the aerosol first indirect effect or to the first and second effects? Please provide further details, in particular if you parameterize the second indirect effect of aerosols that not all climate models consider

11. L257: "historical" is not an AerChemMIP simulation but rather a CMIP6 simulation that will be a basis for a large number of CMIP6 analyses, including some AerChemMIP analyses, but also other MIPs analyses. Please correct this wording throughout the document.

    If the simulation you present is an historical CMIP6 simulation, please indicate the baseline name of the corresponding files on the ESGF. Do you present one ensemble member or several members?

12. L264: "only $O_3$ is a prognostic variable": what about $CH_4$, it is part of the chemistry scheme and therefore it is also a prognostic variable isn't it? what about also $CO_2$?

13. L274: the CMIP6 anthropogenic emissions are meant to cover all that is required for a climate model. Can you explain why this was not the case for your model?

14. L276: to my knowledge there is no such CMIP6 recommendation for hydrophobic and hydrophilic forms. Please rephrase your sentence.

15. L279 and following: please describe in more details the formation of Secondary Organic Aerosol from vegetation that you consider? what comes out of MEGAN2.1, are they related to OC2 only, and not OC2 and OC1? ...

16. L291: factor 2-4 high: this is a strong affirmation! The Ge et al. 2016 study is older than the CMIP6 data. How do they relate? And furthermore, do you have a stratospheric aerosol scheme that uses these data? If yes, please describe the scheme, if not please clarify your sentence.

17. L304: please clarify what the MOZART2 data package include, data? chemistry code?...

18. L307: to my knowledge the CMIP6 data package does not include neither $CH_4$, nor $N_2O$: what do you refer here to?

19. Table 1 and Table 4: there are incoherences between species listed in both Tables. For example, CH3COCHO is not emitted in Table 1 and has emissions in Table 4. Please carefully check consistency between these tables.

20. L318: "only a small warming": please quantify this

21. L324: mean and uncertainty should not be of different orders. Please correct here and in other places in paper.

22. L331: these are not concentrations but rather loads, and what is the reference for these "CMIP5 recommended concentrations"?

23. L338: why do you think there is such a distribution?

24. L350: in addition to pointing out similarities, please address differences between CMIP5 and BCC-ESM1 outputs, and why there are such differences/similarities

25. L376: what is this particular "NCAR data package"

26. L376: sentence "This decrease trend possibly results from the prescribed emissions have not year-to-year variations and ..." is not clear

27. L386: the sentence "The trends of global BC and OC burdens are similar 387 to that of sulfate, but they showed continuous increases from 1950 to present." is not clear

28. L390: "was slightly enhanced from 1950 to 2000" : I rather see a similar burden in 1950 and in 2000. Please be clearer, and do you have evidence of increasing soil dryness during that period?

29. L400: "largely due to stronger wind speed": differences could be due to differences in underlying DMS concentrations in the oceans. What supports your affirmation?

30. L406: air traffic is part of anthropogenic activities; please rephrase your sentence, and what about biomass burning emissions?

31. L407: you indicate that volcanic emissios are not included. I wonder in Figure 3 what corresponds to the area of large loads of sulfate around Central America?

32. L423: it seems that the total of 45.2 Tg/yr for OC is incoherent with what appears in Figure 4b; please correct.

33. L490: please provide some quantitative information with these plots, as for instance appears in the AeroCom web page woth scatter plots (https://aerocom.met.no/cgi-bin/surfobs_annualrs.pl)

34. L500: please provide some quantitative elements on the extinction coefficients, also single scattering albedo and asymmetry parameters

35. L502 (and paragraph): do you show a 1997-2003 average or the 2008 year as indicated in the figure; please provide quantitative information (bias, rmse..., or

normalized figures as you prefer). This comment is valid for all figures. They should all be accompanied with some quantitative information.

36. L516: I don't feel the evaluation is "comprehensive" so far. Please review this affirmation as you make some progress in a future version of the paper.

37. L530: you indicate that you used prescribed concentrations for $CH_4$, and in Table 4 you indicate that you consider $CH_4$ emissions. Please clarify.

38. L541: there is no such compararison of all of these aerosols with observations. Please be more precise.

39. L560: I don't understand "How about the GHGs simulations in the AeroChemMIP historical run?" please be clearer and more precise.

40. Figure 4: please add biomass burning emissions, if not done yet, or indicate if they are already part of the figure

41. Figure 14: do you compare monthly observations averaged over 1998-2005 with monthly model outputs averaged over 1998-2005? please formulate more precisely

Minor questions/remarks

1. L1: the title is misleading and should be changed at the paper focuses on tropospheric aerosols

2. L49: "Besides gaseous"

3. L51: aerosol are particles; so change "aerosol particles" to "aerosols"

4. L59, and others: homogeneise writing of chemical compounds, for instance $O_3$

5. L99: "BCC-ESM1 is a fully-coupled global climate-chemistry-aerosol model ": it seems to me that BCC-ESM1 is more than that; I would say it is an "Earth System Model with interactive chemistry and aerosol components" if you want to insist on these components

6. L120: change "used" to "uses"

7. L122: please clarify "ranged to"

8. L145: it is not clear whether deposition velocities are computed interactively, as in Wesely, or consist of monthly means.

9. L198: remove "Its"

10. L238: please be more precise on the Web page

11. L252: Wu et al 2019 is not in the list of references ; and what do you mean by "it is parameterized", what is "it"?

12. L263: AGCM-Chem1: is this the correct name of the model?

13. L264: please reformulate "at each model step and interacts with radiations"

14. L276: add "see Table 4"

15. L283: MEGAN acronym already introduced

16. L310: change "1850 AD conditions" to "1850 conditions"

17. L317: change "600" to "450"

18. L317: Figure 5b-5d

19. L385: early 1980s

20. L513: please correct the North American coordinates; and correct also in Figure 14 the European coordinates ; and furthermore the coordinates you indicate in the text do not correspond to those of Figure 14

21. L543: in relevant literature

22. Table 1: please indicate that interactive surface emissions are considered for sea salt and dust

23. Table 5: I could not find figures for the sinks of DMS in Liu 2005 Table 4. Where do your figures come from?

24. Table 6: f for Ginoux 2001

25. Figure 1: change SAT into tas official CMIP6 variable

26. Figure 5: what is the "20th historical simulations"? Same question in caption of Figure 11

27. Figure 5: change "blue" to "black"

---

## Author Comment (AC1) · 23 Oct 2019

Response to Reviewer 1 (Dr. Jean-Francois Lamarque)

We thank Dr. Jean-Francois Lamarque for his insightful and constructive comments. We revised our manuscript accordingly. Please see the attached file.

Tongwen Wu and Co-authors.

Please also note the supplement to this comment:
https://www.geosci-model-dev-discuss.net/gmd-2019-172/gmd-2019-172-AC1-supplement.pdf

---

## Author Response (AR1)

**Response to Reviewer 1 (Dr. Jean-Francois Lamarque)**

*We thank Dr. Jean-Francois Lamarque for his insightful and constructive comments. We revised our manuscript accordingly.*

This paper provides a description and evaluation of the aerosols in the BCC-ESM. The paper provides a reasonable overview of the model characteristics and sufficient comparisons to be useful. However, it suffers from a certain number of omissions and lack of details that should be fixed before publication moves forward.

My main concern in this paper is the statement at lines 316-318: "The whole system in BCC ESM1 fluctuates around +0.7Wm-2 net energy flux at TOM without obvious trend in 600 years (Fig. 1b), and the global mean surface air temperature shows only a small warming (Fig. 1a)". If this is the case, then there is a real problem with this model. There cannot be a significant TOA imbalance without a significant trend in surface temperature, unless the ocean is taking up all that excessive forcing. Which would mean huge drifts in the mean ocean temperature. The authors need to clearly identify if this is a mistake, or the difference between TOA and TOM, or whether there is a drift in ocean temperatures. But as stated, this means there is a huge non-conservation of energy in the model.

*We apologize for the confusion. TOM (top of model) should be TOA (top of atmosphere). We double checked our data used and there was indeed an imbalance of net energy flux at TOA. In order to verify whether there is a drift in ocean temperatures, Fig.1c representing the variation of global SST is added. It seems that the ocean is stable, at least for its upper layer. If we refer to other models of similar complexity, it seems that a small imbalance commonly exists (Hansen et al., 2005; Wild et al., 2013) and an average of 1.0 Wm-2 of imbalance is among the CMIP5 models (Wild et al., 2013). In the revised manuscript, we rewrote this paragraph (in lines 390-393 in the revised manuscript) as "This level of TOA energy imbalance is close to the average imbalance (1.0 Wm-2) among CMIP5 models (Wild et al., 2013), and does not cause remarkable climate drift in BCC-ESM1. The global mean TAS and SST keep around 288.1 K (Fig. 1b) and 295.05 K (Fig. 1c) respectively.".*

Another concern is that the authors make considerable use of the CMIP5 concentrations (by the way, a correct reference to this data would be Lamarque et al., ACP, 2010), which is a somewhat circular evaluation. Indeed, the CMIP5 data were generated using a chemistry model very similar to the one used in BCC-ESM1. It is true that the emissions are different, but then the main evaluation this analysis provides is on the similarity of emissions. I would therefore encourage the authors to expand their model evaluations to include more observations. For example, the paper https://www.geosci-model-dev.net/9/1853/2016/gmd-9-1853-2016.pdf includes analysis against aircraft observations. While I understand that the focus is on aerosols, it cannot be ignored that the rate of formation of sulfate is dependent on the levels of oxidants in the troposphere. It would therefore be very useful if some documentation and evaluation of oxidants (at the very least ozone) is included in the paper.

*We appreciate your very relevant comments. The right reference of Lamarque et al. (2010) is now used in the revised manuscript. We also agree entirely that the oxidation capacity should be evaluated, and we followed your suggestion by comparing the simulated O3 in the 20th century against CMIP6 prescribed data and global ozonesonde observations from WOUDC. We added a new section "4. Evaluation of O3 and aerosols simulation in the 20th century". Furthermore, a comparison of BC simulations against HIPPO BC aircraft observations is also added in "4.3 Global aerosol distributions at present day".*

Minor comments

1.  Lines 155-157: why is convective transport not considered?

*Vertical transport of gas tracers and aerosols due to deep convection is not yet included in the present version of BCC-AGCM3-Chem, which process is considered as a part of the deep convection and occurs generally in a small spatial region on a GCM-box with low-resolution (2.8°lat.×2.8°lon.). Another consideration is that a large uncertainty exists to treat transport of those water-soluble tracers by deep convection. We are working on this issue. This effect will be involved in the next version of BCC model. We feel it is important to mention it since we are aware that the issue can partly matter for the quality of results shown in this manuscript. We added this explanation in lines 146-151 in the revised manuscript.*

2. Lines 189-191: Following the work done in CAM4, it would be quite straightforward to include some basic representation of NH3 chemistry (see Lamarque et al., GMD, 2012, section 5).

*We apologize for this mistake about NH3. In fact, a previous version of BCC-ESM did not include NH3. But in the frozen version of BCC-ESM1 that is used in this work, NH3 is indeed a prognostic variable following CAM4 (Lamarque et al., GMD, 2012). So, we added some description about NH3 in "2.1 SO2, DMS, NH3, and Sulfate" and Table 1, Table 2, and Table 4 in the revised manuscript.*

3. Line 207: reference Hoesly et al.

*In the revision, we have added the reference of Hoesly et al. (2018) in line 223 in the revised manuscript.*

4. Lines 251-254: this is an important aspect of the model that needs more discussion. In particular, what is the aerosol indirect effect in this model?

*In the revision, we have added a paragraph in "2.5 Effects of aerosols on radiation, cloud, and precipitation" to describe the treatment of aerosol indirect effect in BCC-ESM1.*

5. Line 257 (and other places): it is AerChemMIP, not AeroChemMIP

*In the revision, we changed "AeroChemMIP" to "AerChemMIP".*

6. Lines 273-276: which emissions are those? The CMIP6 (as the CMIP5) had all emissions

necessary for tropospheric chemistry, as long as some splitting of lumped emissions (like total VOC emissions) were performed.

*In the revised manuscript, we added more details for this issue in lines 323-337.*

*"Most historical emissions from anthropogenic source (surface, aircraft plus ship) and biomass burning from 1850 to 2014 are CMIP6-recommended data (Hoesly et al., 2018; available at https://esgf-node.llnl.gov/search/input4mips). Anthropogenic or biomass burning sources of some tracers which are not included in the CMIP6 dataset (see Table 4), anthropogenic emission of H2 and N2O are from monthly climatological dataset provided by the MOZART-2 standard package. N2O is a prognostic variable in BCC-ESM1 but it is replaced by CMIP6 prescribed concentration in the historical run. Other emissions including biomass burning (CH3COCH3) and anthropogenic emission (CH3CHO, CH3OH, and CH3COCH3) are from the IPCC ACCMIP emission inventory (http://accent.aero.jussieu.fr/ACCMIP.php) covering the period from 1850 to 2010 with 10-year intervals (see Table 4). Monthly lumped emissions of black carbon and organic carbon aerosols from 1850 to 2014 are downloaded from CMIP6-recommended data, but we used 80% (for BC) and 50% (for OC) of them in their hydrophobic forms (BC1 and OC1) and the rest in their hydrophilic forms (BC2 and OC2), following the work of Chin et al. (2002)."*

*We check the CMIP6 data website again and cannot find anthropogenic emission data of H2 and N2O provided.*

7. Line 288: volcanic, not volcano

*In the revision, we have corrected to "3.2 Volcanic eruption, lightning and aircraft emissions"*

8. Line 290: this is confusing. It is really not clear that stratospheric aerosols are represented in this model. Are those really stratospheric emissions, or tropospheric emissions of the non-eruptive volcanoes?

*We apologize for the confusion. We don't have stratospheric chemistry scheme, and no stratospheric emissions at all. That statement in the initial manuscript indicates surface emissions from non-eruptive volcanos. In the revised manuscript, we rewrote the corresponding paragraph in "3.2 Volcanic eruption, lightning and aircraft emissions". It reads in lines 354-360 as "As there is no stratospheric aerosol scheme in BCC-ESM1, concentrations of sulfate aerosol at heights from 5 to 39.5 km, which volcanic origin, are directly prescribed using the CMIP6-recommended data (Thomasson et al., 2018) from 1850 to 2014. The effects of surface SO2 emissions from volcanic eruption on the variation of SO2 in the atmosphere and then on the variation of tropospheric SO42- concentration are considered, and the SO2 emissions from 1850 to 2014 are downloaded from the IPCC ACCMIP emission inventory (http://accent.aero.jussieu.fr/ACCMIP.php)."*

9. Line 293: what are the total NOx emissions from lightning (in TgN/year)?

*The globally-averaged mean of the total NOx emissions from lightning during the period of 1850 to 2014 is 5.19 Tg (N)·yr-1. It is in agreement with observations within the range of 3 to*

*6 Tg(N) yr-1 (Martin et al., 2002). In the revised manuscript, we modified the corresponding description in "3.2 Volcanic eruption, lightning and aircraft emissions"*

10. Lines 301-303: this is not clear. Are you describing the relaxation time (of 10-days) of the concentrations towards the climatology? Is the climatology changing over the course of the historical period?

*Yes, we are describing the relaxation time (10 days) that we used to relax different chemical variables toward their monthly and zonal mean climatological values, prescribed in the top two layers. During the revision, we rewrote the corresponding paragraph in "3.3 Upper boundary of the atmosphere" in lines 371-379 as*

*"Concentrations of different tracers (O3, CH4, N2O, NO, NO2, HNO3, CO, and N2O5) at the top two layers of the model are set to prescribed monthly climatological values, and concentrations from below the top two layers to the tropopause are relaxed at a relaxation time of 10-days towards the climatology. Climatological values of NO, NO2, HNO3, CO and N2O5 at the top two layers are extracted from MOZART2 data package available at the Website (https://www2.acom.ucar.edu/gcm/mozart-4), originated from the Study of Transport and Chemical Reactions in the Stratosphere (STARS, Brasseur et al., 1997). Concentrations for the other tracers (O3, CH4, and N2O) at the top two model layers are the zonally-averaged and monthly values from 1850 to 2014 derived from the CMIP6 data package."*

11. Line 337: there are some anthropogenic/biomass SO2 emissions in 1850, just small ones.
*Yes, that is true, anthropogenic emissions were not entirely negligible, although small in 1850. During the revision, we reformulated the corresponding paragraph in lines 411-415.*

*"We can compare them with CMIP5 recommended concentrations in year 1850, considered as the reference state in the pre-industrial stage. At that time, there are fewer anthropogenic/biomass SO2 emissions, the SO4 over land are evidently smaller than those over oceans especially over the tropical Pacific and Atlantic Oceans, where DMS can be oxidized to SO2 and then form SO4"*

12. Line 373: the correlation really only reflects that the lifetime of SO2 is very short and not changing much, and therefore the burden will directly follow the emissions.
*Yes, we agree entirely with this remark. We modified the descriptions in lines 481-484 as "Due to increasing SO2 emissions from 1850 to present day (Fig. 6), the global SO2 burden in the atmosphere increased from 100 Tg in 1850s to 200 Tg in 1980s (Fig. 7a), and has a high correlation coefficient of 0.996 with the anthropogenic emissions (Fig. 6a), as the lifetime of SO2 is short. The burden directly followed the emission".*

13. Line 376: what is the "NCAR data package"?
*It is MOZART2 package and corrected in lines 485-487 to "Its natural emissions from oceans from 1850 to 2010 in the model are the climatological monthly means (Dentener et al., 2006) from MOZART2 data package." in the revised manuscript.*

14. Line 400 (and others): a lot of analysis compares to Liu et al (2005). It would be useful to

include more publications, especially more recent ones.

*In the revision, we have added more comparison with recent publications such as Liu et al. (2016), Matsui and Mahowald (2017), Tegen et al. (2019) in "4.2 Global aerosols budgets".*

15. Figure 5: why is the BCC ESM1 data also shown as 10-year averages? Also, are those the results of a single ensemble member? More details on the simulation would be useful; in particular I am assuming that this is a fully coupled simulation.

*The 10-year averaged from BCC-ESM1 data used in the previous version of manuscript is only based on consideration for intercomparison with the 10-year interval CMIP5 data. In the revised manuscript, we updated those using the yearly mean simulations (Figure 5 is numbered to Figure 7 in the revised version).*

**Response to Referee #2**
The paper presents a description and an evaluation of the tropospheric aerosols included in the Beijing Climate Center Earth System Model. The paper consists of a quite general overview and my main concerns relate firstly to the description of the aerosol scheme which is rather vague in several parts. Furthermore, it is not so clear what is specific or not to this aerosol scheme, what has been developed and/or adjusted, compared to other schemes already in place in other climate models. Secondly, the evaluation is mostly qualitative, and when some quantitative information is provided, it often refers to quite old references.

Both issues needs to be addressed quite thoroughly for the paper to progress in the review process. This requires quite some work. My list of particular points appears below.

*We thank reviewer #2 for his/her carefully reading our manuscript. We have revised the manuscript accordingly. We showed more details and descriptions of the aerosol scheme, and added an evaluation for O3 simulation which is helpful to complement aerosols. We also presented more quantitative information, and comparisons with recent observation and references.*

**Questions/remarks**

1. paragraph "Model description": this paragraph needs some rewriting as some features, eg ACGM3, AVIM2, are presented twice

*We have rewritten this paragraph in "2. Model description". The components of the atmosphere, the land, the ocean and the sea ice in BCC-ESM1 are described in separate paragraph.*

2. L146: what is the reference for the "weighted-combination"? Please provide more details.

*In the revision, we have reworded this sentence to make our point clearer and rewritten this paragraph in lines 127-137 as "Dry deposition velocities for the 15 trace gases including O3, CO, CH4, CH2O, CH3OOH, H2O2, NO2, HNO3, PAN, CH3COCH3, CH3COOOH, CH3CHO, CH3COCHO, NO, and HNO4 are not computed interactively and directly interpolated from MOZART2 climatological monthly mean deposition velocities (https://en.wikipedia.org/wiki/MOZART(model)) which are calculated offline (Bey et al., 2001; Shindell et al., 2008) using a resistance-in-series scheme originally described in Wesely (1989). The dry deposition velocities for the other 15 species including MPAN, ONIT, ONITR, C2H5OH, POOH, C2H5OOH, C3H7OOH, ROOH, GLYALD, HYAC,*

*CH3OH, MACROOH, ISOPOOH, XOOH, HYDRALD, and H2 are calculated using prescribed deposition velocities of O3, CO, CH3CHO, or land surface type and surface temperature following the MOZART2 (Horowitz et al., 2003)."*

3. L156: is turbulent transport included? if not, then you are missing sub-grid scale transport and the overall distribution of chemical species would be quite different considering this sub-grid scale transport. Please explain what is your rationale for presenting an evaluation without these processes.

*Yes, Vertical transport of gas tracers and aerosols due to deep convection is not yet included in the present version of BCC-AGCM3-Chem, which process is considered as a part of the deep convection and occurs generally in a small spatial region on a GCM-box with low-resolution (2.8°lat. ×2.8°lon.). Another consideration is that a large uncertainty exists to treat transport of those water-soluble tracers by deep convection. But this effect will be involved in the next version of BCC model. We have added those expressions in lines 146-151.*

4. L173: the Wesely approach has 3 terms. Why did you retain only two terms? Please indicate if you compute the terms interactively or not. This is at the moment not clearly stated.

*Yes, the Wesely (1989) approach has 3 terms. We have cleared clarified this in lines 184-187 in the revised manuscript as "The dry deposition velocity of SO2 follows the resistance-in-series approach of Wesely (1989) using the formula, $W_{SO2} = 1/(r_a + r_b + r_c)$, in which $r_a$, $r_b$, and $r_c$ are the aerodynamic resistance, the quasi-laminar boundary layer resistance, and the surface resistance, respectively and they are interactively computed in each model time step."*

5. L179: it seems to me that the reactions listed in Table 2, and their reaction rates, are the same as the ones that appear in Lamarque et al. 2012. This should be noted in the paper, as therefore both the chemistry and the aerosol modules of the BCC-ESM1 and CAM-Chem used for generating what the authors refer as the "CMIP5 recommended" aerosol concentrations are quite similar. This should be made quite clear in the paper. Possibly a paragraph in the paper could be dedicated to what is specific to this scheme, if this is relevant.

*Yes, the reactions listed in Table 2 are referred to CAM-Chem (Lamarque et al., 2012). We have rewritten the description in lines 178-180 in the revised manuscript as "The present version of aerosol scheme belongs to a bulk aerosol model and mainly refers to the scheme of CAM-Chem (Lamarque et al., 2012), but the nucleation and coagulation of aerosols are still ignored."*

6. L182: there is no reference to DMS in Benkowitz 1996. Please clarify what you mean.

*That is our confusion about the reference. In the revision of the manuscript, we have reworded this sentence in lines 193-196 as "The main source of DMS is from oceanic emissions via biogenic processes. It is prescribed with the climatological monthly data that are extracted from MOZART2 package (https://www2.acom.ucar.edu/gcm/mozart-4)."*

7. L191: is there a reference for this assumption?

*That is related to "NH3". In the first version of manuscript, we make a mistake about NH3. In fact, the previous version of BCC-ESM did not include NH3 simulation in the chemistry scheme. But in the frozen version of BCC-ESM1 that is used in this work, NH3 is already set as a prognostic variable following CAM4 (Lamarque et al., GMD, 2012). So we added a description about NH3 in "2.1 SO2, DMS, NH3, and Sulfate" and Table 1, Table 2, and Table 4 in the revised manuscript.*

8. L215: please clarify why in this paragraph about OC and BC your write about "soluble gases"?

*We have rewritten the description in line 226-228 as "OC2 and BC2 are soluble aerosols, and their sinks are primarily governed by wet deposition. Their in- and below-cloud scavenging follows the scheme of Neu and Prather (2011)".*

9. L224: what are the values of this scaling factor?

*We have clarified it in lines 236-237 as "$S$ is a scaling factor and set to $4.05 \times 10^{-15}$, $4.52 \times 10^{-14}$, $1.15 \times 10^{-13}$, $1.20 \times 10^{-13}$ for four bins of sea salt aerosols (Table 4), respectively."*

10. L252: Wu et al 2019 is not in the list of references; and what do you mean by "it is parameterized", what is "it"? Do you refer to the aerosol first indirect effect or to the first and second effects? Please provide further details, in particular if you parameterize the second indirect effect of aerosols that not all climate models consider

*We have added Wu et al 2019 in the list of references, and added a paragraph in "2.5 Effects of aerosols on radiation, clouds, and precipitation" to describe the treatment of aerosol indirect effect in BCC-ESM1. In the first version of manuscript "it is parameterized" means "liquid cloud droplet number concentration is parameterized". Its details are added.*

11. L257: "historical" is not an AerChemMIP simulation but rather a CMIP6 simulation that will be a basis for a large number of CMIP6 analyses, including some AerChemMIP analyses, but also other MIPs analyses. Please correct this wording throughout the document. If the simulation you present is an historical CMIP6 simulation, please indicate the baseline name of the corresponding files on the ESGF. Do you present one ensemble member or several members?

*We have rewritten description about "historical" experiment in "3. Experiment design for the 20th century climate simulation". It followed the historical simulation protocol designed by CMIP6 (Eyring et al., 2016) which is named as "historical" in the Earth System Grid Federation (ESGF). The protocol details the historical experiment forced with emissions evolving from 1850 to 2014 refer to Collins et al. (2017). Three members of historical experiments are conducted and the first member is analyzed in this work.*

12. L264: "only O3 is a prognostic variable": what about CH4, it is part of the chemistry scheme and therefore it is also a prognostic variable isn't it? what about also CO2?

*CH4 and N2O may be selected as prognostic variables. But both are suggested in AerChemMIP to take prescribed values for the historical experiment. CO2 is also prescribed using CMIP6 historical forcing data. We have clarified this point in lines*

*307-312 in the revised manuscript.*

13. L274: the CMIP6 anthropogenic emissions are meant to cover all that is required for a climate model. Can you explain why this was not the case for your model?

*Anthropogenic emissions for most tracers are available in the CMIP6 data. But we cannot find anthropogenic emission data for H2 and N2O that we need. The details about the emission data used are given in the revised manuscript.*

14. L276: to my knowledge there is no such CMIP6 recommendation for hydrophobic and hydrophilic forms. Please rephrase your sentence.

*Yes, there is no such CMIP6 recommendation for hydrophobic and hydrophilic forms. So, we use monthly lumped emissions of black carbon and organic carbon aerosols and then we divided them separately to 80% of BC and 50% of OC emitted in their hydrophobic forms (BC1 and OC1) and the rest being in their hydrophilic forms (BC2 and OC2) following the work of Chin et al. (2002). This is cleared in lines 333-337.*

15. L279 and following: please describe in more details the formation of Secondary Organic Aerosol from vegetation that you consider? what comes out of MEGAN2.1, are they related to OC2 only, and not OC2 and OC1? ...

*OC does not belong to biogenic volatile organic carbons (VOCs). The hydrophilic organic carbon (OC2) can be formed from natural biogenic volatile organic compound (VOC) emissions. It is calculated online in the land component model BCC-AVIM2 and assumed to equal to 10% of monoterpenes emission following the algorithm of Chin et al. (2002). Those expressions are added in lines 348-352.*

16. L291: factor 2-4 high: this is a strong affirmation! The Ge et al. 2016 study is older than the CMIP6 data. How do they relate? And furthermore, do you have a stratospheric aerosol scheme that uses these data? If yes, please describe the scheme, if not please clarify your sentence.

*The work of Ge et al. 2016 is not mentioned, and this statement is now removed in the revised manuscript. As for stratospheric aerosol, we only considered SO4. We have rewritten this paragraph in lines 354-360 as "As there is no stratospheric aerosol scheme in BCC-ESM1, concentrations of sulfate aerosol at heights from 5 to 39.5 km, which volcanic origin, are directly prescribed using the CMIP6-recommended data (Thomasson et al., 2018) from 1850 to 2014. The effects of surface SO2 emissions from volcanic eruption on the variation of SO2 in the atmosphere and then on the variation of tropospheric SO42- concentration are considered, and the SO2 emissions from 1850 to 2014 are downloaded from the IPCC ACCMIP emission inventory (http://accent.aero.jussieu.fr/ACCMIP.php)."*

17. L304: please clarify what the MOZART2 data package include, data? Chemistry code?...

*We have clarified those in lines 374-376 as "Climatological values of NO, NO2, HNO3, CO and N2O5 at the top two layers are extracted from MOZART2 data package available at the Website (https://www2.acom.ucar.edu/gcm/mozart-4), originated from the Study of Transport and Chemical Reactions in the Stratosphere (STARS, Brasseur et al., 1997)."*

*Yes, MOZART2 data package includes data and chemistry code.*

18. L307: to my knowledge the CMIP6 data package does not include neither CH4, nor N2O: what do you refer here to?

*We have checked them. The CMIP6 data package includes zonally and monthly values of CH4 and N2O.*

19. Table 1 and Table 4: there are incoherences between species listed in both Tables. For example, CH3COCHO is not emitted in Table 1 and has emissions in Table 4. Please carefully check consistency between these tables.

*We have corrected the incoherence between Tables 1 and 4.*

20. L318: "only a small warming": please quantify this

*We have rewritten this paragraph in lines 381-395 of the revised manuscript and added the time series of global SST in Figure 1.*

21. L324: mean and uncertainty should not be of different orders. Please correct here and in other places in paper.

*We have corrected those expressions in "3.4 The preindustrial model states"*

22. L331: these are not concentrations but rather loads, and what is the reference for these "CMIP5 recommended concentrations"?

*Figures 2a-2c show the time series of global annual total masses in the troposphere (integrated from the surface to 100 hPa) in the last 450 years of the piControl. It is derived from CMIP5 recommended concentrations. The reference of CMIP5 data is Lamarque et al. (2010) and has added in the text.*

23. L338: why do you think there is such a distribution?

*We added some words about the distribution of SO42- in year 1850 in lines 411-415 of the revised manuscript as "We can compare them with CMIP5 recommended concentrations in year 1850, considered as the reference state in the pre-industrial stage. At that time, there are fewer anthropogenic/biomass SO2 emissions, the SO4 over land are evidently smaller than those over oceans especially over the tropical Pacific and Atlantic Oceans, where DMS can be oxidized to SO2 and then form SO4."*

24. L350: in addition to pointing out similarities, please address differences between CMIP5 and BCC-ESM1 outputs, and why there are such differences/similarities

*We have added sentences in lines 428-432 as "Relative lower relations for sulfate, black carbon and organic carbon are possibly caused as different anthropogenic emission sources are used in BCC-ESM1 and to create CMIP5 data. Dust and sea salts belong to natural aerosols and depend on the land and sea surface conditions, so their spatial distributions are easy to be captured and have relatively higher correlations between CMIP5 data and BCC-ESM1 simulations."*

25. L376: what is this particular "NCAR data package"

*We have corrected it in lines 485-487. It is MOZART2 data package.*

26. L378: sentence "This decrease trend possibly results from the prescribed emissions have not year-to-year variations and ..." is not clear

*It is modified in lines 487-491 as "As shown in Fig 7a, the global amount of DMS in the whole atmosphere was about 0.12 Tg during 1850-1900 and decreased to 0.055 Tg in 2010. This decrease trend maybe partly results from the speeded rate of DMS oxidation with global warming, and the loss of DMS gradually exceeds the source of ocean DMS emission to cause a net loss of DMS in the atmosphere since 1910s"*

27. L386: the sentence "The trends of global BC and OC burdens are similar to that of sulfate, but they showed continuous increases from 1950 to present." is not clear

*This sentence is modified in lines 496-499 of the revised manuscript as "As for global BC and OC burdens, BCC-ESM1 results show continuous increases since 1850s, especially from 1950 to present. From 1910's to 1940's, the CMIP5 data show a slight decrease of BC and OC burdens in the atmosphere."*

28. L390: "was slightly enhanced from 1950 to 2000" : I rather see a similar burden in 1950 and in 2000. Please be clearer, and do you have evidence of increasing soil dryness during that period?

*We have corrected its description in lines 501-504. Global dust burden in the period from 1980 to 2000, not from 1950 to 2000, shows evident increase. The details about the temperature and soil moisture in drought areas will be explored in other paper.*

29. L400: "largely due to stronger wind speed": differences could be due to differences in underlying DMS concentrations in the oceans. What supports your affirmation?

*DMS emission from the ocean is computed by wind near the sea surface. We have not compared the wind simulations in BCC-ESM with the data used in Liu et al. (2005). So, we cancelled the original description to account for their difference of DMS emission from oceans between BCC-ESM1 and the values in Liu et al. (2005).*

30. L406: air traffic is part of anthropogenic activities; please rephrase your sentence, and what about biomass burning emissions? biomass burning emissions, $SO_2$ from volcanic eruption?

*We have modified descriptions in lines 512-516. There are three parts of SO2 source listed in Table 5. One is produced from the DMS oxidation, the second is from airplane emissions to the atmosphere, and the rest included emissions from anthropogenic activities and volcanic eruption at surface.*

31. L407: you indicate that volcanic emissios are not included. I wonder in Figure 3 what corresponds to the area of large loads of sulfate around Central America?

*Corrected it. Volcanic emission of SO2 at surface is included.*

32. L423: it seems that the total of 45.2 Tg/yr for OC is incoherent with what appears in Figure 4b; please correct.

*In the Table 6, the units of OC sources and sinks are Tg (OM)/yr in order to compare with the data of Liu et al. (2012), and assumed OC equal to OM/1.4. We have transferred the units of OC sources and sinks to Tg (OC) yr−1 in Table 6 to keep coherence with the data in Figure 4b.*

33. L490: please provide some quantitative information with these plots, as for instance appears in the AeroCom web page with scatter plots (https://aerocom.met.no/cgi-bin/surfobs_annualrs.pl)

*We have added some statistical values such as Table 7 to list the regional mean and spatial standard deviation, minimum and maximum values at IMPROVE and EMEP network sites versus simulated concentrations of sulfate (SO42-), organic carbon (OC), black carbon (BC), and the spatial correlation between observed and simulated multi-years averaged annual means.*

34. L500: please provide some quantitative elements on the extinction coefficients, also single scattering albedo and asymmetry parameter3

*As limited length of the text, the other optical feature of aerosols such as extinction coefficients, single scattering albedo and asymmetry parameters, and even their feedbacks on radiation and global temperature change will be explored in the other paper. It is mentioned in lines 706-709 in "5. Summary and discussions"*

35. L502 (and paragraph): do you show a 1997-2003 average or the 2008 year as indicated in the figure; please provide quantitative information (bias, rmse..., or normalized figures as you prefer). This comment is valid for all figures. They should all be accompanied with some quantitative information

*We have added Table 7 to list the regional mean and spatial standard deviation, minimum and maximum values at IMPROVE and EMEP network sites versus simulated concentrations of sulfate (SO42-), organic carbon (OC), black carbon (BC), and the spatial correlation between observed and simulated multi-years averaged annual means.*

36. L516: I don't feel the evaluation is "comprehensive" so far. Please review this affirmation as you make some progress in a future version of the paper3

*"comprehensive" is changed to "primary" in line 660.*

37. L530: you indicate that you used prescribed concentrations for CH4, and in Table 4 you indicate that you consider CH4 emissions. Please clarify

*CH4 is a prognostic variable in the chemistry scheme of BCC-ESM1. So, emission of CH4 listed in Table 4 is used to simulate CH4 concentration, but some WMGHGs such as CH4, N2O, CO2, CFC11 and CFC12 according to the experimental protocol of AerChemMIP are prescribed using CMIP6 prescribed concentrations (to replace prognostic values of*

*CH4 and N2O from the chemistry scheme). It is clarified in "3. Experiment design for the 20th century climate simulation" in the revised manuscript.*

38. L541: there is no such compararison of all of these aerosols with observations. Please be more precise.

*Modified the description in lines 684-685 in the revised manuscript as "Global budgets of aerosols were evaluated through comparisons of BCC-ESM1 results for 1990-2000 with reports in various literatures for sulfate, BC, OC, sea salt, and dust.".*

39. L560: I don't understand "How about the GHGs simulations in the AeroChemMIP historical run?" please be clearer and more precise4

*O3 is evaluated in this work. Other GHGs such as CH4 and N2O concentrations can be simulated when forced with emissions and their simulations also need to be evaluated in future. Those are added at the end of "5. Summary and discussions".*

40. Figure 4: please add biomass burning emissions, if not done yet, or indicate if they are already part of the figur4

*Modified the captions of Figure 4, and all the biomass burning emissions are included in natural emissions in (a)-(c).*

41. Figure 14: do you compare monthly observations averaged over 1998-2005 with monthly model outputs averaged over 1998-2005? please formulate more precise.

*The data plotted in Figure 14 (it is numbered to Figure 15 in the revised manuscript) are multi-years averaged annual means over the available years 1990–2005 for IMPROVE sites and 1995–2005 for EMEP sites and corresponding simulations. The caption of Figure 15 is rewritten as "Scatter plots showing observed versus simulated multi-years averaged annual mean sulfate (SO4), organic carbon (OC), black carbon (BC) mixing ratios at IMPROVE and EMEP network sites. Observations are averages over the available years 1990–2005 for IMPROVE sites, and 1995–2005 for EMEP sites."*

**Minor questions/remark1at the paper focuses on tropospheric aerosols**

1. L1: the title is misleading and should be changed at the paper focuses on tropospheric aerosols

*This is a very good suggestion. The title "Beijing Climate Center Earth System Model version 1 (BCC-ESM1): Model Description and Aerosols Simulation Evaluation" is changed to "Beijing Climate Center Earth System Model version 1 (BCC-ESM1): Model Description and Aerosols Simulation Evaluation"*

2. L49: "Besides gaseous"

*In the revision, we have rewritten this sentence to "Besides gaseous components, atmosphere also contains various aerosols, which are important for cloud formation and radiative transfer." in lines 52-53 of the revised manuscript.*

3. L51: aerosol are particles; so change "aerosol particles" to "aerosols"

*It is modified in line 54 of the revised manuscript.*

4. L59, and others: homogeneise writing of chemical compounds, for instance O3

*Expressions for chemical compounds similar to O3 in the whole text are modified to keep homogenies.*

5. L99: "BCC-ESM1 is a fully-coupled global climate-chemistry-aerosol model ": it seems to me that BCC-ESM1 is more than that; I would say it is an "Earth System Model with interactive chemistry and aerosol components" if you want to insist on these components

*We thank the reviewer for pointing this out. This sentence is rewritten to "BCC-ESM1 is an Earth System Model with interactive chemistry and aerosol components." in lines 101-102 of the revised manuscript.*

6. L120: change "used" to "uses"

*It is corrected.*

7. L122: please clarify "ranged to"

*The sentence is rewritten to "MOM4-L40 uses a tripolar grid of horizontal resolution with 1º longitude by 1/3º latitude between 30ºS and 30ºN ranged to 1º longitude by 1º latitude from 60ºS and 60ºN poleward and 40 z-levels in the vertical." in lines 161-163.*

8. L145: it is not clear whether deposition velocities are computed interactively, as in Wesely, or consist of monthly means.

*We have clarified those expressions in lines 127-137 of the revised manuscript.*

9. L198: remove "Its"

*In the revision, we have removed the word "Its".*

10. L238: please be more precise on the Web page

*The Web page is https://svn-ccsm-inputdata.cgd.ucar.edu/trunk/inputdata/atm/cam/dst/. It is denoted as in line 251 in the revised manuscript.*

11. L252: Wu et al 2019 is not in the list of references; and what do you mean by "it

is parameterized", what is "it"?

*The reference of Wu et al.(2019) is added in the list of references. The sentence "it is*

*parameterized" means "liquid cloud droplet number concentration is parameterized". We have added the description about its parameterization in "2.5 Effects of aerosols on radiation, cloud, and precipitation" in the manuscript.*

12. L263: AGCM-Chem1: is this the correct name of the model?

*"BCC-AGCM3-Chem" is the name of the atmosphere component model of BCC-ESM1. It is corrected in line 261 of the revised manuscript.*

13. L264: please reformulate "at each model step and interacts with radiations"

*We have rewritten this expression in lines 307-310 of the revised manuscript.*

14. L276: add "see Table 4"

*It is modified.*

17. L283: MEGAN acronym already introduced

*It is modified.*

18. L310: change "1850 AD conditions" to "1850 conditions".

*It is modified.*

19. L317: change "600" to "450"

*It is modified.*

20. L385: early 1980s

*It is modified.*

21. L513: please correct the North American coordinates; and correct also in Figure 14 the European coordinates; and furthermore the coordinates you indicate in the text do not correspond to those of Figure 14

*Figure 14 is numbered to Figure 19 in the revised manuscript. We have corrected the legends in Figure 19 and the expressions in the text.*

22. L543: in relevant literature

*"in relevant literatures" is corrected to "in relevant literature".*

23. Table 1: please indicate that interactive surface emissions are considered for sea salt and dust

*We added the expression "In the column of surface emission, interactive surface emissions*

*are considered for sea salt and dust." in the caption of Table 1.*

24. Table 5: I could not find figures for the sinks of DMS in Liu 2005 Table 4. Where do your figures come from?

*It is our mistake as our references confusing and cancelled in the revised manuscript.*

25. Table 6: f for Ginoux 2001

*Ginoux et al. (2001) is added in the list of references.*

26. Figure 1: change SAT into tas official CMIP6 variable

*It is modified in Figure 1.*

27. Figure 5: what is the "20th historical simulations"? Same question in caption of Figure 11

*The expression of "20th historical simulations" is changed to "CMIP6 historical simulations" in Figures 7 and 14 in the revised manuscript.*

28. Figure 5: change "blue" to "black"

*It is modified. Figure 5 is renumbered to Figure 7 in the revised manuscript.*

[revised manuscript text omitted]
 land component is BCC Atmosphere and Vegetation Interaction Model version 2.0 (BCC-AVIM2.0) with terrestrial carbon cycle.The land component BCC-AVIM2.0 is described in details in Li et al. (2019). It includes biophysical, physiological, and soil carbon nitrogen dynamical processes, and the terrestrial carbon cycle operates through a series of biochemical and physiological processes on photosynthesis and respiration of vegetation. Biogenic emissions from vegetation are computed online in BCC-AVIM2.0 following the algorithm of the Model of Emissions of Gases and Aerosols from Nature version 2.1 (MEGAN2.1, Guenther et al., 2012).The oceanic component is the Modular Ocean Model version 4 with 40 levels (hereafter MOM4-L40). The land component is BCC Atmosphere and Vegetation Interaction Model version 2.0 (BCC-AVIM2.0) with terrestrial carbon cycle. The sea ice component is Sea Ice Simulator (SIS). Different components of BCC-ESM1 interact with each other through fluxes of momentum, energy, water, carbon and other tracers at their interfaces. The coupling between the atmosphere and the ocean is done every hour. BCC-AGCM3-Chem is able to simulate global atmospheric composition and aerosols with anthropogenic emissions as forcing. It is developed on the basis of the recent version 3 of the Beijing Climate Center atmospheric general circulation model (hereafter BCC-AGCM3, Wu et al., 2019). The horizontal resolution of BCC-AGCM3-Chem is T42 (approximately $2.8125^\circ \times 2.8125^\circ$ 
[revised manuscript text omitted]

1009     submitted to Geos.Model.Dev.

1010 Madronich, S.: Photodissociation in the atmosphere 1. Actinic flux and the effect of ground

1011     reflections and clouds, J. Geophys. Res., 92, 9740–9752, 1987.

1012 Mahowald, N., Lamarque, J.-F., Tie, X., and Wolff, E.: Sea salt aerosol response to climate

change: last glacial maximum, preindustrial and doubled carbon dioxide climates, J. Geophys. Res., 111, D05303, doi:10.1029/2005JD006459, 2006.

Martin, R. V., et al.: Interpretation of TOMS observations of tropical tropospheric ozone with a global model and in situ observations, J. Geophys. Res., 107(D18), 4351, doi:10.1029/2001JD001480, 2002.

Matsui, H., and Mahowald, N.: Development of a global aerosol model using a two-dimensional sectional method: 2. Evaluation and sensitivity simulations, J. Adv. Model. Earth Syst., 9, 1887–1920, doi:10.1002/2017MS000937, 2017.

Mora, C., Wei, C.-L., Rollo, A., Amaro, T., Baco, A.R., Billett, D., Bopp, L., Chen, Q., Collier, M., Danovaro, R., Gooday, A.J., Grupe, B.M., Halloran, P.R., Ingels, J., Jones, D.O.B., Levin, L.A., Nakano, H., Norling, K., Ramirez-Llodra, E., Rex, M., Ruh, H.A., Smith, C.R., Sweetman, A.K., Thurber, A.R., Tjiputra, J. F., Usseglio, P., Watling, L., Wu, T., Yasuhara, M.: Biotic and human vulnerability to projected ocean biogeochemistry change over the 21st century, PLoS Biol 11(10): e1001682. doi:10.1371/journal.pbio.1001682, 2013.

NCAR Command Language (Version 6.6.2) [Software], Boulder, Colorado: UCAR/NCAR/CISL/TDD. http://dx.doi.org/10.5065/D6WD3XH5, 2019.

Neale, R. B., et al.: Description of the NCAR Community Atmosphere Model (CAM 4.0), NCAR Tech. Note, TN-485, pp. 212, Natl. Cent. for Atmos. Res., Boulder, Colo., 2010

Neale, R. and co-authors: Description of the NCAR Community Atmosphere Model (CAM5.0). NCAR/TN-486+STR, NCAR Technical Note, 2012.

[revised manuscript text omitted]

定义网格后不调整右缩进, 在相同样式的段落间不添加空格, 无孤行控制, 不调整西文与中文之间的空格, 不调整中文和数字之间的空格, 制表位: 不在 3.43 字符

字体:11 磅, 字体颜色: 文字 1

字体:11 磅, 字体颜色: 文字 1

字体:11 磅, 字体颜色: 文字 1

字体:11 磅, 字体颜色: 文字 1

字体:11 磅, 倾斜, 字体颜色: 文字 1

字体:11 磅, 字体颜色: 文字 1

字体:11 磅, 字体颜色: 文字 1

字体:11 磅, 字体颜色: 文字 1

字体:11 磅, 字体颜色: 文字 1

字体:11 磅, 字体颜色: 文字 1

缩进: 首行缩进: 0 字符, 定义网格后自动调整右缩进, 在相同样式的段落间添加空格, 孤行控制, 调整中文与西文文字的间距, 调整中文与数字的间距, 制表位: 0 字符, 列表制表位

字体: 11 磅, 字体颜色: 文字 1

字体: 11 磅, 字体颜色: 文字 1

字体: (默认) Times New Roman, 英语(澳大利亚)

字体: (默认) Times-Roman, 11 磅, 字体颜色: 文字 1

缩进: 首行缩进: 2 字符

字体: (默认) Times-Roman, 11 磅, 字体颜色: 文字 1

字体: (默认) Times-Roman, 11 磅, 字体颜色: 文字 1

字体: (默认) Times-Roman, 11 磅, 字体颜色: 文字 1

字体: (默认) Times-Roman, 11 磅, 字体颜色: 文字 1

字体: (默认) Times-Roman, 11 磅, 字体颜色: 文字 1

字体: (默认) Times-Roman, 11 磅, 字体颜色: 文字 1

字体: (默认) Times-Roman, 11 磅, 字体颜色: 文字 1

字体: (默认) Times-Roman, 11 磅, 字体颜色: 文字 1

字体: (默认) Times-Roman, 11 磅, 字体颜色: 文字 1

字体: (默认) Times-Roman, 11 磅, 字体颜色: 文字 1

字体: (默认) Times New Roman, (中文) +中文正文 (宋体), 字体颜色: 文字 1

字体: 小五, 字体颜色: 文字 1

字体: (默认) Times New Roman, 小五, 字体颜色: 文字 1

字体: 小五, 字体颜色: 文字 1

字体颜色: 文字 1, 下标

字体颜色: 文字 1, 下标

字体颜色: 文字 1, 下标

字体颜色: 文字 1, 下标

字体颜色: 文字 1, 下标

字体颜色: 文字 1, 下标

字体颜色: 文字 1, 下标

字体颜色: 文字 1, 下标

字体颜色: 文字 1

字体颜色: 文字 1

字体颜色: 文字 1

字体颜色: 文字 1

字体: 小五, 字体颜色: 文字 1

字体: +西文正文 (Calibri), 小五, 非加粗, 字体颜色: 文字 1

左, 定义网格后自动调整右缩进, 行距: 固定值 12 磅, 到齐到网格

字体: 小五, 字体颜色: 文字 1

字体: 小五, 字体颜色: 文字 1

字体: 小五, 字体颜色: 文字 1

字体: 小五, 字体颜色: 文字 1

字体: 小五, 字体颜色: 文字 1

字体: 小五, 字体颜色: 文字 1

字体: 小五, 字体颜色: 文字 1

|---|---|---|

字体: 小五, 字体颜色: 文字 1

|---|---|---|

字体: 小五, 字体颜色: 文字 1

|---|---|---|

字体: 小五, 字体颜色: 文字 1

|---|---|---|

字体: 小五, 字体颜色: 文字 1

|---|---|---|

字体: 小五, 字体颜色: 文字 1

|---|---|---|

字体: 小五, 字体颜色: 文字 1

|---|---|---|

字体: 小五, 字体颜色: 文字 1

|---|---|---|

字体: 小五, 字体颜色: 文字 1

|---|---|---|

字体: 小五, 字体颜色: 文字 1

|---|---|---|

字体: 小五, 字体颜色: 文字 1

|---|---|---|

字体: 小五, 字体颜色: 文字 1

|---|---|---|

字体: 小五, 字体颜色: 文字 1

|---|---|---|

字体颜色: 文字 1

|---|---|---|

字体颜色: 文字 1

|---|---|---|

字体: 小五, 字体颜色: 文字 1

|---|---|---|

字体: 小五, 字体颜色: 文字 1

|---|---|---|

字体: 小五, 字体颜色: 文字 1

|---|---|---|

字体: 小五, 字体颜色: 文字 1

|---|---|---|

字体: 小五, 字体颜色: 文字 1

|---|---|---|

字体: 小五, 字体颜色: 文字 1

|---|---|---|

字体: 小五, 字体颜色: 文字 1

|---|---|---|

字体: 小五, 字体颜色: 文字 1

|---|---|---|

字体: 小五, 字体颜色: 文字 1

|---|---|---|

字体颜色: 文字 1

|---|---|---|

字体颜色: 文字 1

|---|---|---|

字体: 小五, 字体颜色: 文字 1

|---|---|---|

字体: 小五, 字体颜色: 文字 1

|---|---|---|

字体: 小五, 字体颜色: 文字 1

|---|---|---|

字体: 小五, 字体颜色: 文字 1

|---|---|---|

字体: 小五, 字体颜色: 文字 1

| | | |
|---|---|---|

字体: 小五, 字体颜色: 文字 1

| | | |
|---|---|---|

字体: 小五, 字体颜色: 文字 1

| | | |
|---|---|---|

字体: 小五, 字体颜色: 文字 1

| | | |
|---|---|---|

字体: 小五, 字体颜色: 文字 1

| | | |
|---|---|---|

字体: 小五, 字体颜色: 文字 1

| | | |
|---|---|---|

字体: 小五, 字体颜色: 文字 1

| | | |
|---|---|---|

字体: 小五, 字体颜色: 文字 1

| | | |
|---|---|---|

字体: 小五, 字体颜色: 文字 1

| | | |
|---|---|---|

字体: 小五, 字体颜色: 文字 1

| | | |
|---|---|---|

字体: 小五, 字体颜色: 文字 1

| | | |
|---|---|---|

字体: 小五, 字体颜色: 文字 1

| | | |
|---|---|---|

字体: 小五, 字体颜色: 文字 1

| | | |
|---|---|---|

字体: 小五, 字体颜色: 文字 1

|---|---|---|

字体: 小五, 字体颜色: 文字 1

|---|---|---|

字体: 小五, 字体颜色: 文字 1

|---|---|---|

字体: 小五, 字体颜色: 文字 1

|---|---|---|

字体: 小五, 字体颜色: 文字 1

|---|---|---|

字体: 小五, 字体颜色: 文字 1

|---|---|---|

字体: 小五, 字体颜色: 文字 1

|---|---|---|

字体: 小五, 字体颜色: 文字 1

|---|---|---|

字体: 小五, 字体颜色: 文字 1

|---|---|---|

字体: 小五, 字体颜色: 文字 1

|---|---|---|

字体: 小五, 字体颜色: 文字 1

|---|---|---|

字体颜色: 文字 1, 上标

|---|---|---|

字体颜色: 文字 1, 上标

|---|---|---|

字体颜色: 文字 1, 非上标/ 下标

| --- | --- | --- |

字体颜色: 文字 1, 非上标/ 下标

| --- | --- | --- |

非上标/ 下标

| --- | --- | --- |

非上标/ 下标

| --- | --- | --- |

非上标/ 下标

| --- | --- | --- |

上标

| --- | --- | --- |

上标

| --- | --- | --- |

字体颜色: 文字 1, 非上标/ 下标

| --- | --- | --- |

字体颜色: 文字 1, 非上标/ 下标

| --- | --- | --- |

字体颜色: 文字 1, 非上标/ 下标

| --- | --- | --- |

字体颜色: 文字 1, 非上标/ 下标

| --- | --- | --- |

字体颜色: 文字 1, 非上标/ 下标

| --- | --- | --- |

上标

| --- | --- | --- |

上标

|---|---|---|

字体颜色: 文字 1

|---|---|---|

字体颜色: 文字 1

|---|---|---|

字体颜色: 文字 1

|---|---|---|

字体颜色: 文字 1, 非上标/ 下标

|---|---|---|

字体颜色: 文字 1, 非上标/ 下标

|---|---|---|

字体颜色: 文字 1, 非上标/ 下标

|---|---|---|

字体颜色: 文字 1, 上标

|---|---|---|

字体颜色: 文字 1, 上标

|---|---|---|

字体颜色: 文字 1, 非上标/ 下标

|---|---|---|

字体颜色: 文字 1, 非上标/ 下标

|---|---|---|

字体颜色: 文字 1, 非上标/ 下标

|---|---|---|

字体颜色: 文字 1, 非上标/ 下标

|---|---|---|

字体: (默认) Times New Roman, 字体颜色: 文字 1

|---|---|---|

字体: (默认) Times New Roman, 字体颜色: 文字 1

|---|---|---|

字体: (默认) Times New Roman, 字体颜色: 文字 1

|---|---|---|

字体: (默认) Times New Roman, 字体颜色: 文字 1

|---|---|---|

字体: (默认) Times New Roman, 字体颜色: 文字 1

|---|---|---|

字体: (默认) Times New Roman, 字体颜色: 文字 1

|---|---|---|

字体: (默认) Times New Roman, 字体颜色: 文字 1

|---|---|---|

字体颜色: 文字 1, 非上标/ 下标

|---|---|---|

字体颜色: 文字 1, 非上标/ 下标

|---|---|---|

字体颜色: 文字 1, 非上标/ 下标

|---|---|---|

字体颜色: 文字 1, 非上标/ 下标

|---|---|---|

字体颜色: 文字 1, 非上标/ 下标

|---|---|---|

字体颜色: 文字 1, 非上标/ 下标

|---|---|---|

字体颜色: 文字 1, 非上标/ 下标

|---|---|---|

字体颜色: 文字 1, 非上标/ 下标

|---|---|---|

字体颜色: 文字 1, 非上标/ 下标

|---|---|---|

字体颜色: 文字 1, 非上标/ 下标

|---|---|---|

字体颜色: 文字 1, 非上标/ 下标

|---|---|---|

字体颜色: 文字 1, 非上标/ 下标

|---|---|---|

字体颜色: 文字 1, 非上标/ 下标

|---|---|---|

字体颜色: 文字 1, 非上标/ 下标

|---|---|---|

字体颜色: 文字 1, 非上标/ 下标

|---|---|---|

字体颜色: 文字 1

|---|---|---|

字体颜色: 文字 1

|---|---|---|

字体颜色: 文字 1

|---|---|---|

字体颜色: 文字 1

|---|---|---|

字体颜色: 文字 1

字体颜色: 文字 1

字体颜色: 文字 1

字体颜色: 文字 1

---

## Author Response (AR2)

**Response to** Topical Editor (**Dr.** Fiona O'Connor)

*We thank your insightful comments. We revised our manuscript accordingly.*

1. The model source code is "freely available upon request". This does not meet the requirements of the journal. If the source code is freely available, can you please archive it in accordance with GMD policy and update the code/data availability section of your manuscript? If it cannot be published, then you will need to indicate the licence conditions in the manuscript.

Response:
*We have not finished the preparation of a user guide how to run the BCC-ESM1 model, and some source codes at the present version would be difficult for users to comprehend. So, we have modified the statements in section of "Code and data availability" as "The source codes of BCC-ESM1 model are available for use under licence agreement. Readers interested in BCC-ESM1 codes may contact Dr. Tongwen Wu (twwu@cma.gov.cn) for further details."*

2. Model outputs on ESGF is very good practice but can be difficult to find. ESGF provides the correct data citation for each piece of data on the corresponding catalogue page, so can I please ask that you precisely cite the data? Precise instructions for citing CMIP6 data are here: https://docs.google.com/document/d/1SnwBL9MJQNEU1_nJ661SN3-SSxR0j1mAYan6-WXfaSU/edit

*Response: We have added the data citation for CMIP6 data on ESGF.*

3. The model input files required to reproduce the simulations are not referred to. Again, may I please ask that you publicly archive these and update the code/data availability section accordingly?

*Response: We have modified the statements as "Readers interested in BCC-ESM1 codes and the model input files required to reproduce the simulations may contact Dr. Tongwen Wu (twwu@cma.gov.cn) for further details."*

You can find full details of the requirements of the journal for model description papers at:
https://www.geoscientific-model-development.net/about/manuscript_types.html#item1

Thirdly, on reading the reviewer comments, your responses to them, and the revised manuscript, I feel that you have addressed the majority of comments from Reviewer #1 with one exception as follows.

Reviewer #1 raised an important point about the imbalance in the TOA radiative fluxes in BCC-ESM. Although you cite Wild et al. (2013) to support that such an imbalance is common among other models, the comparison of models and the observational dataset used in that paper is referring to the present day, in which case an imbalance would be expected. However, in a pre-industrial control experiment as described in your manuscript, there is little anthropogenic forcing and as such, the TOA is expected to be more in balance. You clearly show little drift in surface temperature and sea surface temperatures in your Figure 1 but can you confirm whether BCC-ESM is indeed conserving energy?

*Response:*
*Yes, +0.7 Wm-2 net energy flux at TOA in BCC-ESM1 means that BCC-ESM1 has not absolute energy balance in the whole system. So we added a sentence as "It means that there exists surplus energy of 0.7 Wm-2 obtained by the whole system, but do not cause remarkable climate drift in BCC-ESM1."*

In relation to Reviewer #2, they had concerns about the level of detail in the description of the aerosol scheme and how it compared with other existing schemes, and noted that the evaluation of the scheme itself in BCC-ESM1 was mainly qualitative and when quantitative, referred to quite old references.

I think the additional comparisons that you've made (e.g. HIPPO, Taylor plot) and the inclusion of more quantitative measures of skill (e.g. Table 7) add to the quality of the evaluation and the manuscript.

*Response: Following your suggestion, we have added quantitative evaluation for Black Carbon using HIPPO aircraft observations in Table 7. As HIPPO data spans a short time period, the Taylor plot is not made.*

However, I would ask that you include information on the requested optical properties used (e.g. extinction co-efficient, asymmetry parameter etc..).

*Response:*
*We have added a paragraph in section 2.5 to describe the information on requested optical properties of aerosols used. "The treatment of aerosol single scattering (optical) properties (such as mass extinction efficiency, single scattering albedo, and asymmetric factor) follows the look-up table approach in CAM [Collins et al., 2004]. The optics for black, organic carbon, sea salt, and sea salt particles is assumed to be same as the optics for soot and water-soluble aerosols in the Optical Properties of Aerosols and Clouds (OPAC) data set [Hess et al., 1998]. The optics for dust is derived by Mie calculations for the size distribution represented by each size bin [Zender et al., 2003]. Similarly, for sulfate and nitrate particles, same set of aerosol optical properties for ammonium sulfate are used and are taken from Wang*

*et al. [2008] with treatment of aerosol hygroscopicity. The volcanic stratospheric aerosols are assumed to be comprised of 75% sulfuric acid and 25% water, as in Hess et al. [1998]. For each model year, different aerosol types are assumed to be externally mixed in the calculation of bulk aerosol single scattering properties that are in turn used in the radiative transfer calculations."*

Finally, I also have a number of specific comments below that I ask you to implement.

Specific Comments:
1. When referring to GHGs, replace NO2 with N2O? (Line 86 of revised manuscript) and give full names e.g. nitrous oxide (N2O)

*Response: NO2 has replaced by N2O. Its full name appears first in line 52.*

2. Please give complete names. As an example, replace CH2O with "formaldehyde (CH2O)" (Lines 142-144) and do likewise for all species referred to in the manuscript.

*Response: Modified for all species.*

3. Replace "Wet removals …. are …" with "Wet removal … is …" (Line 138)
*Response: Modified.*

4. Please include appropriate references for CMIP6 biomass burning emissions, GHGs and solar forcing (e.g. van Marle et al., 2017)

*Response: We have added the references of CMIP6 forcing data used in lines 315 to 326*

5. The protocol for the CMIP6 coupled historical experiment is from Eyring et al. (2016). AerChemMIP (Collins et al., 2017) is using those historical ensemble members as controls for parallel sensitivity experiments. Please remove Collins et al. (2017) as the reference for the coupled historical simulation protocol (Line 307-308)

*Response: Following your comment, to remove Collins et al. (2017) as the reference for the coupled historical simulation protocol. We have rewritten this sentence to "The historical experiment is forced with emissions evolving from 1850 to 2014 include biomass burning emissions (Van Marle et al. 2017), anthropogenic and open burning emissions (Hoesly et al., 2017; Hoesly et al., 2018; Feng et al., 2019)"*

6. For N2O and CH4, it is still unclear whether they are only prescribed at the surface (and at the top boundary) and their concentrations in the remaining atmosphere are interactive. When you say "CH4, N2O, CO2, CFC11, and CFC12 are prescribed using CMIP6 historical forcing data as suggestion in AerChemMIP protocol. Although CH4 and N2O are prognostic variables in the chemistry scheme (Table 1), their prognostic values at each model step in the historical experiment are replaced by CMIP6 data", are their concentrations replaced throughout the model domain or only at the surface and top boundary? Please clarify. Again here, Collins et al. (2017) and AerChemMIP are not the appropriate reference for the coupled historical protocol (Lines 310-313 and 378-380)

*Response: The original purpose of our work is to follow the AerChemMIP described in Collins et al., (2017). The legend of Table 1 in Collins et al., (2017) is* **"… Models should always be run with the maximum complexity available. The species columns refer to the specifications for concentrations (CH4, N2O and CFC/HCFC) or emissions (aerosol and ozone precursors). "Hist" means the concentrations or emissions should evolve as for the CMIP6 historical simulation …".**
*We understand that as "concentrations of CH4, N2O, CFC are specified for the coupled historical experiment of AerChemMIP". Anyway, we have removed "as suggestion in AerChemMIP protocol" in lines 310 to 331 and "the reference of Collins et al. (2017) in line 318 to 320.*

7. Give full name for ACCMIP (Line 333)
*Response: Modified.*

8. Change "eruption" to "eruptions" in Section heading (Line 354)
*Response: Modified.*

[revised manuscript text omitted]

---

## Author Response (AR3)

**Response to** Topical Editor (**Dr.** Fiona O'Connor)

**Comments**:

 "…. Having read your response and your revised manuscript, and after consulting with the executive editors of GMD, I am writing to confirm that I can only recommend publication in GMD once you have adhered to GMD's strict policy on model code availability. …."

**Response**:

Following the GMD's policy, we have modified the section of "6. Code and data availability" as: "
[revised manuscript text omitted]

---

## Author Response (AR4)

**Response to** Topical Editor (**Dr.** Fiona O'Connor)

**Comments**:

 "…. Thank you for making the BCC climate model source code and input files available through a repository, including the provision of a doi.

However, I note that my request to precisely cite the model output data from the CMIP6 archive hasn't been addressed. May I please ask that you correct this, so that the revised manuscript fully complies with the GMD data policy?

As mentioned previously, model output on ESGF is very good practice but can be difficult to find. ESGF provides the correct data citation for each piece of data on the corresponding catalogue page. Precise instructions for citing CMIP6 data are here: https://docs.google.com/document/d/1SnwBL9MJQNEU1_nJ661SN3-SSxR0j1mAYan6-WXfaSU/edit

As soon as I'm satisfied that the model output is being cited precisely, I will accept your manuscript for final publication in Geoscientific Model Development. "

**Response**:

Dear **Dr.** Fiona O'Connor

Thanks for your helpful comments. We have modified the description in the section of "6. Code and data availability" as: "…Model output data of BCC CMIP6 AerChemMIP simulations described in this paper are available on the Earth System Grid Federation (ESGF) (https://cera-www.dkrz.de/WDCC/ui/cerasearch/cmip6?input=CMIP6.AerChemMIP.BCC.BCC-ESM1, https://doi.org/10.22033/ESGF/CMIP6.1733; Zhang et al., 2019) …" That modification maybe help readers to easily find the BCC model output data.

Best regards,

Tongwen Wu, and Co-authors.

[revised manuscript text omitted]